# Recent Advances in SARS-CoV-2 Main Protease Inhibitors: From Nirmatrelvir to Future Perspectives

**DOI:** 10.3390/biom13091339

**Published:** 2023-09-02

**Authors:** Andrea Citarella, Alessandro Dimasi, Davide Moi, Daniele Passarella, Angela Scala, Anna Piperno, Nicola Micale

**Affiliations:** 1Department of Chemistry, University of Milan, Via Golgi 19, 20133 Milano, Italy; alessandro.dimasi@unimi.it (A.D.); daniele.passarella@unimi.it (D.P.); 2Department of Chemical and Geological Sciences, University of Cagliari, S.P. 8 CA, 09042 Cagliari, Italy; davide.moi2@gmail.com; 3Department of Chemical, Biological, Pharmaceutical and Environmental Sciences, University of Messina, Viale Ferdinando Stagno D’Alcontres 31, 98166 Messina, Italy; ascala@unime.it (A.S.); apiperno@unime.it (A.P.)

**Keywords:** COVID-19, SARS-CoV-2 M^pro^, protease inhibitors, coronavirus, peptidomimetics, Paxlovid, nirmatrelvir

## Abstract

The main protease (M^pro^) plays a pivotal role in the replication of severe acute respiratory syndrome coronavirus 2 (SARS-CoV-2) and is considered a highly conserved viral target. Disruption of the catalytic activity of M^pro^ produces a detrimental effect on the course of the infection, making this target one of the most attractive for the treatment of COVID-19. The current success of the SARS-CoV-2 M^pro^ inhibitor Nirmatrelvir, the first oral drug for the treatment of severe forms of COVID-19, has further focused the attention of researchers on this important viral target, making the search for new M^pro^ inhibitors a thriving and exciting field for the development of antiviral drugs active against SARS-CoV-2 and related coronaviruses.

## 1. Introduction

Since the outbreak of the COVID-19 pandemic, a considerable part of the pharmaceutical research has directed its efforts towards the discovery of new and effective antiviral drugs with the ultimate goal of preventing (or at least being prepared for) further dramatic health emergencies [1].

The following review includes the most promising scientific discoveries since 2021 concerning the synthesis and biological evaluation of SARS-CoV-2 M^pro^ inhibitors and aims to update the state-of-the-art about COVID-19 treatments with a future perspective on other related coronaviral diseases. A detailed discussion of the structure and role of SARS-CoV-2 M^pro^ during the viral replication of SARS-CoV-2 has already been addressed in our previous review [2]. Then, adequate attention will be paid to the newly discovered molecules, starting from **Paxlovid^®^**, the first oral antiviral drug developed by Pfizer for the treatment of COVID-19, which consists of a combination of **Nirmatrelvir** (SARS-CoV-2 M^pro^ inhibitor) and **Ritonavir** (HIV-1 protease inhibitor) [3]. The purpose of this review is to analyze the recent results of the scientific literature. Examples of significant advances in patents dealing with SARS-CoV-2 M^pro^ inhibitors can be found extensively elsewhere [4].

## 2. SARS-CoV-2 Mpro as a Drug Target: Structural Features and Proteolytic Mechanism

SARS-CoV-2 M^pro^ came out as a topical antiviral drug target due to several unique characteristics. In general, the combined action of SARS-CoV-2 M^pro^ and SARS-CoV-2 PL^pro^ constitutes a key aspect for the virus replication. The inhibition of their activity leads to the inability of the virus to replicate and infect other cells or subsequent hosts [5]. Indeed, M^pro^ and PL^pro^ were considered plausible drug targets since the evolving of the SARS and MERS pandemics due to their active role in the post-translational processing of viral polyproteins [6]. By comparing the primary sequences of M^pro^ and PL^pro^ of SARS-CoV, MERS-CoV and SARS-CoV-2, the amino acid sequences were observed to be highly conserved among these three viruses [5]. This means that, despite the high mutation rate of coronaviruses, M^pro^ and PL^pro^ tend to be conserved as mutations of these proteases could be fatal to the virus [7]. Therefore, the risk of developing drug resistance is reduced for M^pro^ and PL^pro^ inihibitors [8]. PL^pro^ is endowed with a high degree of structural similarity with various human cysteine proteases. As a consequence, the development of PL^pro^ inhibitors may lead to cross-reactivity and consequent drawbacks [2]. M^pro^ instead has no homologous human proteases [9]. Thus, its proteolytic activity could be specifically inhibited with no significant impact on host cells [10]. In this perspective, M^pro^ has been considered a more attractive target as compared to PL^pro^ for anti-SARS-CoV-2 drug development. Nowadays, more than 400 crystal structures of SARS-CoV-2 M^pro^ appear in the Protein Data Bank (PDB) database [11]. The native structure of SARS-CoV-2 M^pro^ is shown in Figure 1A (PDB: 7ALH) [12]. The active form of M^pro^ is given by two identical monomers (Figure 1B) of approximately 34.21 kDa, oriented with a right angle to each other to create the active homodimer [11]. Monomers are formed by three domains: domains I and II are six-stranded antiparallel β-barrels that host the active site, whereas domain III contains an antiparallel globular cluster formed by five α-helices that promote the dimerization process [13]. The crystal structure of the SARS-CoV-2 M^pro^ monomer is shown in Figure 1B (PDB: 1P9S) [14].

The active site of SARS-CoV-2 M^pro^ is located at the interface of domain I and II of each monomer (Figure 2). However, in the dimeric form, only one catalytic site is active [11]. This active site can be divided into six sub-pockets, named S1′, S1, S2, S3, S4 and S5, to which correspond six substrate-bindings. S1, S2 and S4 pockets are well built in the protein cavity, while, on the top of the protein surface, the S1′, S3 and S5 subsites, which show an undefined shape [15], are found. SARS-CoV-2 M^pro^ is a cysteine protease that performs a hydrolytic cleavage mechanism due to a catalytic dyad formed by Cys145 and His41 residues. Normally, serine or cysteine proteases exploit a standard catalytic triad, formed by Cys/Ser, His and Asp; in this case, the absence of the third element is counterbalanced by the presence of a water molecule H-bonded to His41 [2]. Another water molecule stabilizes the oxoanionic hole, establishing H-bonds with Phe140, His163 and Glu166. M^pro^ is responsible for the cleavage of 11 sites of pro-protein pp1ab, occurring when a Gln residue is found in the P1 position [16]. In particular, the enzyme can recognize the sequence of Leu-Gln↓Ser-Ala-Gly, where ↓ indicates the amide bond position cleaved by M^pro^ [15]. This amino acid sequence is not recognized by any human proteases, which is why it is possible to design selective M^pro^ inhibitors [2]. 

## 3. SARS-CoV-2 Mpro Inhibitors: An Overview

Having ascertained the pivotal role of M^pro^ during the replicative cycle of coronaviruses and its relevance as a therapeutic target, the efforts in the development of selective inhibitors have been enormous [17]. Firstly, inhibitors of the active site (competitive inhibitors) were found to be more successful in comparison to the small group of allosteric ones (non-competitive inhibitors). Indeed, many attempts have been made to develop competitive inhibitors that may interact with the residues present in the active site in a covalent or non-covalent fashion [18,19]. Beyond the reproposition of drugs (which was reasonably the most immediate countermeasure used to stop the COVID-19 pandemic), the development of new SARS-CoV-2 M^pro^ competitive inhibitors has been mainly focused on peptidomimetics as compared to non-peptidyl compounds [2]. Peptidomimetics have the capacity of mimicking natural peptidic substrates and displaying at the same time superior pharmacokinetics characteristics [20]. During the first SARS outbreak, the peptidomimetic approach was widely used to develop new SARS-CoV M^pro^ inhibitors. Thus, the same approach has been proposed to counteract the new COVID-19 pandemic [21]. Despite the high similarity between the two M^pros^, the accessibility of the active site can be quite different. Therefore, the development of non-peptidyl small molecules also turned out to be a viable strategy for developing novel SARS-CoV-2 M^pro^ inhibitors [22]. In general, both peptidomimetics and non-peptidyl compounds can act as covalent inhibitors due to the presence of an (*C*-terminal) electrophilic warhead able to bind, temporarily or permanently, the thiol group of the catalytic Cys145, leading to the inactivation of the enzyme. 

On the basis of the (*C*-terminal) electrophilic warhead, it is possible to classify the SARS-CoV-2 M^pro^ inhibitors into different groups:Nitriles;Aldehydes;Ketones;α-Haloacetamides;α-Ketoamides;Michael acceptors;Esters and thioesters;Selenium/sulfur-based derivatives.

In addition to the above conventional electrophilic warheads commonly found in a large number of cysteine protease inhibitors, there are various examples of original electrophilic fragments that have been implemented for the design of novel, more potent and selective inhibitors [23,24]. Besides competitive inhibitors, research also provides allosteric or non-competitive inhibitors of SARS-CoV-2 M^pro^. In fact, six allosteric sites have been found on SARS-CoV-2 M^pro^, whose functions can be reassumed in assisting the catalytic activity through substrate processing and dimerization phenomena [25]. The latter are fundamental for the activity of the enzyme; thus, allosteric inhibitors that target sites involved in the dimerization process seem to be very interesting for the inactivation of SARS-CoV-2 M^pro^. Three of these sites are located at the dimerization site (allosteric site 1), while the other three sites are located between domains II and III (allosteric site 2) to confer conformational freedom (Figure 3) [25].

A complete list of the inhibitors discussed in this review is reported in Appendix A.

## 4. Nitriles and Nirmatrelvir

The nitrile warhead has a moderate electrophilic character. The electron-poor carbon of the nitrile group is able to undergo nucleophilic attack from the thiol group of the SARS-CoV-2 M^pro^ Cys145 residue to afford a reversible covalent thioimidate adduct [17]. In 2013, Chuck et al. demonstrated that nitrile-based peptidomimetics have broad-spectrum inhibition toward human coronaviruses M^pro^ [26]. This function has been taken into consideration by Pfizer’s scientists to develop the first oral drug for the treatment of COVID-19) [27]. This drug contains a SARS-CoV-2 M^pro^ bearing a nitrile warhead in P1′ called **Nirmatrelvir** (compound **1** in Figure 4). Beyond the P1′ warhead, **Nirmatrelvir** embodies a P1 γ-lactam moiety that resembles the Gln residue of the native substrate and acts as a recognition motif for the S1 pocket. The introduction of the dimethylcyclopropyl proline moiety at the P2 position was made to achieve a deep fit within the S2 subsite. The *N*-terminal recognition unit is composed of a trifluoroacetamide group that provides interaction with the S3/S4 subpocket and gives the compound better pharmacokinetic properties. The discovery and development of **Nirmatrelvir** started from two hit compounds: **PF-00835231** (**2** in Figure 4), an α-hydroxy ketone-based peptidomimetic, and the corresponding phosphate prodrug **PF-07304814** (**3** in Figure 4), developed by Pfizer Inc (NYSE: PFE). against SARS-CoV during the 2002 SARS outbreak [27]. Both compounds were selected as SARS-CoV-2 M^pro^ inhibitors for in vitro and in vivo biological evaluation, showing promising results as intravenously administered drug candidates. In particular, **PF-00835231** showed high potency (*K*_i_ = 4 nM) and good antiviral activity (EC_50_ = 231 nM) [28].

Owen et al. designed and synthesized a series of **PF-00835231** analogues in order to improve its pharmacokinetic properties and obtain compounds suitable for oral administration [28]. The rationale of the design was to decrease the number of H-bond donors (HBD) without affecting the drug–target interaction. The first modification involved the α-hydroxy ketone warhead, replaced with two new electrophilic moieties, namely a benzothiazole-7-yl ketone and a nitrile group. The nitrile derivate (**4** in Figure 5) showed higher oral absorption in rat models but lower enzymatic inhibitory and cell-based antiviral activity than **2** (*K*_i_ = 27.7 nM; EC_50_ = 1.4 µM) [28]. Another HBD that was not critical for the drug–target interaction was the leucine residue at P2, replaced with a cyclically modified proline residue (6,6-dimethyl-3-azabicyclo [3.1.0]hexane) able to fit inside the S2 pocket. This modification, combined with the benzothiazole-7-yl ketone warhead in the P1′ position, led to compound **5** (Figure 5). In this case, the cell permeability increased, while enzymatic inhibitory and cell-based antiviral activity decreased as compared to compound **4** (*K*_i_ = 230 nM; EC_50_ = 5.6 µM), probably due to the loss of a key H-bond with Gln189 in the S2 subsite. To regain potency, further investigations led to the replacement of the indole ring in P3 to reach a better fit with the S3 subsite. Two branched acyclic moieties were proposed: a methanesulfonamide and a trifluoroacetamide functionality (compounds **6** and **7**, respectively; Figure 5). Compounds **6** and **7** showed comparable *K*_i_ values (12.1 nM vs. 7.93 nM) but different antiviral activity (**6**: EC_50_ = 909 nM; **7**: EC_50_ = 85.3 nM). Moreover, **7** showed much higher oral absorption than **6** in rat and monkey in vivo models. The combination of the nitrile warhead in P1′ with the trifluoroacetamide moiety in P3 eventually led to the identification of **PF-07321332**, i.e., **Nirmatrelvir** (**1** in Figure 4).

The co-crystal structure of **Nirmatrelvir** with SARS-CoV-2 M^pro^ clarified its mechanism of action. The nitrile warhead in P1′ forms a reversible covalent bond with Cys145 at S1′, with the formation of the thioimidate adduct that establishes a H-bond with the Gly143 residue. The γ-lactam moiety interacts with His163 and Glu166 in the S1 pocket, while the aliphatic dimethyl-bicycloproline fits inside the S2 pocket via the formation of several van der Waals interactions with the surrounding apolar residues. Moreover, the carbonyl oxygen of the trifluoromethyl acetamide acts as a H-bond acceptor toward the Gln189 residue, while the trifluoromethyl group establishes interactions with the S4 pocket (Figure 6) [17].

The nitrile moiety was selected over the benzothiazole-7-yl ketone warhead due to several features: (*i*) superior water solubility allowing for high-concentration solutions of a drug candidate in pre-clinical trials; (*ii*) reduced tendency of epimerization of the near P1 stereocenter; (*iii*) easier synthesis for a scale-up process [27]. **Nirmatrelvir** showed superior enzymatic inhibitory and antiviral activity (*K*_i_ = 3.11 nM; EC_50_ = 74.5 nM) as compared to the parent compounds (Figure 4). The introduction of a structure with fewer HBDs and a trifluoroacetamide moiety assured good gut permeability, oral bioavailability and proper drug clearance in in vivo rat models. In vivo evaluations in monkey models showed a decrease in oral bioavailability and gut permeability linked to the first-pass metabolism [28]. To decrease the first-pass metabolism, **Nirmatrelvir** is co-administered with the anti-HIV-1 drug **Ritonavir**, which is also an inhibitor of CYP450. Remarkable results in preclinical and clinical studies led this new combination to be authorized by the Food and Drug Administration (FDA) [29], European Medicine Agency (EMA) [30] and UK Medicines and Healthcare products Regulatory Agency (MHRA) as the only orally administered drug to treat COVID-19 [31] under the commercial name of **Paxlovid^®^**.

The success of the nitrile warhead on **Nirmatrelvir** led Pfizer scientists to investigate other nitrile-based compounds as inhibitors of SARS-CoV-2 M^pro^, resumed in a new patent published on 16 December 2021, identifying compound **8** (Figure 7) as a new SARS-CoV-2 M^pro^ inhibitor (*K*_i_ = 4 nM; EC_50_ = 19 nM) [32]. Bai et al. explored other nitrile analogs, investigating the 4-methoxyindole moiety at P3 [33]. Among all products, compound **9** (Figure 7), having a 6-chloro-4-methyloxyindole function at P3, showed the best results in terms of in vitro enzymatic inhibitory and antiviral activity (IC_50_ = 9 nM; EC_50_ = 2.2 μM) [25].

Starting from **Nirmatrelvir**, Zhu et al. developed a series of novel SARS-CoV-2 M^pro^ inhibitors by optimizing the P2 and P4 positions of the peptide framework [34]. Within this series, compounds **10** and **11** (Figure 8) exhibited optimal inhibitory activity toward SARS-CoV-2 M^pro^, with IC_50_ values of 18 nM and 22 nM, respectively. Compound **10** derived from an optimization study of the P2 site in which the dimethyl-bicycloproline of **Nirmatrelvir** was replaced with a cyclopentenyl proline. Compound **11** derived from an optimization attempt of the P4 site with the aim of improving the metabolic stability. In particular, the trifluoromethyl group of **Nirmatrelvir** was replaced with a trifluoromethanesulfonyl group. Compounds **10** and **11** exhibited higher antiviral activity than **Nirmatrelvir** against SARS-CoV-2-infected VeroE6 cells, with EC_50_ values of 313 nM and 170 nM, respectively. They also exhibited better metabolic stability than **Nirmatrelvir** and similar PK properties [34].

Inspired by the structures of hepatitis C virus NS3/4A serine protease inhibitors **Narlaprevir** and **Boceprevir**, and SARS-CoV M^pro^ inhibitors such as **GC-376** (Figure 9), Kneller et al. developed three SARS-CoV-2 M^pro^ inhibitors bearing a nitrile warhead [35]. Compounds **12** and **13** (the two most active derivatives), called **BBH-2** and **NBH-2**, respectively (Figure 9), were designed by replacing the *C*-terminal di-ketoamide group of **Boceprevir** and **Narlaprevir**, respectively, with a nitrile warhead, whereas the hydrophobic P1 groups present in the serine protease inhibitors were replaced in both derivatives with a γ-lactam ring as per **GC-376**.

These compounds have been shown to have a high binding affinity toward M^pro^, comparable to that of **Nirmatrelvir** (*K*_d_ values of 26 nM and 30 nM for **12** and **13**, respectively, and 7 nM for **Nirmatrelvir**). Furthermore, the antiviral activity of these inhibitors were measured in SARS-CoV-2 M^pro^-infected VeroE6 TMPRSS cells in the presence of a P-glycoprotein inhibitor (CP-100346), exhibiting EC_50_ values of 0.88 and 1.82 μM for **12** and **13**, respectively [35]. No cytotoxicity was detected at 10 μM. The X-ray crystal structures of SARS-CoV-2 M^pro^ in complex with **12** and **13** (Figure 10) showed that these two inhibitors bind in the same way to the active site, forming a covalent bond between Cys145 and the nitrile warhead, and the newly formed thioimidate group is involved in a H-bond with the Cys145 backbone. The γ-lactam ring at P1 forms H-bonds with His163 and Glu166 main chains and a H-bond with the His163 side chain. The other interactions are hydrophobic or involve a H-bond with Glu166 [35].

Brewitz et al. developed different **Nirmatrelvir** derivatives by accomplishing the isoelectronic replacement nitrile/alkyne group [36]. The most promising compounds of this series turned out to be the alkyne derivate **14** and the CF_3_-capped alkyne **15** (Figure 11). They displayed noteworthy SARS-CoV-2 M^pro^ inhibitory and antiviral activity, high selectivity (no activity against SARS-CoV-2 PL^pro^) and a low cytotoxicity profile. MS analyses and crystallographic studies demonstrated that, unlike nitrile derivates that inhibit M^pro^ in a reversable manner, these alkyne derivatives inhibit M^pro^ by an apparent irreversible covalent mechanism, forming an internal vinyl thioether adduct with the Cys145 residue [37].

## 5. Aldehydes

The aldehyde functional group is considered the most widely used electrophilic warhead among covalent SARS-CoV-2 M^pro^ inhibitors due to the susceptibility of the electron-poor carbonyl carbon to undergo nucleophilic addition by the thiol group of the Cys145 residue, with the formation of a reversible hemi-thioacetal adduct [2]. The latter highly resembles the tetrahedral intermediate of the hydrolysis of the endogenous substrate, ensuring a longer residence time and an enhanced I-E complex stability due to several H-bonds between the adduct and residues of the S1′ pocket. Furthermore, the hemithioacetal adduct is able to stabilize the ligand–target interaction, acting as a H-bond donor toward the Cys145 backbone. This critical H-bond is normally provided in the natural substrate by a carbonyl oxygen of the amide group next to a Gln residue [38]. Thus, this critical interaction can be provided only by carbonyl- and ketoamide-based inhibitors, and this could explain their higher efficacy in comparison to other electrophilic warheads. The first peptidomimetics bearing an aldehyde warhead targeting SARS-CoV-2 M^pro^ were compounds **16** and **17** (Figure 12), designed and synthesized by Dai et al. [39]. The work was mainly devoted to explore the enzyme S2 site. Compound **16** has a cyclohexyl ring at P2, while, in compound **17**, we find a 3-fluorophenyl ring. Besides the aldehyde warhead at P1′, other common features are the 2-indole moiety at P3 and the (*S*)-γ-lactam group at P1. Also, compound **18** (Figure 12) was synthetized later on by Dai et al., and it turned out to be a broad-spectrum M^pro^ inhibitor of enterovirus and SARS-CoV-2 [40].

The prementioned aldehyde derivatives exhibited a high enzymatic inhibitory and antiviral activity (**16**: IC_50_ = 53 nM and EC_50_ = 0.53 μM; **17**: IC_50_ = 40 nM and EC_50_ = 0.72 μM; **18**: IC_50_ = 34 nM and EC_50_ = 0.29 μM) [40]. From the X-ray analysis of the structures of SARS-CoV-2 M^pro^ in complex with compounds **17** and **18**, it was possible to observe the main interactions involved in the mechanism of action: a C-S covalent bond was detected between the carbonyl carbon of the warhead and the thiol group of the Cys145 of the S1′ pocket; this adduct is stabilized by an additional H-bond between the thio-hemiacetal hydroxyl group and the Cys145 backbone. The (*S*)-γ- lactam moiety at P2 is deeply inserted into the S2 subsite, forming several H-bonds with key residues of this pocket, while the indole group at P3 interacts via a H-bond with the Glu166 residue located in the surface of S3/S4 sites. The binding modes of compounds **16** and **17** are quite similar, except for the interaction with S3, involving two different P3 moieties. In compound **17**, the 3-fluorophenyl group undergoes a downward rotation in the S2 pocket, unlike the cyclohexyl substituent in **16**, due to an additional H-bond involving the fluorine atom and the Gln189 residue and several hydrophobic interactions of the aromatic ring with the surrounding residues. Compound **16** was chosen as a potential drug candidate due to its better pharmacokinetic properties [39].

Other examples of aldehyde-based SARS-CoV-2 M^pro^ inhibitors are **GC373** (i.e., compound **19** in Figure 13A) and its bisulfite prodrug **GC376** (i.e., compound **20** in Figure 13A) [41]. These compounds were initially used to treat feline infectious peritonitis, caused by feline coronaviruses FCoV, and then they were repurposed as SARS-CoV-2 M^pro^ inhibitors. In both compounds, there is a benzyl moiety at P3, a leucine residue at P2 and a (*S*)-γ-lactam group at P1. Compounds **19** and **20** inhibited the activity of SARS-CoV-2 M^pro^, with an IC_50_ value of 0.40 and 0.19 μM, respectively. The X-ray crystal structure of SARS-CoV-2 M^pro^ in complex with both compounds confirmed the ability of the prodrug **20** to provide the aldehyde function since they showed an identical binding mode: the aldehyde warhead reacted covalently with the Cys145 thiol group to give a hemithioacetal adduct that is stabilized by several H-bonds inside the oxyanionic hole with Cys145 (Figure 13B) [41]. The P1 γ-lactam moiety forms a H-bond with His163 and Glu166 and interacts with the main chain of Phe140, while the P2 leucine is inserted into the P2 pocket, establishing hydrophobic interactions with Met149, His41 and Met49. The benzyl group at P3 interacts with the S3/S4 superficial sites by means of hydrophobic interactions. Both compounds exhibited high in vitro antiviral activity observed in VeroE6 cell infected with SARS-CoV-2 (**19:** EC_50_ = 1.5 μM; **20**: EC_50_ = 0.9 μM) and low cell toxicity (CC_50_ > 100 μM for both compounds). Nevertheless, the use of the bisulfite prodrug **20** has been preferred due to better outcomes in terms of bioavailability.

Structural modifications of **20** were conducted in order to improve the in vitro antiviral activity against SARS-CoV-2 [41]. Specifically, its P2 position was redesigned by substituting the leucine side chain with a constrained cyclopropylmethyl ring, while its benzyl group at P3 was decorated with a *meta*-F substituent (i.e., compound **21** in Figure 14A) or elongated with a supplementary methylene group that connects a 3-chlorophenyl ring to the carbamate linkage (i.e., compound **22** in Figure 14A) [42]. Both compounds displayed improved in vitro inhibitory activity against M^pro^ as well as antiviral activity in comparison to the lead (**21**: IC_50_ = 0.07 and EC_50_ = 0.57 μM; **22**: IC_50_ = 0.08 μM and EC_50_ = 0.7 μM) [42]. The X-ray structure of the complex between M^pro^ and **21** indicated that the P2 cyclopropyl fragment is able to fit more deeply within the S2 subsite, while the substitution at P3 with a halogenated phenyl ring allowed the inhibitor to fit the S3/S4 pocket more deeply instead of remaining on the surface of the protease (Figure 14B) [42].

Other analogs of **20** with improved in vitro enzymatic inhibitory and antiviral activity are represented by compounds **23** and **24** (Figure 15) [43,44]. In compound **23**, also called **UAWJ247**, a structural variation in comparison to **20** involved the P2 position, where a phenyl ring replaced the isobutyl moiety. This compound exhibited an enzymatic inhibitory activity comparable to **20** (IC_50_ = 45 nM) and a moderate antiviral activity. Compound **24**, also called **NK01-63** or **Coronastat**, contains a 3-trifluoromethylbenzyl group as a replacement of the benzyl group at P3 present in **20**. Compound **24** showed a potent inhibitory activity against SARS-CoV-2 M^pro^ (IC_50_ = 16 nM) and an excellent antiviral activity (EC_50_ = 6 nM in Huh7^ACE2^ infected cell with SARS-CoV-2). The improvement in potency and antiviral activity of **24** can be explained by the presence of two additional H-bonds provided by the trifluoromethyl substituent at P3 with the Asn142 residue found in the S3/S4 pocket, as observed in the X-ray structure [43,44].

Another interesting approach was undertaken to enhance the interaction inside the S2 pocket. It involved the introduction of a bicyclic proline derivative, a motif present in the structure of two potent inhibitors of HCV protease, i.e., **Boceprevir** (Figure 9) and **Telaprevir** (Figure 16). In particular, the proline residue was fused with three- (**Boceprevir**) or five-membered (**Telaprevir**) carbocyclic rings to form constrained bicyclic synthons [45]. The most interesting compounds are depicted in Figure 16. The aldehyde derivative **25** (also **MI-09** in Figure 16) contains the same P2 fragment extrapolated from **Boceprevir** and connected to a *para*-OMe benzyl carbamate functional group, while the aldehyde derivatives **26** and **27** (also **MI-23** and **MI-30**, respectively, in Figure 16) are endowed with the P2 fragment of **Telaprevir** and variously halogenated benzyl carbamate moieties at P3 [45].

Compounds **25**–**27** showed inhibitory activity against SARS-CoV-2 M^pro^ in the nanomolar range. The X-ray analysis of the complex **26**/M^pro^ showed that the bicyclic proline is in its *trans-exo* conformation, providing a deep fit within the S2 pocket (Figure 17) [45].

Another example of the **GC376** derivative emerged from a study based on activity-guide optimization [46]. Compound **28** (Figure 18) is a novel dipeptidyl structure, bearing a constrained bicyclic ring in the P3 position. This compound exhibited an inhibitory activity against SARS-CoV-2 M^pro^ in the micromolar range (IC_50_ = 0.18 μM) and antiviral activity in the sub-micromolar range (EC_50_ = 0.035 μM; VeroE6 infected cells).

In parallel, Xia et al. designed and synthetized compounds **29** and **30** (Figure 19), called **UAWJ9-36-1** and **UAWJ9-36-3**, respectively, by retaining the P2 bicyclic proline synthons found in **Telaprevir** and **Boceprevir** [47]. The insertion of a chemically modified bicyclic proline fragment represents a new interesting strategy in the development of recent anticoronavirus agents [48]. Compound **29** was designed as a hybrid form of **GC376** and **Telaprevir**, whereas **30** was designed as a hybrid form of **GC376** and **Boceprevir**.

Compounds **29** and **30** underwent enzymatic assays against all seven human coronavirus M^pros^, providing a similar inhibition profile against SARS-CoV-2 M^pro^ (and overall for all M^pros^) in comparison with **GC-376** (**29**: IC_50_ = 51 nM with a *k_2_/K_i_* = 85257.5 M^−1^s^−1^; **30**: IC_50_ = 54 nM with a *k_2_/K_i_* = 92770.6 M^−1^s^−1^). The antiviral activity was assessed against two different cell lines (VeroE6 and Caco2-ACE2) infected with SARS-CoV-2, hCoV-OC43, hCoV-229E and hCoV-NL63. Compound **30** showed an improved antiviral activity against SARS-CoV-2 as compared to **GC-376** (EC_50_ = 0.37 μM on VeroE6 and EC_50_ = 1.1 μM on Caco2-ACE2). The X-ray structures of SARS-CoV-2 M^pro^ in complex with both compounds confirmed the capacity of the bicyclic prolines to fit within the S2 pocket (Figure 20) [47].

Other **Boceprevir** analogs with an aldehyde warhead at P1′ came from the work of Alugubelli et al. [49]. Among the 19 synthesized compounds, the most promising derivatives were compounds **31**, **32** and **33** (Figure 21), called **MPI43**, **MPI44** and **MPI46**, respectively, displaying high-potency in vitro and in cellulo assays with IC_50_ values in the range 45–120 nM (in vitro) and EC_50_ values in the range 0.14–0.31 µM (in cellulo) [49]. These compounds were also evaluated for their antiviral efficacy against VeroE6 cells infected with USA-WA1/2020 and beta and delta strains of SARS-CoV-2. Interestingly, these three compounds bear a P4 N-terminal carbamate moiety, which seems to be critical for high cellular and antiviral potency and low cytotoxicity [49].

Another aldehyde-based compound was found by Günther et al. [50] using large-scale X-ray crystallography to screen a library of more than 5000 compounds among approved drugs or drugs in clinical trials. The selected compounds were then tested for their antiviral activity against SARS-CoV-2 in VeroE6 cells. Among them, **Calpeptin** (**34** in Figure 22A) showed the highest antiviral activity, with an EC_50_ value of 72 nM and known enzymatic inhibitory activity (IC_50_ = 10.7 μM) [50,51]. This compound contains an aldehyde warhead at P1′, an *n*-Leu residue at P1, a Leu residue at P2 and a Boc group at P3. The X-ray crystal structure of SARS-CoV-2 M^pro^ in complex with **34** showed the standard thiohemiacetal covalent adduct Cys145/aldehyde warhead (Figure 22B). Compound **34** also inhibits cathepsin L, making it a good anti-COVID-19 drug candidate as a dual inhibitor [50].

The aldehyde-based inhibitors discussed so far are dipeptidyl derivatives. However, there are some examples of notable tripeptidyl derivatives (Figure 23). Compounds **35** and **36**, called **MPI3** and **MPI8**, respectively, are the most active tripeptides designed and synthetized by Yang et al. [52]. Both compounds have the aldehyde warhead at P1′ and a (*S*)-γ-lactam group at P1. Compound **35** is endowed with l-Leu at P2 and a Cbz-protected l-Val at P3, while, in **36**, we have a cyclohexyl ring at P2 and a Cbz-protected *t*-Bu Thr at P3. In the in vitro enzymatic inhibitory activity assay, **35** turned out to be the most active compound, displaying an IC_50_ value of 8.5 nM. Conversely, in the in vitro antiviral test, **35** turned out to be less active than **36** (EC_50_ = 2.5 μM in VeroE6 cells), probably due to the presence of all natural amino acids that can be hydrolyzed by cellular proteases [52]. Such a high potency could be related to the dual inhibition of SARS-Co-2 M^pro^ (IC_50_ = 105 nM) and cathepsin L (IC_50_ = 1.2 nM). This assumption led to consideration of the tripeptidyl proteasome inhibitor **MGI-132** (compound **37** in Figure 23) [53] and **calpain/cathepsin inhibitor I** and **II** (compounds **38** and **39**, respectively, in Figure 23) as potential SARS-CoV-2 M^pro^ inhibitors [43]. In vitro enzymatic inhibition and cell-based antiviral tests brought about interesting results, particularly for the **calpain inhibitor II**. This could represent an interesting starting point for the development of dual inhibitors of calpain/cathepsin and SARS-CoV-2 M^pro^.

Ma et al. developed another series of tripeptidyl derivatives, maintaining the aldehyde warhead at P1′ and the (*S*)-γ-lactam group at P1 and then varying the chemical composition at both P2 and P3 positions [54]. The most interesting compounds were **40** and **41**, also called **MPI16** and **MPI17** (Figure 24). At the P3 position, an *O*-*t*-butyl-Thr residue is present in both inhibitors, while, at P2, a *t*-butyl group and a cyclopropyl ring are in **40** and **41**, respectively. These compounds showed excellent enzymatic inhibitory activity (**40**: IC_50_ = 150 nM; **41**: IC_50_ = 60 nM), high potency in the cell-based assays (**40**: EC_50_ = 56 nM; **41**: EC_50_ = 97 nM) and high antiviral activity against different variants of SARS-CoV-2, suggesting that the *O*-*t*-butyl-Thr residue at P3 can be a key structural element for future design. From the X-ray structure of the SARS-CoV-2 M^pro^ in complex with **40**, it emerged that the *O*-*t*-butyl-Thr residue may furnish supplementary Van der Waals interactions with Pro168 and Glu166 [54].

## 6. Ketones

The ketone group is extensively exploited as an electrophilic warhead in the design of cysteine protease inhibitors, including SARS-CoV-2 M^pro^ inhibitors. Ketones are less reactive than aldehydes; thus, electron-withdrawing groups (especially halogens) in the α position are often needed to increase the electrophilic character of carbonyl carbon and make it more susceptible in order to undergo nucleophilic attack. Among the carbonyl compounds, functionalized derivatives of isatins were also recently shown to possess inhibitory activity against SARS-CoV-2 M^pro^, highlighting the importance of this scaffold in the design of antiviral agents [55,56]. The first example of ketone-based inhibitor was the already cited **PF-00835231** (**2** in Figure 4), an α-hydroxy ketone-based peptidomimetic, and its corresponding phosphate prodrug **PF-07304814** (**3** in Figure 4), developed by Pfizer Inc. against SARS-CoV infection during the 2002 SARS outbreak and reproposed as SARS-CoV-2 M^pro^ inhibitors [27]. Compound **2** showed potent inhibitory activity toward SARS-CoV M^pro^ (*K_i_* = 4 nM in SARS-CoV-1 protease FRET assay) [57], as well as toward SARS-CoV-2 M^pro^ (IC_50_ = 6.9 nM; *K*_i_ = 0.27 nM [28]). Successively, the antiviral activity of **2** was assessed on epithelial VeroE6 cells infected with SARS-CoV-2, showing promising results (EC_50_ = 231 nM). Both **2** and **3** entered in vivo preclinical studies, showing comparable enzymatic inhibitory and antiviral activity. Compound **3** displayed superior solubility and pharmacokinetics properties as compared to **2**. The X-ray structure of the covalent complex SARS-CoV-2 M^pro^/**2** evidenced the key interactions with the target: the carbonyl carbon of the α-hydroxy ketone moiety at P1′ covalently binds the Cys145 thiol group to form a hemithioketal adduct that, together with the α-hydroxy group, establishes additional H-bonds with His41 and Gly143 within the S1 subpocket. The γ-lactam moiety at P1 and the Leu residue at P2 insert into subsites S1 and S2, respectively, while the P3 indole moiety establishes Van der Waals interactions with residues at S3 [28].

Taking into consideration the structures of the already published SARS-CoV-2 inhibitors **PF-00835231** and **GC-376**, Bai et al. explored different α-acyloxymethyl ketones bearing a six-membered lactam moiety at P1 in order to mimic the Gln residue [58]. The most promising compounds turned out to be **42**, **43** and **44** (Figure 25), which bear a 2,6-bis(trifluoro)methylbenzoate, 2,4,6-trimethylnicotinate and 4-trifluoromethyl-2,6-dimethylnicotinate fragment as the *C*-terminal α-acyloxy moiety, respectively.

Compound **42**, which was already investigated by Kratz et al. as an inactivator of cathepsin B [59], showed the best results in terms of enzymatic inhibitory activity against SARS-CoV-2 M^pro^ (IC_50_ = 1.0 nM) and excellent SARS-CoV-2 replication inhibition in vitro (EC_50_ = 0.16 μM) [58]. In regard to the two nicotinic-based inhibitors, **43** exhibited a better inhibition profile (IC_50_ = 19.0 nM; EC_50_ = 0.30 μM). The irreversible mechanism of action of **43** was confirmed by X-ray analysis (Figure 26): the attack of the Cys145 thiol group to the α-carbon of the ketone group and the loss of the 2,4,6-trimethylnicotinate as a leaving group. The six-membered lactam inserts into the S1 pocket, establishing the same H-bonds as for the five-membered lactam of **GC-376**. Similar interactions involving other positions are in accordance with the already discussed **GC-376** [58].

Other remarkable ketone-based SARS-CoV-2 M^pro^ inhibitors are represented by the **PF-00835231** benzothiazolyl analogs **5**, **6** and **7** (Figure 5), already discussed in the section of **Nirmatrelvir** [28]. Many other benzothiazolyl ketones with anti-SARS-CoV activity have been repurposed as SARS-CoV-2 M^pro^ inhibitors. One example is represented by the compound **45** (Figure 27) also called **YH-53** [60,61]. Compound **45** bears a P1′ benzothiazolyl ketone, a P1 *γ*-lactame moiety, a P2 Leu residue and a P3 indole substituent. The SARS-CoV-2 M^pro^ assays reported for this compound an IC_50_ of 0.13 μM and an antiviral activity of 2.6 μM assessed in VeroE6 cells [60]. Starting from this lead compound, Higashi-Kuwata et al. considered the introduction of fluorine atoms to improve pharmacokinetics properties due to the higher lipophilicity of the C-F bond compared to the C-H bond [62]. Among several modifications, they explored 4-fluorinated benzothiazole ketones, obtaining compounds **46** and **47** (Figure 27), named **TKB245** and **TKB248**, respectively, as the most promising M^pro^ inhibitors. By replacing the 4-methoxy-indole ring at P3 with the trifluoroacetyl l-α-*tert*-butyl Gly, and the Leu residue at P2 with the 6,6-dimethyl-3-azabicyclohexane fragment, they achieved **46**, which exhibited an improved enzymatic inhibitory and antiviral activity with respect to **45** (IC_50_ = 7 nM μM; EC_50_ = 0.03 μM). To reduce the hydrolysis rate of this compound and improve its pharmacokinetic properties, the carbonyl group at P2 was replaced with a thiocarbonyl group, affording compound **47** (named **TKB248**). Pharmacokinetic studies evidenced higher T_1/2_ for **47** as compared to **46** (4.34 h vs. 3.82 h), although the enzymatic inhibitory and antiviral activity were lower (IC_50_ = 74 nM; EC_50_ = 0.22 μM) [62].

The in vivo evaluation of efficacy and pharmacokinetic parameters of both compounds were performed on human ACE2-knocked-in mice exposed to SARS-CoV-2, showing promising results. From the mass spectrometric analysis, it was found that both compounds promote dimerization of M^pro^, which is bound to these inhibitors, preventing the entrance of the natural substrate. To elucidate the inhibition mechanism, the X-ray structures of **46** and **47** in complex with SAR-CoV-2 M^pro^ were obtained, showing that both compounds have an identical binding mode with the active site [62]. As an example, Figure 28 shows the co-crystal structure of **46** with SAR-CoV-2 M^pro^: the carbonyl carbon at P1′ forms a covalent thiohemiacetal adduct with Cys145 and the thiohemiacetal-OH establishes H-bonds with Gly143 and Ser144, while the 4-fluorobenzothiazole ring fills the S1′ subsite with the 4-fluorine atom that points out of the subpocket. At the P1 position, the γ-lactame ring establishes H-bonds with His-163, Glu166 and Phe140, while the centered amide group forms another H-bond with His164. The dimethyl-bicyclo[3.1.0]-proline moiety fits into the S2 pocket, while the trifluoromethylacetamide group at P3 forms halogen interactions with residues of the S3 pocket [62].

Yang et al. reported novel benzothiazolyl-based peptidomimetics acting as SARS-CoV-2 M^pro^ inhibitors, which were designed starting from **Nirmatrelvir** [63]. They firstly introduced the [2.2.1]azabicylic ring at the P2 position and the benzothiazolyl warhead at the *C*-terminus; then, they began to explore different substitutions at P3. The most promising derivative was compound **48** (Figure 29), containing a trifluoromethanesulfonamide group at the *N*-terminus and an adamantly group at P3. Compound **48** exhibited an interesting inhibitory activity against SARS-CoV-2 M^pro^ (IC_50_ = 1.65 μM), a good antiviral activity against SARS-CoV-2-infected VeroE6 cells (EC_50_ = 0.18 μM) and low cytotoxicity. Most importantly, the PK properties and target selectivity of compound **48** were superior to those of other derivatives [63].

In the already mentioned work of Kneller et al., besides the development of nitrile-based compounds, the authors designed a novel SARS-CoV-2 M^pro^ inhibitor bearing a benzothiazolyl ketone warhead [35]. Compound **49**, called **BBH-1**, was obtained by replacing the P1′ ketoamide group of **Boceprevir** with a benzothiazolyl ketone warhead, while the hydrophobic P1 group was replaced with a γ-lactam ring found in **GC-376** (Figure 30).

The antiviral activity of this inhibitor was determined in SARS-CoV-2-infected VeroE6 TMPRSS cells in the presence of CP-100356, a P-glycoprotein inhibitor. These cells express significant amounts of P-glycoprotein that acts as an efflux pump capable of efficiently removing these compounds from the cytoplasm. Therefore, the authors performed an antiviral assay both in the presence and absence of the P-glycoprotein inhibitor: an EC_50_ value of 1.54 μM was recorded in the presence of the inhibitor, while the same experiment performed in the absence of the inhibitor displayed an EC_50_ value of 16.1 μM. The X-ray/neutron (XN) crystal structure of SARS-CoV-2 M^pro^ in complex with **49** was obtained (Figure 31) in order to obtain insights on the binding mode of the inhibitor and the protonation state of the **49-**M^pro^ complex. The XN structure revealed the formation of a covalent bond between the Cys145 residue and the ketone warhead with the newly formed hemithioketal group unprotonated and its negatively charged oxygen inserted into the oxyanion hole of the target. The alkoxy anion is hydrated by a water molecule and stabilized by a H-bond with the Cys145 backbone. Water molecules also interact with the aromatic rings of the benzothiazolyl moiety, which, due to its bulkiness, pushes the His41 residue away from its original position, causing the elimination of the catalytic water molecule from the active site and deprotonation of His164. The γ-lactam of the P1 group inserts into the substrate-binding subsite S1 cavity, forming a H-bond between carbonyl oxygen and the His163 sidechain. The remaining amide groups and urea group of **49** are involved in several H-bonds with His164 and Glu166, while the carbonyl oxygen of the urea moiety forms a H-bond with the side chain of Gln189 [35].

Another interesting example of ketone-based SARS-CoV-2 M^pro^ inhibitors is represented by fluoromethyl ketones (FMKs). The electron-withdrawing effect exerted by the fluorine atoms in the α-position of a methylketones enhances the electrophilicity of the carbonyl carbon toward nucleophilic attack of the thiol group of cysteinyl proteases [2]. Based on the number of fluorine atoms in the α-position, it is possible to distinguish mono-fluoromethyl ketones (MFMKs), di-fluoromethyl ketones (DFMKs) and tri-fluoromethyl ketones (TFMKs) [64]. Zhu et al. reported an MFMK as a SARS-CoV-2 M^pro^ inhibitor from a quantitative high-throughput screening of existing compounds [65]. This compound, called **Z-FA-FMK** (**50** in Figure 32A), is also an irreversible inhibitor of caspase-3. Compound **50** showed a potent enzymatic inhibitory activity toward M^pro^ (IC_50_ = 11.4 μM) and high antiviral activity (EC_50_ = 0.13 μM), obtained from SARS-CoV-2-infected VeroE6 cells, exploiting a CPE reduction assay [65]. Docking models of **50** with SARS-CoV-2 M^pro^ showed that the thiol group of Cys145 performs a nucleophilic attack at the α-position of the methylketone group in an S_N_2-like reaction, where the fluorine atom acts as a leaving group (Figure 32B) [2]. However, MFMKs are easily catabolized, forming toxic metabolites such as fluoroacetate, a feature that compromises their therapeutic utility [64].

On the other hand, DFMKs and TFMKs seem to be more prone to therapeutic use as they enhance the electrophilicity of the carbonyl group without manifesting metabolic issues [2]. In fact, TFMKs have already been investigated as SARS-CoV M^pro^ inhibitors, particularly with the development of compound **51**, as reported by Shao et al. (Figure 33) [66]. Normally, they act as reversible covalent inhibitors toward the target but, in their hydrated form, they can also act as transition-state competitive analogues [2]. Citarella et al. synthetized a new dipeptidyl DFMK as a SARS-CoV-2 M^pro^ inhibitor (i.e., compound **52**; Figure 33) [67], based on the corresponding TFMKs SARS-CoV M^pro^ inhibitor reported by Shao et al., using a direct and chemoselective difluoromethyl unit transfer reaction [68,69]. Compound **52** bears a Cbz-Leu-HomoPhe sequence as a peptide framework linked to a *C*-terminal DFMK moiety. Compound **52** exhibited important antiviral activity toward an MRC5 cell monolayer infected with hCoV-229E (EC_50_ = 12.9 μM). In silico studies showed that the two fluorine atoms are able to establish a halogen bond with residues of the S1′ pocket [67].

The ketone warhead was also applied in a non-peptidic compound, particularly in a bispidine-based ketone **53** (Figure 34). In silico studies suggested that the central carbonylic function acts as an electrophilic warhead during the formation of the covalent adduct with Cys145. The enzymatic inhibitory activity of compound **53** toward SARS-CoV-2 M^pro^ showed noteworthy results (IC_50_ = 0.75 μM) [70].

## 7. α-Haloacetamides

The design of new SARS-CoV-2 M^pro^ inhibitors during these years has also been focused on α-haloacetamide-based compounds. The halogen atoms in the α-position act as leaving groups during the cysteinyl nucleophilic attack on C-X, leading to an irreversible inhibition of the enzyme due to the formation of a S-C covalent bond. A first example of of an α-haloacetamide SARS-CoV-2 M^pro^ inhibitor derives from the work of Stille et al. [71]. Starting from the non-covalent SARS-CoV-2 M^pro^ inhibitor **54** (Figure 35A), called **X77**, Stille and co-workers carried out a study of covalent docking by replacing the P1′ imidazole ring with different electrophile warheads. From this study, the most promising compound turned out to be **55** (Figure 35A), which showed an interesting enzymatic inhibitory activity against SARS-CoV-2 M^pro^ (IC_50_ = 0.41 μM and *K_I_* = 16 μM). Structure–activity relationship (SAR) analysis of the entire panel of synthesized compounds revealed that the pyridine ring is fundamental to achieving inhibitory activity and the *tert*-butyl group is beneficial in terms of potency, while the replacement of C-Cl/C-F is detrimental [71]. The co-crystal structure of SARS-CoV-2 M^pro^/**55** is shown in Figure 35B: a covalent bond is formed between the thiol group of Cys145 and α-carbon, while the pyridyne and *tert*-butylphenyl rings fit into the S2 and S4 subpockets, respectively.

The same approach of the previous work was used by Wang et al. for the design of new SARS-CoV-2 α-haloacetamide inhibitors [72]. Starting from the non-covalent inhibitor **56** (Figure 36), called **23R**, they replaced its P1′ furyl ring with a series of di- and tri-haloacetamides. The most promising compounds of this series turned out to be **57**, **58**, **59** and **60**, called **Jun9-62-2R**, **Jun9-90-3R**, **Jun9-90-4R** and **Jun9-88-6R**, respectively. Compounds **57**, **58** and **59** are endowed with an α,α-dichloroacetamide moiety as an electrophilic warhead at P1′, while **60** contains an α,α,α-tribromoacetamide [72]. All compounds exhibited outstanding and selective M^pro^ inhibition (IC_50_ range 0.08–0.46 μM) and good antiviral efficacy on Caco2-hACE2 cells infected with SARS-CoV-2 (EC_50_ range 0.58–2.07 μM).

The co-crystal structure of SARS-CoV-2 M^pro^ with compound **57** is shown in Figure 37 and confirms the ability of this electrophile warhead to establish a covalent bond with the Cys145 residue. Moreover, the *R* configuration of the pyridine ring at the P1 position seems to be fundamental for the fitting inside the S1 subsite.

Based on compound **56**, a new series of α-chloro-fluoroacetamide (CFA) was proposed as covalent SARS-CoV-2 M^pro^ inhibitors by Yamane et al. [73]. This weak electrophilic warhead for the cysteine –SH group was firstly explored in the design of tyrosine kinase inhibitors, demonstrating high selectivity with respect to other warheads. The CFA-S bond is reversable and can be hydrolyzed under neutral conditions to regenerate the cysteine residue [74]. For this reason, off-target reactions with cysteine proteases and related drawbacks are limited. The most effective compound of this series was **61** (Figure 38A), which contains a pyrimidine ring at P1 and a phenyl-pentafluorosulfanyl group at P2 [73]. Furthermore, it contains two stereocenters, and thus four stereoisomers. From the biological evaluation, it emerged that only the isomer with an (*R*,*R*)-configuration exhibits outstanding and selective enzymatic inhibitory activity (IC_50_ = 56 nM). Docking simulations were performed for all stereoisomers of 61 (Figure 38B): the most stable pose of (*R*,*R*)-61 suggests that the fluorine atom of the CFA group is able to establish a H-bond with the backbone -NH of the Gly143 residue in the oxyanion hole at S1′, inducing the activation of the inhibitor toward the Cys145 catalytic residue [73].

Other α-haloacetamide-based SARS-CoV-2 M^pro^ inhibitors were discovered by Xiong et al. [75]. They first performed a virtual screening of a set of commercial non-peptidic compounds from the ChemDiv database, followed by covalent docking. From this study, eight compounds were selected for the enzymatic inhibitory activity tests toward SARS-CoV-2 M^pro^. Among them, three compounds showed remarkable activity, with the piperazin-2-one-based compound **62** (Figure 39), called **Y020-9948**, as the most promising derivative (IC_50_ = 8.5 μM). The discovery of this new non-peptidyl structure active toward M^pro^ led Xiong and co-workers to resolve the X-ray crystal structure of the complex SARS-CoV-2 M^pro^/**62** in order to understand the binding mode and in view of future developments of more potent analogs. From the X-ray analysis, it is possible to observe the formation of a covalent bond between the Cys145–SH group and the methylene group of the electrophilic warhead. Furthermore, the carbonyl carbon of the warhead establishes a H-bond with the Gly143 residue, while the *meta*-chlorophenyl moiety interacts by π–π stacking with His41 and a halogen bond with the sulfur atom of the Met165 residue [75].

The non-covalent inhibitor **63** (Figure 40), called **MCULE-5948770040**, was discovered via high-throughput virtual screening by Clyde et al. [76]. It showed moderate inhibitory activity against M^pro^ (IC_50_ = 4.2 μM). From the X-ray structure of the complex SARS-CoV-2 M^pro^/**63**, it can be observed that the P1-uracil and P2-dichlorobenzene moieties are inserted into the S1 and S2 subsites, respectively, linked through the piperazine ring that lies above the Cys145 catalytic residue, while the S3/S4 subsites remain unoccupied [76]. Starting from this piperazine-based hit compound, Gao et al. designed a panel of parent compounds from which **64** (Figure 40), called **GC-14**, emerged as the most promising derivative [77]. It differs from the reference compound by the presence of a *N*-(thiophen-2-ylmethyl)acetamide group hanging on the piperazine core and a 3-pyridin ring in place of the pyrimidine-2,4(1*H*,3*H*)-dione nucleus at the *C*-terminus. Compound **64** acted as a potent non-covalent inhibitor of SARS-CoV-2 M^pro^ and also showed potent in vitro antiviral activity against SARS-CoV-2 (IC_50_ = 0.4 μM and EC_50_ = 1.1 μM) [76]. On the basis of these encouraging results, Gao et al. decided to introduce different electrophilic warheads on the piperazine ring of **64** in order to develop covalent inhibitors [78]. Thirty novel compounds came out from this work, where the electrophilic warheads were linked to the piperazine ring via amide or sulfonamide bonds [78]. Among them, the most promising inhibitors were the α-chloroacetamide **65** (IC_50_ = 0.18 μM) and the α-bromoacetamide **66** (IC_50_ = 0.31 μM; Figure 40), called **GD-9** and **GD-13**, respectively (Figure 40). The in vitro antiviral activity of these new potent SARS-CoV-2 M^pro^ inhibitors was also evaluated on VeroE6 cells infected with SARS-CoV-2, showing quite good results for compound **65** (EC_50_ = 2.6 μM), but not for the selectivity profile (CC_50_ = 12.5 μM) [78].

The X-ray co-crystalized structure of SARS-CoV-2 M^pro^ with compound **66** confirmed the irreversible covalent binding mode that occurs between the Cys145 thiol group and the methylene carbon of the α-bromoacetamide moiety. The thiophen-2-ylmethyl substituent is surface-exposed and forms hydrophobic contacts with the Gln189 residue, and the halogenated phenyl ring is inserted into the S2 pocket, forming π–π stacking interactions with the His41 residue and a halogen bond involving the *para*-chloro substituent and the Asp187 residue (Figure 41) [78].

Other interesting non-peptidyl SARS-CoV-2 M^pro^ inhibitors containing an α-chloroacetamide warhead are represented by the pyrazoline-based compounds **67** and **68** and the difluorinated amide **69** (Figure 42). Compound **67**, called (***R***)-**EN82**, was developed by Moon et al. and underwent activity-based protein profiling, a preliminary screening test applied on a library of 582 chloroacetamides and acrylamides [79]. From this first screening, four compounds emerged as promising antiviral candidates. Within this selection, **67** turned out to be the most active compound, with an IC_50_ value of 0.53 μM (Figure 42). Afterwards, Moon and co-workers carried out an exhaustive SAR analysis of this class of derivatives by exploring the effect of different substituents at the C-4 pyrazoline-ring. This work led to the discovery of compound **68**, called ***cis*-HW-2-010B**, which showed inhibitory activity against SARS-CoV-2 M^pro^ in the nanomolar range (IC_50_ = 14 nM; Figure 42) [79]. In another study, El Khoury Léa et al. applied advanced in silico techniques for the design and synthesis of the cyclic α,α-difluoro-amide **69** (Figure 42), called **QUB-00006-Int-07**, acting as a covalent SARS-CoV-2M^pro^ inhibitor (IC_50_ = 0.83 μM) [80].

## 8. α-Ketoamides

Another important electrophilic warhead that has been extensively explored in the design of SARS-CoV-2 M^pro^ inhibitors is the α-ketoamide moiety. The α-carbonyl group undergoes nucleophilic attack by the thiol group of the Cys145 catalytic residue, leading to the formation of a reversible covalent C-S bond. Furthermore, the α-ketoamide warhead establishes additional H-bonds with residues surrounding the active site, stabilizing the whole drug–target interaction. Examples of α-ketoamide-based M^pro^ inhibitors came from the outstanding works of Zhang et al. [13,81]. As a first approach, they designed and synthetized a new series of peptidomimetic α-ketoamides acting as broad-spectrum inhibitors of enteroviruses and alpha- and beta-CoVs M^pro^ [81,82]. Among them, the most interesting derivative was **70**, which showed noteworthy enzymatic inhibitory activity against SARS-CoV-2 M^pro^ (IC_50_ = 0.67 μM) and antiviral activity (performed on human Calu-3 lung cells) in the micromolar range (EC_50_ = 4–5 µM). Cooper et al. demonstrated that the synthesis used to afford compound **70** led to two diastereomers differing for a single chiral center in P2 moiety (Figure 43) [83]. The diastereomeric resolution of **70**, followed by the evaluation of the enzymatic inhibitory activity of the single isomers against M^pro^, led to the discovery that the diastereomer (*S*,*S*,*S*)-**70** is more active than the diastereomer (*R*,*S*,*S*)-**70** (IC_50_ = 0.12 μM for (*S*,*S*,*S*)-**70**; IC_50_ > 5 μM for (*R*,*S*,*S*)-**70**). The antiviral activity of (*S*,*S*,*S*)-**70** was then assessed in different cell lines, finding that, in human Calu-3 lung cells infected with SARS-CoV-2, it exhibited an EC_50_ value of 2.4 μM [83].

Another SARS-CoV-2 M^pro^ inhibitor bearing an α-ketoamide warhead was **71** (Figure 44A), a non-peptidic compound reported by Quan et al. [84]. This compound was obtained by multiple optimization rounds driven by in vitro and in vivo assays, involving the P1, P2 and P3 group of a bis-amide bearing an α-ketoamide warhead. Afterwards, different groups linked to the α-ketoamide moiety at the P1′ position were explored. Compound **71** is made up of an α-acetylketoamide at P1′, a pyridyne ring at P1, a dibenzo(*b,d*)furan group at P2 and a 4-fluorophenyl ring at P3. During the optimization steps, the configuration of the stereocenter linking the P1–P2 units was explored, finding that the (*R*)-epimers are always more active than the (*S*)-epimers. However, it was found that the (*R*)-epimers are prone to epimerization in vivo. In order to reduce such R/S interconversion, at the stereocenter of **71** was inserted a deuterium atom. Compound **71** exhibited a potent inhibitory activity against SARS-CoV-2 M^pro^ (IC_50_ = 8.1 nM) and outstanding oral bioavailability (92.9%, 31.9% and 85.7% in mice, rats and dogs, respectively). The in vitro antiviral activity of **71** was assessed on wild-type SARS-CoV-2 and three emerging variants—B.1.1.7 (Alpha), B.1.617.1 (Kappa) and P.3 (Theta)—displaying EC_50_ values of 11.4 nM, 20.3 nM, 34.4 nM and 23.7 nM, respectively. Finally, the in vivo antiviral activity of the orally administered **71** was evaluated against Alpha and Kappa variants in an infected K18-hACE2 transgenic mouse model, showing potent antiviral activity [84]. The X-ray co-crystal structure of SARS-CoV-2 M^pro^ in complex with **71** was also obtained (Figure 44B), showing a covalent bond between terminal ketone moiety and the Cys145 thiol group. The thioemiketal adduct is stabilized by two H-bonds formed by the thioemiketal-OH and the oxygen of the α-keto moiety with the Cys145-amide and Gly143-backbone, respectively. The pyridine ring at P1 fits into the S1 pocket, forming a H-bond with the His163 residue. The dibenzo[*b,d*]furan group at P2 is deeply inserted into the S2 pocket, forming π–π interactions with the His41 residue. Moreover, this group is further stabilized through hydrophobic interactions with the S2 His41 and the 4-fluorophenyl ring at P3. The latter is also stabilized by hydrophobic interactions involving Met49 and Gln189 [84].

An α-ketoamide-based SARS-CoV-2 M^pro^ inhibitor currently on phase III clinical trials is **72** (Figure 45A), called **RAY1216**, [85]. The design of this compound started from several SAR-optimization rounds of **Nirmatrelvir** involving the P1, P2, P3 and P4 moieties and the nitrile warhead at P1′. In particular, the α-ketoamide warhead at P1′ has been functionalized with a cyclopentyl ring. Moreover, the inhibitor bears a γ-lactam moiety at P1, a cyclopentylproline at P2 (previously used in **Telaprevir**), a cyclohexylglycine at P3 and a tri-fluoroacetamide group at P4. Compound **72** exibited a *K_i_* = 8.4 nM in the enzymatic assay (a value comparable with **Nirmatrelvir**) [85]. The antiviral assays were performed on VeroE6 cells infected with different variants of SARS-CoV-2: the obtained EC_50_ was 95 nM for WT SARS-CoV-2, 130 nM for the Alpha variant, 277 nM for the Beta variant, 97 nM for the Delta variant, 86 nM for the Omicron BA.1 variant and 158 nM for the Omicron BA.5 variant. The in vivo antiviral activity of **72** was tested at different doses in a human ACE2 transgenic mouse model, displaying similar efficacy to **Nirmatrelvir**. Pharmacokinetic properties of **72** were evaluated and compared with **Nirmatrelvir** in different animal species (mice, rats and cynomolgus macaques), showing a shorter elimination half-live and faster plasma clearance. The X-ray structure of **72** bound to SARS-CoV-2 M^pro^ showed the following (Figure 45B): (*i*) in general, the structural elements in common with **Nirmatrelvir** maintain the same interactions with the target; (*ii*) the formation of a covalent bond between the Cys145 residue and the α-ketoamide warhead is confirmed; (*iii*) the thioemiketal-OH establishes a H-bond with His41; (*iv*) the α-ketoamide carbonyl oxygen accepts H-bonds from the oxyanion hole residues; (*v*) the cyclopentyl moiety establishes hydrophobic interactions with Leu27; (*vi*) P2 cyclopentylproline fits into the S2 subsite; (*vii*) the P3 appears to have no important interactions with the target [85].

A recent paper from Huang et al. proposed a novel α-ketoamide-based inhibitor with potent activity against SARS-CoV-2 Omicron variants [86]. They started with an in vitro screening of an in-house library containing more than 30,000 compounds. Compound **73** (Figure 46A) was identified as a new hit compound, showing an IC_50_ value of 1.3 μM. From the X-ray analysis of the complex SARS-CoV-2 M^pro^/**73** (Figure 46B), it emerged that the α-ketoamide warhead covalently binds the Cys145 catalytic residue, interacting with several H-bonds with His41, His164 and residues of the oxyanion hole. The P1 benzyl and P1′ thiazole moieties occupy the S1 and S1′ subsites, respectively, while the P2 phenyl does not fit inside the S2 pocket, but it was observed as exposed to the solvent region [60].

The same research group performed a structural optimization of **73**, focusing on three portions: the P1′ thiazole ring and the P1 and P2 phenyl rings. Each position was investigated, keeping the other two fixed. The most active compound turned out to be **74** (Figure 47A), called **SY110** [86]. This compound was selected based on its enzymatic inhibition activity, low cytotoxicity profile and PK parameters evaluated in Sprague–Dawley rats, displaying an IC_50_ value of 14.4  nM and a higher area under the curve and oral bioavailability as compared to other derivatives. Furthermore, **74** exhibited potent antiviral activity in plaque reduction assays against SARS-CoV-2 Omicron BA.1, its sub-lineages B.1.1.7, B.1.351 and BA.2 and against SARS-CoV-1 and MERS-CoV, demonstrating a pan-coronavirus antiviral efficacy (EC_50_ = 1.3, 0.38, 1.2, 2.8, 0.45 and 6.3 μM against SARS-CoV-2 Omicron BA.1, BA.2, B.1.1.7, B.1.351, SARS-CoV-1 and MERS-CoV). In vivo experiments on Omicron-infected K18-hACE2 mouse models demonstrated the ability of **74** to alleviate the virus-induced pathology and, most importantly, to partially overcome **Nirmatrelvir**-resistant M^pro^ mutants in an FRET assay [86]. The X-ray co-crystal structure of SARS-CoV-2 M^pro^ in complex with **74** showed an atypical binding mode (Figure 47B): (*i*) the α-ketoamide warhead was found to be covalently bound to the Cys145 catalytic residue in the (*R*)-configuration, forming several H-bonds with His41, Cys145 and Gly163; (*ii*) the P1′ thiazole moiety occupies the S1′ subsite, establishing a H-bond with His163; (*iii*) the P1 benzyl moiety occupies the P1 pocket, establishing hydrophobic interactions with Thr25 and Leu27; (*iv*) the P2 chiral ether is exposed to the solvent region near to a flexible loop; (*v*) the P3 3,3-difluorocyclohexyl ring establishes hydrophobic interactions with the residues Met49, Met165 and Gln189 [86].

## 9. Michael Acceptors

Michael acceptors (MAs) such as α,β-unsaturated carbonyl, esters, vinyl, sulfonamides and nitriles have been widely used in the development of SARS-CoV-2 M^pro^ inhibitors. These groups exploit the electrophilic unsaturated β-position to form an irreversible covalent adduct with Cys145 through a Michael addition mechanism. The first example of an MA as a SARS-CoV-2 M^pro^ inhibitor is compound **75** (Figure 48A), called **inhibitor N3**, a peptidomimetic derivative previously reported as a proteases inhibitor of SARS-CoV and MERS-CoV [87]. This pseudo-tetrapeptide contains a vinyl group and a benzyl-ester moiety at P1′ (the MA warhead), a γ-lactam ring at P1, a Leu residue at P2, a Val residue at P3 and a *N*-terminal Ala residue (P4) capped with an isoxazol-3-yl group (P5). As expected, **75** showed a time-dependent enzymatic inhibition with *k_obs_* = 11,300 M^−1^s^−1^, while the plaque antiviral assays on SARS-CoV-2-infected VeroE6 cells showed an EC_50_ of 16.8 μM. The X-ray crystal structure of SARS-CoV-2 M^pro^ in complex with **75** (Figure 48B) confirmed the formation of a covalent bond between β-vinyl carbon and the Cys145 –SH group, while the P1′ benzyl group fits into the S1′ subsite, forming van der Waals interactions with Thr24 and Thr25. The P1 γ-lactam ring fits into the S1 subpocket and is H-bonded with His163 and Glu166. The P2 Leu side chain is deeply inserted into the S2 pocket, while the P3 Val side chain is solvent-exposed. The P4 Ala side chain forms hydrophobic interactions with Met165, Leu167, Phe185 and Gln192, while the isoxazole ring at P5 establishes van der Waals interactions with Phe168. In the same study, another compound was taken into consideration for drug repurposing, i.e., **CINANSERIN** (**76** in Figure 48A), a well-characterized hydroxytryptamine receptor antagonist discovered in 1960 (Figure 48A). Compound **76** displayed an IC_50_ value of 125 μM and an EC_50_ value of 20.6 μM in infected VeroE6 cells [87]. Also in this case, the presence of an α-β unsaturated amide was supposed to be reactive toward catalytic Cys145 via Michael addition [25].

Iketani et al. discovered compound **77** (Figure 49A) by an in vitro screening of existing SARS-CoV M^pro^ inhibitors [88]. It is characterized by a pseudo-tripeptide structure containing an acrylic ethyl ester moiety at P1′, a γ-lactam ring at P1, a Leu residue at P2 and a *N*-Boc-*O*-*tert*-butyl-Thr at P3 position. Compound **77** was demonstrated to inhibit SARS-CoV-2 M^pro^, with an IC_50_ value of 151 nM, while kinetic studies showed an inactivation rate (*k_inact_*/*K_i_*) of 4.13 × 10^5^ M^−1^s^−1^. Its ability to inhibit SARS-CoV-2 viral replication was tested in infected VeroE6 cells in CPE assays, showing an EC_50_ value of 2.9 μM. Also for this compound, crystallographic studies confirmed the MA standard covalent binding mode (Figure 49B) [88].

Other examples of MAs as SARS-CoV-2 M^pro^ inhibitors came from a virtual screening campaign of an in-house database of ligands containing different MAs warheads, such as vinyl sulfones, vinyl amides, vinyl esters vinyl ketones, vinyl nitriles and vinyl phosphonates [89]. This strategy allowed for the identification of two compounds, **78** and **79** (Figure 50), containing a vinyl ketone moiety. Enzyme inhibition assays revealed that both compounds were able to moderately inhibit SARS-CoV-2 M^pro^, with an IC_50_ of 47.2 μM and 157.5 μM, respectively. Docking and molecular dynamics studies validated the covalent inhibition at the active site via Michael addition, underlining the importance of the aliphatic residue at P2 and the aromatic ring with EWG groups in the *para* position at P3 [89].

Lead optimization of **78** and **79** was performed by the same authors in a second work, consisting of the introduction of the γ-lactam moiety at P1 (as for most potent SARS-CoV-2 M^pro^ inhibitors) and different moieties at P2 [90]. From the enzymatic assays, compounds **80**, **81** and **82** (Figure 51), called, respectively, **SPR38**, **SPR39** and **SPR41**, turned out to be the most promising derivatives, exhibiting activity in the sub-micromolar range (*K*_i_ = 0.18–0.26 μM). These compounds were also selected for cross-reactivity tests toward human cathepsin L and B, showing activity in the micromolar range, with the exception of **82**, which displayed a *K*_i_ value of 0.25 μM against cathepsin L and was claimed as a dual inhibitor. The detected antiviral activity ranged from 1.5 to 18.5 μM.

Another lead optimization work (starting from a set of probes bearing an α-chloromethyl ketone warhead at P1′) was performed by Mondal et al. and led to the identification of compounds **83** and **84** (Figure 52A) as promising MAs-based SARS-CoV-2 M^pro^ inhibitors [91]. The new MA warhead replaced the α-chloromethyl ketone moiety as the latter is associated with the already mentioned high reactivity and cytotoxicity. Compounds **83** and **84** showed inhibitory activity in the micromolar range and antiviral activity in the nanomolar range (**83**: IC_50_ = 0.9 μM and EC_50_ = 8.2 nM; **84**: IC_50_ = 1.8 μM and EC_50_ = 14.7 nM). From a structural point of view, **84** contains a *N*-terminal Cbz group and a *para*-fluoro-d-Phe residue at P3, while **83** contains a free terminal amine and l- and d-Phe at P2 and P3, respectively. The high antiviral activity (as compared to the enzymatic inhibition activity) of these compounds may be explained by a dual inhibition of M^pro^ and other proteases involved in the SARS-CoV-2 replication cycle. Further investigation confirmed this hypothesis; indeed, they inhibited cathepsin L in the nanomolar range (IC_50_ = 60 nM for **83**, IC_50_ = 145 nM for **84**). The good cytotoxic profile of these compounds led to in vivo investigations: the results showed that **83** and **84** inhibit the viral replication in SARS-CoV-2-infected K18-hACE2 mice when administered intraperitoneally, indicating them as potentially anti-SARS-CoV-2. The crystal structure of the complex M^pro^/**83** (Figure 52B) confirmed the formation of the Michael adduct with Cys145 and the fit of the γ-lactam ring into the S1 pocket [91].

Citarella and co-workers explored the cinnamic ester moiety as an MA warhead for the covalent inhibition of SARS-CoV-2 M^pro^ [92]. In this work, they replaced the epoxyketone warhead in an already existing M^pro^ inhibitor, obtaining a pseudo-dipeptide with a backbone functionalized with different fragments at its *N*-terminus. From the enzymatic inhibition activity test, the most promising compound turned out to be the carbamate derivate **85** (Figure 53), containing a *p*-OMe substituent on the *N*-terminal phenyl ring. The antiviral activity test on representative hCoVs demonstrated that **86** displayed EC_50_ values within the low micromolar range against hCoV-229E replication (α-CoV), while compounds **87** and **88** (Figure 53), both containing an indole moiety at the *N*-terminus, exhibited interesting antiviral activity against hCoV-OC43 replication (β-CoV). Docking studies and mass experiments suggested the formation of a Michael adduct between the β position of the cinnamic ester and Cys145 [92].

Stille et al. (already mentioned in the section of α-haloacetamides) developed a library of covalent inhibitors with different electrophile warheads by replacing the P1′ imidazole ring in compound **54** (Figure 35A and Figure 54A) [71]. Compounds **89** and **90**, bearing a vinyl sulphone moiety as a warhead, were selected as promising SARS-CoV-2 M^pro^ inhibitors as they showed enzymatic inhibitory activity in the sub-micromolar range (IC_50_ = 0.42 μM and *K_i_* = 4.5 μM for **89**; IC_50_ = 0.17 μM and *K_I_* = 2.3 μM for **90**). Compound **89** differs from **90** in having of a cyclohexyl group at the *N*-terminus in place of a 2-(*meta*-chlorophenyl)ethyl group. The binding of **89** was confirmed by X-ray crystallography (Figure 54B) [71].

Zaidman et al. designed MAs as SARS-CoV-2 inhibitors exploiting an automated pipeline, called *Covalentizer*, able to suggest new covalent inhibitors from non-covalent compounds [93]. Starting from the non-covalent SARS-CoV M^pro^ inhibitor **91** (Figure 55), called **ML**-**188**, they obtained a library of SARS-CoV-2 M^pro^ inhibitors by replacing its furan ring at P1′ with an acrylamide warhead. The most active compound was **92** (Figure 55), which, apart from the P1′ position, differs from **91** due to its *N*-terminus (*meta*-fluorophenetylamide moiety). This compound, obtained as a racemic mixture, exhibited M^pro^ inhibition in the micromolar range (IC_50_ = 2.95 μM). After chiral chromatography separation and a test on single isomers, the (*S*)-enantiomer turned out to be by far the most potent derivative ((*S*)-**92**: IC_50_ = 2.86 μM; (*R*)-**92**: IC_50_ = 86.3 μM). The X-ray crystal structure of (*S*)-**92** in complex with SARS-CoV-2 M^pro^ confirmed the covalent binding mode at the acrylamide warhead, with the *p*-*tert*-butylphenyl group fitting into the S2 subsite and the fluorophenyl moiety establishing hydrophobic interactions with Met165 and Gln189 in the S4 cleft [93].

Also, flavonoids were identified by in vitro screening as SARS-CoV-2 M^pro^ inhibitors acting as Mas. **Myrecitin** (**93** in Figure 56) is one of the most important examples of M^pro^ inhibitors belonging to natural products [94]. Its pyrogallic motif in vivo undergoes oxidation to give an *ortho*-quinone function able to exert Michael’s reactivity toward Cys145 [95]. The X-ray crystallography confirmed such a hypothesis (Figure 56), highlighting the presence of a covalent bond between C6′ and the Cys145 –SH group. In vitro evaluations revealed high potency in the enzymatic assay (IC_50_ value 0.2–0.6 μM) and micromolar antiviral activity in the cell-based assay (EC_50_ = 8 μM).

In view of these promising results, several derivates of **Myrecitin** were synthetized (Figure 57) [95]. The introduction of a *para*-methyl group at its C7 led to compounds **94** and **95**, which showed higher potency (IC_50_ = 0.30 μM for compound **94**; IC_50_ = 0.26 μM for compound **95**) and good antiviral activity (EC_50_ = 12.6 μM for **94**; EC_50_ = 11.5 μM for **95**). The addition of a phosphonate group in the same position led to compound **96**, which exhibited good enzymatic inhibitory activity (IC_50_ = 3.1 μM) and the highest antiviral efficacy (EC_50_ = 3.2 μM) among these series of derivates [95].

## 10. Esters and Thioesters

Compounds bearing an ester moiety as an electrophilic warhead are another class of M^pro^ inhibitors that has been taken into consideration during the COVID-19 outbreak. The mechanism of action of this type of inhibitors involves a nucleophilic attack by the Cys145 catalytic residue on the carbonyl carbon of the ester moiety and a subsequential expulsion of the alkoxy group, resulting in an irreversible acylation of the target. Interestingly, this type of warhead was mostly applied on non-peptidyl small molecules bearing rigid heterocycles such as a pyridine and/or indole ring [25]. In fact, the first examples of ester-based SARS-CoV-2 M^pro^ inhibitors were indole/indoline-based chloropyridinyl compounds, previously widely reported in literature as therapeutic agents against SARS-CoV. Specifically, Ghosh et al. developed **GRL-0920** and **GRL-0820** (**97** and **98** in Figure 58A) and both compounds exhibited good inhibitory activity against SARS-CoV-2 M^pro^, displaying IC_50_ values of 0.25 μM and 73 nM  for **97** and **98**, respectively [96]. The activity on virus-infected VeroE6 cells was 2.8 µM and 15 μM for **97** and **98**, respectively [97]. The binding at the carbonyl group of the indole ring was demonstrated via ESI-QTOF/MS analysis and via the X-ray crystal structure of SARS-CoV-2 M^pro^ in complex with **97** (Figure 58B) [96]. Another important compound is **GRL-1720** (**99** in Figure 58A), bearing an indoline scaffold [98]. This compound was able to irreversibly inhibit the activity of SARS-CoV-2 M^pro^, with an IC_50_ value of 0.32  μM, *k*_inact_ = 2.53 min^−1^, *K*_i_ = 2.15 μM and *k*_inact_/*K*_i_  = 19,610 M^−1^s^−1^. The antiviral assays on VeroE6 cells exposed to SARS-CoV-2 provided an EC_50_ value of 15 μM for **99** without significant cytotoxicity [98].

On the basis of the structure of **97**, Breidenbach et al. identified other SARS-CoV-2 M^pro^ inhibitors by exploring the different position of the ester moiety at the indole scaffold [99]. Compounds **100** and **101** (Figure 59), bearing the ester moiety at position 7 and 2, respectively, exhibited the highest activity (IC_50_ = 55 nM and 34 nM, respectively).

Interestingly, Iketani et al. identified another chloropyridinyl ester as a potent SARS-CoV-2 M^pro^ inhibitor from a wide in vitro screening of already existing anti-SARS-CoV-agents targeting M^pro^ [88]. This compound, called **MAC-5576** (**102** in Figure 60A), exhibited potent inhibition activity, with an IC_50_ value of 81 nM. The X-ray crystal structure of **102** in complex with SARS-CoV-2 M^pro^ showed a covalent bond formation between Cys145 and acyl thiophene moiety (Figure 60B), although no time-dependent inhibition was observed in the enzymatic assay for this compound. The presence of the thiophene ring induces a conformational rotation of the catalytic His41 side chain, and the pyridine ring rotates to arrange itself parallel to the thiophene nucleus, forming π–π interactions [88].

On the indole scaffold, Pillaiyar et al. investigated the thioester warhead [100]. First, they performed a virtual screening on a library of more than 10,000 kinase inhibitors. Then, on the basis of these results, a wide panel of thioesters (i.e., compounds **103**–**109**; Figure 61A,C) were designed, synthetized and biologically tested. The top two compounds of this set were **103** (which displayed an IC_50_ value of 11 nM, *K_I_* value of 14 nM and *k_inact_/K_I_* value of 58,700 M^−1^s^−1^) and **104** (which displayed an IC_50_ value of 88 nM, *K_I_* value of 33 μM and a *k_inact_/K_I_* value of 27,200 M^−1^s^−1^). All compounds showed promising results in the cell-based antiviral assay; in particular, **108** exhibited an EC_50_ value of 38 nM. The X-ray crystal structure of the complex SARS-CoV-2 M^pro^/**103** showed that the Cys145 residue cleaves the inhibitor by the formation of a thioester adduct, with the indole nucleus forming several hydrophobic interactions with His41, Met165, Asp187, Arg188, Gln189 and Met49 (Figure 61B) [100].

## 11. Selenium/Sulfur-Based Derivatives

The investigation of selenium/sulfur-based derivatives as SARS-CoV-2 M^pro^ inhibitors emerged by the repurposing of **Ebselen** (**110** in Figure 61), a non-specific binder that showed an IC_50_ value of 0.67 μM in the enzymatic assay and an EC_50_ value of 4.67 μM in virus-infected VeroE6 cells [87,101]. Zmudzinski et al. screened a panel of **Ebselen** derivates with mono- or di-substitutions within the phenyl ring [102]. Four compounds (i.e., **111**–**114**; Figure 62) exhibited higher activity than **Ebselen** in the enzymatic assay, highlighting the beneficial impact of such substitutions.

Further SAR analyses on these **Elselen**-like derivatives bearing substitutions at the *N*-phenyl ring were carried out by Huff et al. [103]. All compounds underwent enzymatic evaluation in an FRET-based assay and cell-based test in virus-infected VeroE6 cells. Five of them (i.e., **115**–**119**; Figure 63) exhibited IC_50_ values in the sub-micromolar range, while the most effective antiviral compound was **119**, showing an EC_50_ of 0.8 μM. Docking studies indicated the *meta*-substitution as the most favorable due to additional H-bond interactions with the target.

Amporndanai et al. developed another series of **Ebselen**-like derivatives with different *N*-benzyl or *N*-aryl functionalities [101]. Among them, five derivatives (i.e., **120**–**124**; Figure 64) turned out to be more active than **Ebselen**, both in the enzymatic and cell-based assay. In particular, **124** (or **MR6-31-2**) was ~three-fold more effective than **Ebselen** as a SARS-CoV-2 M^pro^ inhibitor. The X-ray structure of the complex enzyme/**124** confirmed the same promiscuous mechanism of action of **Ebselen** (PDB: 7BAL) [101].

**Ebsulfur** (i.e., 2-phenylbenzisothiazol-3(2*H*)-one **125**; Figure 65) is the sulfur isosteric analogue of **Ebselen**. Like **Ebselen**, **Ebsulfur** has been taken into consideration as a lead compound for the design of SARS-CoV-2 M^pro^ covalent inhibitors [104]. Sun et al. performed an in vitro screening of several analogous compounds, noting that derivates containing the furane substituent displayed higher inhibition against M^pro^ than **Ebselen** and **Ebsulfur**. In fact, compounds **126** and **127** (Figure 65) displayed IC_50_ values of 0.11 μM and 74 nM, respectively. Molecular modeling studies evidenced that the furan ring plays a beneficial role in the binding mode with the target by forming hydrophobic interactions with Met165, Arg188, Asp187 and Met49 [104].

Further **Ebsulfur** analogues have been explored by Chen et al., providing structural key features responsible for the activity [105]. Using a high-throughput screening of diverse compound libraries, they identified the hit compound **128** (IC_50_ = 190 nM; Figure 66), whose optimization involved three recurring moieties, namely the benzoisothiazole core, a linker and a terminal variously substituted phenyl ring (Figure 66). The first series of compounds involved the replacement of the 3,5-dichlorophenyl ring of **128** with other substituted aromatic rings or other moieties [105]. In this first series, derivates **129**–**134** exhibited higher enzymatic inhibition activity than the hit **128** [105].

The SAR analysis evidenced that the electronic effects on this ring are less important than the potential establishment of π-π interaction within the target. Therefore, the simple phenyl ring was chosen later on for further optimizations. However, both modifications of the acetamide linker and benzoisothiazole core ring did not bring about more active compounds. Substitutions on the benzoisothiazole core instead led to interesting results (i.e., compounds **135**–**137**; Figure 67), particularly by switching the fluorine atom from position 6 to position 4 as for **135** (IC_50_ = 116 nM against SARS-CoV-2 M^pro^) [105].

## 12. Non-Covalent Inhibitors

Non-covalent inhibitors do not possess an electrophilic warhead and generally exhibit lower reactivity and better selectivity than covalent inhibitors, thus representing an attractive alternative in the context of the development of anti-coronavirus agents [106]. They establish only secondary interactions with the active site, such as H-bonds, hydrophobic stackings, Van der Waals forces, electrostatic interactions and salt bridges [107]. Some examples of SARS-CoV-2 M^pro^ non-covalent inhibitors have already been introduced throughout the article as they may have acted as lead structures for the development of covalent inhibitors. These include compound **54** (**X77**), **56** (**23R**), **63** (**MCULE-5948770040**) and **64** (**GC-14**). Compound **54** (Figure 68A), previously identified as a SARS-CoV M^pro^ inhibitor [108], was investigated by Stille et al. in complex with SARS-CoV-2 M^pro^ via X-ray analysis [71]. The imidazole-amide moiety interacts via H-bonds with His41 and Gly143, while it is positioned at 3.2 Å from Cys145-SH. The pyridine ring fits into the S2 pocket, forming a H-bond with the His163 side chain, while the *tert*-butylphenyl group is deeply inserted into the S4 subpocket. The carbonyl oxygen of the cyclohexyl-amide moiety accepts a H-bond from the –NH function of Glu166 (Figure 68B). Its enzymatic inhibitory activity is in the micromolar range (IC_50_ = 4.1 μM) [71].

Kitamura et al. developed **56** (compound **23R** in Figure 69A), which displayed an IC_50_ value of 0.2 μM in the enzymatic test and an EC_50_ value of 1.3 μM in the antiviral assay. The X-ray crystal structure of SARS-CoV-2 M^pro^ in complex with **56** is reported in Figure 69B. The furylamide moiety at P1′ forms a bifurcated H-bond with Gly143 and the biphenyl group at P2 fits into the S2 pocket, forming hydrophobic interactions, while the pyridine ring at P1 occupies the S1 pocket and establishes several hydrophobic interactions and a H-bond with the His163 residue. The amide group that links the pyridine ring to the α-methylbenzyl group establishes a H-bond with the main chain of Glu166n, while the phenyl ring of the α-methylbenzyl moiety is partially positioned in both the S2 and S4 pockets, forming π-π interactions with the external phenyl of the biphenyl group [109].

As already mentioned, the non-covalent inhibitor **63** (Figure 70A) was discovered via high-throughput virtual screening by Clyde et al., and it was the starting point for the design of piperazine-based covalent compounds [76]. From the X-ray structure of the complex SARS-CoV-2 M^pro^/**63**, it can be observed that the P1-uracil and P2-dichlorobenzene moieties are inserted into the S1 and S2 subsites, respectively, linked through the piperazine ring, which lies above the catalytic Cys145, while the S3/S4 subsites remain unoccupied (Figure 70B) [76].

Gao et al. performed several modifications of **63** in order to occupy each pocket of the active site of M^pro^, and the most promising compound of this work turned out to be **64** (Figure 71A) [77]. The co-crystal structure of complex M^pro^/**64** (Figure 71B) showed that the nicotinic group at P1 fits into the S1 subsite and establishes a H-bond with the His163 residue. The dichlorophenyl group at P2 inserted into the S2 pocket, forming π-π interactions with the His41 side chain and a halogen bond between the chlorine atom at C-4 and Asp187 backbone. The 2-methylthiophene group at P4 fits into the S4 pocket, forming a H-bond with the Glu166 residue, while the piperazine ring shows the same binding mode as **63**. Additionally, the P4 thiophene, P2 3,4-dichlorophenyl ring and imidazole side chain of His41 are involved in sandwich-like π-π interactions, similarly to **56** [77].

In early 2022, Unoh et al. discovered **S-217622** (compound **140** in Figure 72A), the first oral non-covalent, nonpeptidic SARS-CoV-2 M^pro^ inhibitor clinical candidate [110]. This compound was obtained via the screening of an in-house library subjected to a docking-based virtual screening with predefined filters (a hydrophobic moiety in S2 and two H-bond acceptors in S1) followed by biological screening. The obtained hit compound **138** (Figure 72A) was characterized by good in vitro inhibitory activity (IC_50_ = 8.6 μM) and an optimal in vitro and in vivo PK profile for oral administration. The X-ray crystal structure of SARS-CoV-2 M^pro^ in complex with **138** was resolved to understand important binding interactions to maintain for a further hit optimization [110]. To optimize the interaction at S1′, the 4-difluoromethoxy-2-methylbenzene group was replaced with a 6-chloro-2-methyl-2*H*-indazole scaffold, which maintained a H-bond with Thr26, achieving compound **139** (Figure 72A). This compound displayed an outstanding improvement in the inhibitory activity (IC_50_ = 96 nM) and a good antiviral activity against infected VeroE6 cells (EC_50_ = 12.5 μM), maintaining the favorable PK properties like **138** [110]. Additional modifications entailed the replacement of the methyl-amide function at P1 with a 1-methyl-1*H*-1,2,4-triazole ring, affording the clinical candidate **140** (Figure 72A). This compound displayed excellent enzymatic inhibitory and antiviral activity (IC_50_ = 13 nM; EC_50_ = 0.37 μM on WT SARS-CoV-2) and an optimal PK profile, especially in monkey and dog models. Compound **140** was also demonstrated to inhibit the viral replication of a series of SARS-CoV-2 variants (WT, Alpha, Beta, Gamma, Delta and Omicron, with EC_50_ values from 0.29 to 0.5 μM), as well as different human coronaviruses (SARS-CoV, MERS-CoV, hCoV-229E and hCoV-OC43, with EC_50_ values from 74 nM to 5.5 μM) [110]. These promising results prompted Unoh and co-workers to test **140** in vivo in mice models infected with the SARS-CoV-2 Gamma strain: **140** was administered orally immediately and 12 h after infection, exhibiting rapid efficacy measured after 24 h. These promising results in pre-clinical studies make the SARS-CoV-2 M^pro^ inhibitor **140** a potential clinical candidate for oral administration. The X-ray crystal structure of SARS-CoV-2 M^pro^ complexed with **140** showed that the 1-methyl-1*H*-1,2,4-triazole group occupies the S1 pocket, establishing a H-bond with the His163 backbone. The 2,4,5-trifluorobenzylic group at P2 fits into the S2 pocket, forming face-to-face π-π interactions with a rotated His41 residue, while the 6-chloro-2-methyl-2*H*-indazole scaffold at P1′ maintains the H-bond with Thr26 and establishes hydrophobic interactions with the Met49 residue (Figure 72B) [110].

Another contribution in this field was made by Yang et al., who performed a multiple-conformational-based virtual screening on a library of more than 8000 compounds [111]. This screening was based on nine different conformations of the SARS-CoV-2 M^pro^ substrate-binding site, and the 49 top-ranked compounds were then evaluated for their binding profiles. Six compounds were able to bind M^pro^; therefore, they were selected for in vitro evaluation of their enzymatic inhibitory activity on FRET assays. All of them were able to inhibit M^pro^, with IC_50_ values ranging from 0.69 to 2.05 μM. The two most promising compounds from the enzymatic inhibitory assay, namely compound **141 (**IC_50_ = 0.73 μM), called **Z1244904919**, and compound **142** (IC_50_ = 0.69 μM)**,** called **Z1759961356**, were selected for the evaluation of their antiviral activity on SARS-CoV-2-infected VeroE6 cells in plaque reduction assays, displaying EC_50_ values of 5.0 and 8.5 μM, respectively (Figure 73A). The hypothetical binding modes between these compounds and SARS-CoV-2 M^pro^ were elucidated by MD simulation and binding free energy analysis and are depicted in Figure 73B,C: in **141**, the fluorophenol moiety fits into the S1 pocket, while the piperidine ring interacts with the residues of the S4 pocket and connect with the indoline nucleus, which, in turns, establishes interactions with the residues of the S2 and S3 sites. Compound **142** has a higher binding affinity due to suggested additional interactions with the His164 and Met165 in the S1 subsite and Asn47 in the S2 subsite [111].

Han et al. developed a novel non-covalent SARS-CoV-2 M^pro^ inhibitor starting from the already known benzotriazole-based SARS-CoV inhibitor **ML-300** (**143** in Figure 74) [112]. The optimization of **143** led to compound **144**, called **CCF981** (Figure 74), the most potent compound of the work, with an IC_50_ value of 68 nM against SARS-CoV-2 M^pro^ and EC_50_ values in virus-infected VeroE6 cells of 0.50 and 0.56 μM in cytopathic effect (CPE) inhibition and in a plaque reduction assay, respectively.

Another interesting compound was obtained by Elsegini et al. through a pharmacophore-based virtual screening against M^pro^, PL^pro^ and human furin protease, an enzyme involved in the cleavage of spike protein during viral entry, with the aim of selecting a potential dual inhibitor to maximize the antiviral efficacy [113]. After mapping more than 500,000 compounds, only 16 of them were selected for biological evaluation. Among them, compounds **145** and **146** showed inhibition activity against M^pro^ and human furin protease in the sub-micromolar range, but activity against PL^pro^ in the nanomolar range (Figure 75A). Both compounds exhibited promising antiviral activity when tested in vitro against NRC-03-nhCoV, the SARS-CoV-2 strain isolated in Egypt, and in VeroE6 cells, exhibiting EC_50_ values of 0.77 and 0.11 µM, respectively. Docking studies suggested that both compounds form H-bonds with Asn142 and hydrophobic interactions with Met49 and Ile41; compound **145** forms H-bonds with Thr24, Thr25 and Gln189, while compound **146** forms a H-bond with Glu166 and a π-π interaction with His41 (Figure 75B,C) [113].

Luttens et al. were able to identify non-covalent M^pro^ inhibitors through a virtual screening applied on ultralarge chemical libraries [114]. This strategy brought about the selection of top-ranked compounds for the enzyme inhibition assay. Three of them were able to inhibit SARS-CoV-2 M^pro^. The hydantoin-based compounds **147** and **148** (Figure 76) were co-crystalized with SARS-CoV-2 M^pro^ and the X-ray crystal structures (PDB: 7B2U and PDB: 7AU4) showed that the hydantoin carbonyl groups interact with the Gly143 and Glu166 backbone via H-bonds, while the substituents on the hydantoin core fit into S2 and S1 pockets. Furthermore, compounds **147** and **148** were subjected to hit-to-lead optimization guided by docking predictions and determined crystal structures [114]. This strategy was focused on the maintenance of the hydantoin core and variation in the groups at P1 and P2 positions. From this study, it emerged that the isoquinoline and spyro-cyclic scaffolds were the best chemical frameworks for the P1 and P2 position, respectively. Compound **149** (Figure 76), characterized by an *ortho*-chlorophenyl ring as an aromatic tail at the P2 position, displayed the best inhibitory activity against SARS-CoV-2 M^pro^, with IC_50_ = 77 nM and *K_D_* = 38 nM. This compound was also subjected to CPE-based assays performed in infected Huh7 cells, showing an EC_50_ value of 0.11 μM. Compound **149** was also screened against SARS-CoV and MERS-CoV in CPE-based assays, demonstrating broad-spectrum activity against coronaviruses (SARS-CoV-1 EC_50_ = 0.39 μM in VeroE6 cells, MERS-CoV EC_50_ = 0.20 μM in Huh7 cells) [114].

Another important non-covalent SARS-CoV-2 M^pro^ inhibitor was developed by Rossetti et al. by the examination of hits derived from two in silico screening studies (REAL Space and ZINC compound libraries) on two different structures of M^pro^ [115]. From the top-ranked hits, a total of 486 compounds were selected on the basis of fragment screening, drug-likeness and chemical diversity. Only five of them were effective M^pro^ inhibitors. The top two are characterized by a dihydro-quinolinone core (i.e., **150** and **151**; Figure 77). They underwent a first round of chemical structure similarity searches that led to the discovery of three significantly more potent derivatives, i.e., **152**, **153** and **154** (Figure 77), called **Z228770960**, **Z393665558** and **Z225602086**, respectively, with two of them able to enhance the melting temperature of M^pro^ in a thermal-shift assay. Further optimizations were obtained after a second step of chemical structure similarity searches (i.e., **155**–**158**, called **Z222979552**, **Z228166018**, **Z222977344** and **Z222978028**, respectively; Figure 77) [115].

The co-crystal structure of the most potent derivative **155** with SARS-CoV-2 M^pro^ confirmed the non-covalent mechanism of action (Figure 78): the dihydro-quinolinone core forms several H-bonds with Glu166, His163 and His172, while the carbonyl oxygen of the amide bond accepts two H-bonds from the Cys145 thiol and Glu166 backbone, respectively. The *para*-iodobenzene ring establishes π-π interactions with the His41 residue and additional hydrophobic interaction with Asn142, Met49 and Met165 [115].

Other quinazolin-4-one-based non-covalent inhibitors of SARS-CoV-2 M^pro^ are compounds **160** and **161** (Figure 79A), proposed by Zhang et al. [116], which were developed starting from the lead **Baicalein** (**159** in Figure 79A) by replacing its chromen-4-one ring and varying the substitution pattern in C2 and N3 positions. The most interesting compound (**160**) contains a phenyl group both at the C2 and N3 position and posseses higher activity as compared to **Baicalein** [63]. Furthermore, **160** and **161** showed optimal selectivity against other human proteases and low cytotoxicity [116]. The co-crystal structure of SARS-CoV-2 M^pro^/**161** is reported in Figure 79B: the three hydroxy groups form a H-bond pattern with the backbone of Gly143, Ser144 and Gly145 and with the imidazole ring of the His163 residue via a water molecule, while this aromatic portion forms hydrophobic interactions with Cys145, Asn142 and His41. The carbonyl oxygen of the quinazolin-4-one core forms a H-bond with the Glu166 backbone, while the 3′-methyl-4′-fluorophenyl substituent at the N3 position deeply fits into the S2 pocket. The *sec*-butyl substituent at the C2 position matches a newly formed cavity called an S2c pocket, a result of a ligand-induced conformation change involving the side chains of Met49 and Gln189 [116].

Using a similar rational approach, Citarella et al. designed a novel SARS-CoV-2 M^pro^ inhibitor characterized by a trifluoromethyl diazirine ring [117]. In the context of developing fluorinated functionalities able to target druggable enzymes [118], they replaced the epoxyketone warhead in an already existing M^pro^ inhibitor with a trifluoromethyl diazirine ring to afford compound **162**, called **MPD112** (Figure 80A). This compound showed in vitro inhibition activity against SARS-CoV-2 M^pro^ at a low micromolar level (IC_50_ = 4.1 μM), no cytotoxicity and selectivity toward M^pro^ against PL^pro^. Furthermore, mass experiments with SARS-CoV-2 M^pro^ revealed no covalent interaction between the target and ligand, suggesting a non-covalent mechanism of inhibition. A molecular docking approach was exploited to shed light on the binding mode of **162** within the target active site (Figure 80B): docking results showed that the trifluoromethyl diazirine ring is located in close proximity to the Cys145 thiol group, forming a halogen bond with the side chain of Thr26. The benzyl carbamate moiety establishes several hydrophobic interactions with Phe140 and Glu166 and a H-bond with the oxygen of the Glu166 backbone. The central phenyl group forms hydrophobic interactions with Pro168, Leu167 and Gln189 [117].

In early 2022, Alhadrami et al. reported the indole alkaloid **Neoechinulin A** (**163** in Figure 81A) [119], isolated from the Red Sea fungus *Aspergillus fumigatus*, as a potential SARS-CoV-2 M^pro^ inhibitor through bio-guided screening. This compound showed promising inhibitory activity, with an IC_50_ value of 0.47 μM. The mechanism of inhibition was elucidated using docking and molecular dynamics simulations: **163** is characterized by a diketopiperidine nucleus that forms four hydrogen bonds with the Leu141, Asn142 and Gly143 residues of the S1′ sub-pocket and with the Glu166 residue of the S2 sub-pocket. Important hydrophobic interactions were also observed with the His41 residue (Figure 81B) [119].

Zhang et al. demonstrated that the natural product **Shikonin** (**164** in Figure 82A) is a broad-spectrum anti-coronaviruses agent, with micromolar IC_50_ values against the M^pro^ of SARS-CoV-2, SARS-CoV, MERS-CoV, hCoV-HKU1, hCoV-NL63 and hCoV-229E [120]. The X-ray crystal structure of SARS-CoV-2 M^pro^ in complex with **Shikonin** was determined by Li et al. in a previous work [121]. They found out that the presence of **Shikonin** in the active site cleft of the enzyme causes a conformational change in the catalytic dyad His41-Cys45 and establishes a face-to-face π-π interaction involving the His41 residue of the target and the naphthoquinone ring of the inhibitor. One of the two –OH groups of **Shikonin** establishes H-bonds with the Cys145 –SH group and His164 backbone, while the *iso*-hexenyl group establishes H-bonds with Arg188 and Gln189 residues. Another important difference is the change in position of a flexible loop, including Cys44 to Tyr54, Asp187 to Ala191 and Leu141 to Ser144.

Zhang et al. discovered two 9,10-dihydrophenanthrene derivatives as novel SARS-CoV-2 M^pro^ inhibitors through an in vitro screening of an in-house compound library based on the same scaffold [122]. The first round of optimization pointed out the importance of the *para*-bromo phenyl group at R1 and the pyridyne ring at R3 for the inhibitory activity. Then, these two units were maintained for further optimizations carried out on the pyridine ring. The aliphatic –OH group was taken into account for obtaining hydrolyzable esters. The enzymatic assay revealed that compounds **165** and **166** (Figure 83), called **C1** and **C2**, were the most promising derivatives, with IC_50_ values of 1.5 μM and 1.8 μM, respectively. Kinetic studies provided *K_i_* values of 6.1 μM and 7.6 μM for **165** and **166**, respectively, and suggested a dose-dependent mixed-inhibition mechanism of SARS-CoV-2 M^pro^, which means that these inhibitors can bind at least two sites in the target protein, probably the substrate-binding site and the dimer interface. Indeed, molecular docking simulations confirmed this bimodal binding mechanism for **165** [122].

## 13. Allosteric Inhibitors

**Pelitinib** (compound **167** in Figure 84A) is an irreversible covalent pan-inhibitor of the epidermal growth factor receptors developed as an anticancer agent and repurposed as a SARS-CoV-2 M^pro^ inhibitor. Surprisingly, **167** was not able to covalently bind the Cys145 –SH group. Crystallographic studies instead revealed that this compound fits into the hydrophobic pocket of the dimerization site by means of its halogenated aromatic group (Figure 84B) [50]. Its 3-cianoquinoline scaffold interacts with the Ser301 residue, while the ethyl ether substituent interacts with the Tyr118 and Asn142 residues of the opposite monomer. Compound **167** showed anti-SARS-CoV-2 activity at a low micromolar level [50].

Similarly to **164**, **Ifenoprodil** and **RS-102895** (compounds **168** and **169** in Figure 85) bind the same hydrophobic pocket, showing moderate antiviral activity in VeroE6 cells (EC_50_ = 47 μM and CC_50_ > 100 μM for **168**; EC_50_ = 19.8 μM and CC_50_ = 55 μM for **169**) [50].

In the same study, Günther and co-workers identified compound **170,** called **AT7519**, binding a different region located in a deep grove between the catalytic and dimerization domains (Figure 86A) [50]. The halogenated aromatic ring of **170** establishes Van der Walls interactions with several hydrophobic residues, while the pyrazole ring and the external piperidine ring form important H-bonds with Gln110 and Asp153, respectively (Figure 86B). The H-bonding with Asp153 causes a shift in loop 153–155, allowing for an additional salt bridge between Asp153 and Arg298; this last residue has been found to be crucial for the dimerization process by mutational experiments, explaining the moderate antiviral activity of **170** (EC_50_ = 25.2 μM, CC_50_ > 100 μM) [50].

From an FRET-based enzymatic screening of **Niclosamide** derivatives, Samrat et al. identified compounds **171**–**173** (Figure 87), called **JMX0286**, **JMX0301** and **JMX0941,** as potent inhibitors of SARS-CoV-2 M^pro^ (IC_50_ values 3.9–4.8 μM) [123]. After evaluation of their antiviral activity (EC_50_ = 1.7–2.3 μM) and cytotoxicity (CC_50_ = 10.6–53.1 μM), kinetic experiments highlighted a non-competitive inhibition mechanism, while computational studies suggest that these compounds occupy the same allosteric site of **170** [123].

Due to the similarity between the active sites of SARS-CoV-2 M^pro^, FXa and thrombin, Chaves et al. tested different FXa and thrombin inhibitors against SARS-CoV-2 M^pro^ [124]. Among several anticoagulants, **Apixaban** (**174** in Figure 88) turned out to be the most potent compound, with a K*_i_* value of 9.7 nM. From kinetic experiments, Chaves and collaborators observed a non-competitive mechanism of inhibition due to the fact that that V_max_ decreased in the presence of **174**, but the value of the K_m_ was not affected. Among different hypotheses, the authors observed by molecular docking that **174** could bind the dimerization domain at allosteric site 1. The antiviral activity test performed on SARS-CoV-2 Calu-3-infected cells demonstrated that **174** showed important antiviral properties, with an EC_50_ value of 1.8 μM [124].

## 14. Conclusions and Future Perspectives

The COVID-19 pandemic has proved to be an unexpected and unprecedented health emergency for the entire world population and has highlighted the lack of effective and available antiviral therapies to treat infected people. The SARS-CoV-2 virus has afflicted billions of humans worldwide in different aspects of life and, compared to the previous strains of the genus β-CoV MERS and SARS, it has been shown to possess significantly higher transmissibility, with serious implications for the health of the most fragile patients. The lack of effective emergency treatments and the shortage of life-saving drugs prompted scientists and industrial experts to focus important efforts on the development of effective antiviral treatments. In this context, the main proteases emerged as the most appealing targets in the development of anticoronavirus drugs, and this led to the successful marketing of **Nirmatrelvir**, the first oral drug to treat severe forms of COVID-19. Considering the recurrence of coronavirus infections in human history, the search for therapies directed against future pandemics should not be stopped, and this fact again places M^pro^ as the target of choice due to its low mutability among coronaviruses. Modern drug discovery aims to identify broad-spectrum antiviral compounds capable of acting on a wide range of already known coronaviruses (alpha and beta genera) by acting on the preserved M^pro^ in order to prevent future developments of pandemics. Another interesting approach that was highlighted during the discussion of the review consists in the dual mechanism of action of some inhibitors capable of targeting both viral M^pro^ and host cathepsin L, a promising strategy that has already provided interesting results in the panorama of antiviral inhibitors. Taking a critical look at the future of SARS-CoV-2 M^pro^ inhibitors, we hope that research on this front remains a thriving field for preventing the spread of pandemic outbreaks caused by new coronaviruses in the future.

## Figures and Tables

**Figure 1 biomolecules-13-01339-f001:**
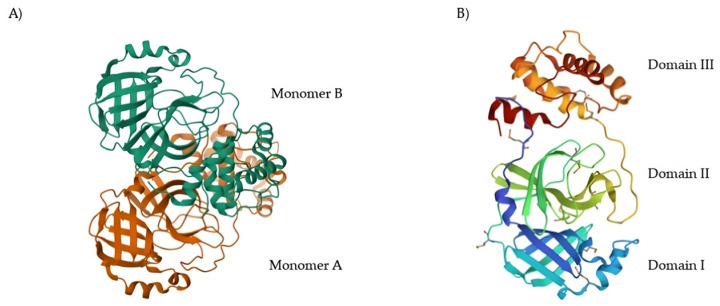
(**A**) Crystal structure of SARS-CoV-2 M^pro^ (PDB: 7ALH) enlightening the two monomers (monomer A is depicted in orange, monomer B in cyan); (**B**) crystal structure of SARS-CoV-2 M^pro^ monomer (PDB: 1P9S) enlightening the three domains (domain I is depicted in blue, domain II in green, domain III in orange).

**Figure 2 biomolecules-13-01339-f002:**
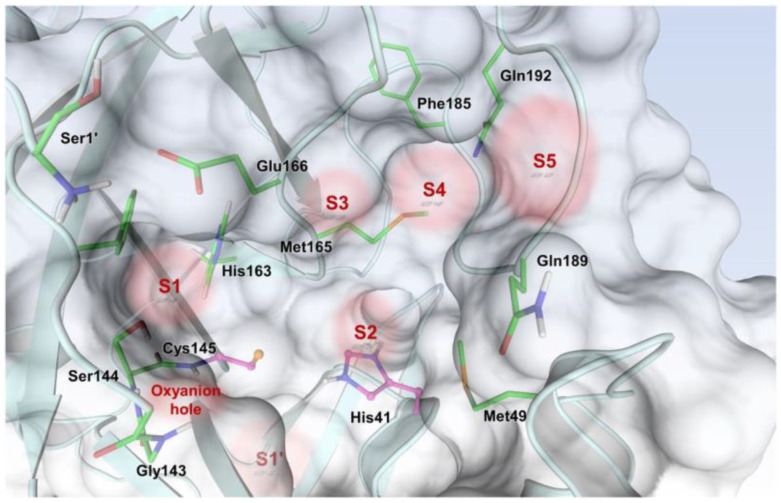
Active site of the SARS-CoV-2 M^pro^ with position of the catalytic dyad, oxyanion hole and subsites S1′–S5.

**Figure 3 biomolecules-13-01339-f003:**
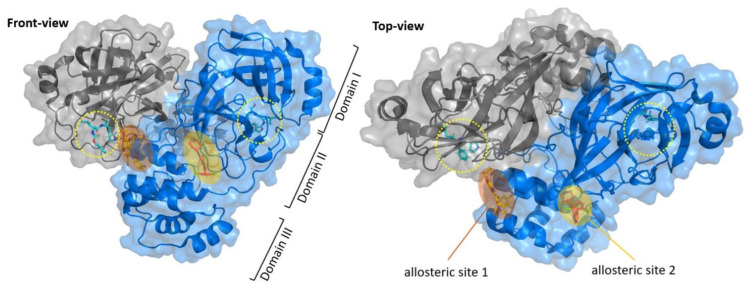
Schematic representation of the allosteric sites of M^pro^. The allosteric sites are located near the dimerization site (allosteric site 1) or between domains II and III (allosteric site 2). The active site in each monomer is indicated with a yellow circle.

**Figure 4 biomolecules-13-01339-f004:**
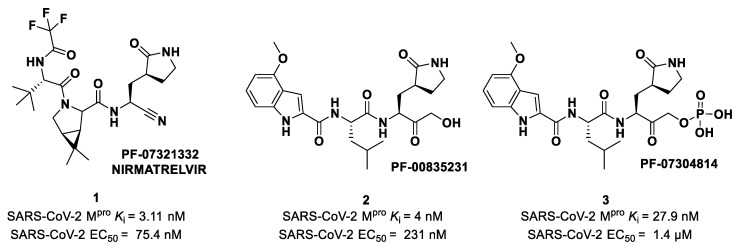
Chemical structure and biological activity of Pfizer inhibitors **Nirmatrelvir**, **2** and **3**.

**Figure 5 biomolecules-13-01339-f005:**
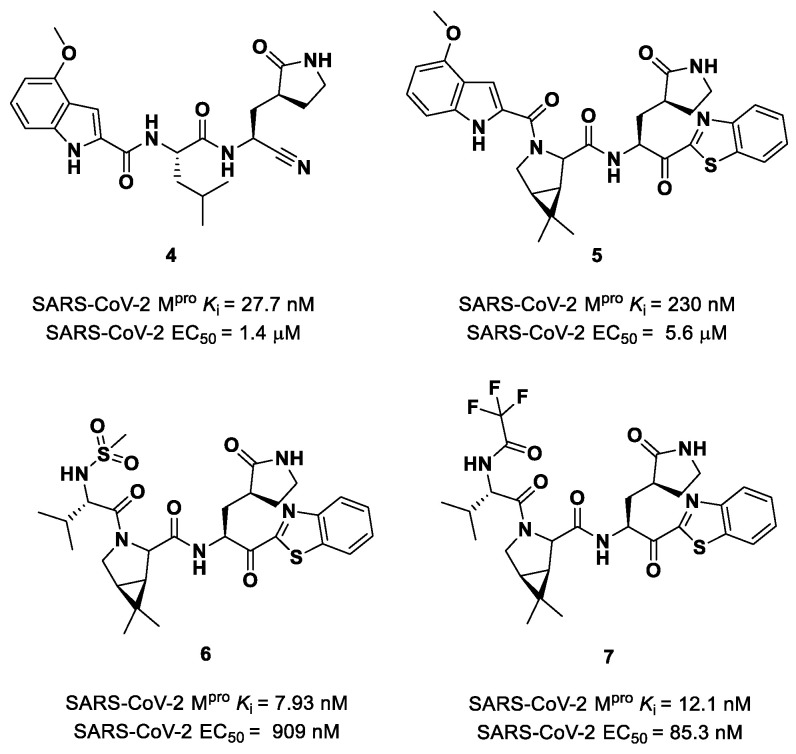
Chemical structure and biological activity of compounds derived from **2**.

**Figure 6 biomolecules-13-01339-f006:**
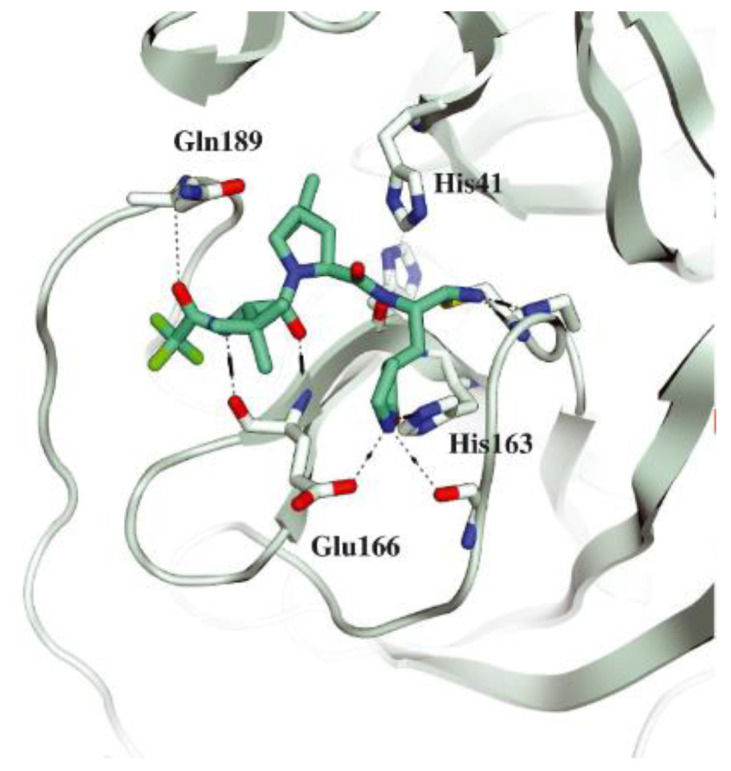
Crystal structure of the complex of SARS-CoV-2 M^pro^/**Nirmatrelvir** (PDB: 7MLF).

**Figure 7 biomolecules-13-01339-f007:**
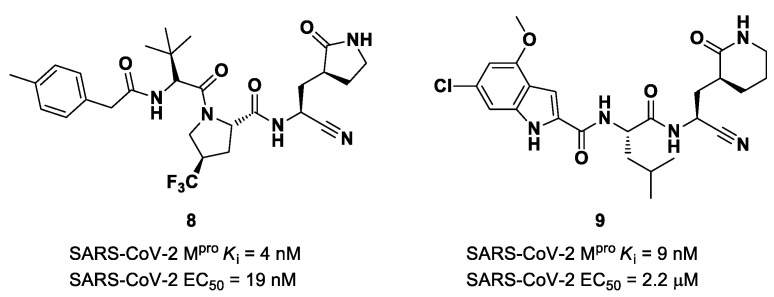
Chemical structure and biological activity of compounds **8** and **9**.

**Figure 8 biomolecules-13-01339-f008:**
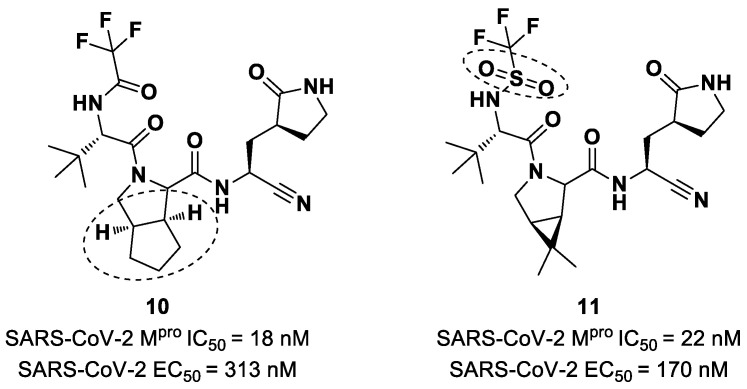
Chemical structure of compounds **10** and **11.** The part marked with dotted circle highlights the modifications with respect to **Nirmatrelvir**.

**Figure 9 biomolecules-13-01339-f009:**
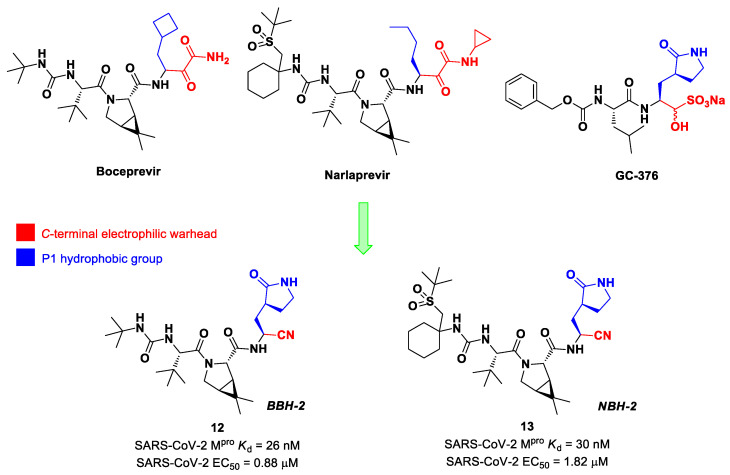
Chemical structure and biological activity of **Boceprevir**, **Narlaprevir**, **GC-376**, **12** and **13**. The colors highlight the chemical features that the authors work on for the design of new compounds.

**Figure 10 biomolecules-13-01339-f010:**
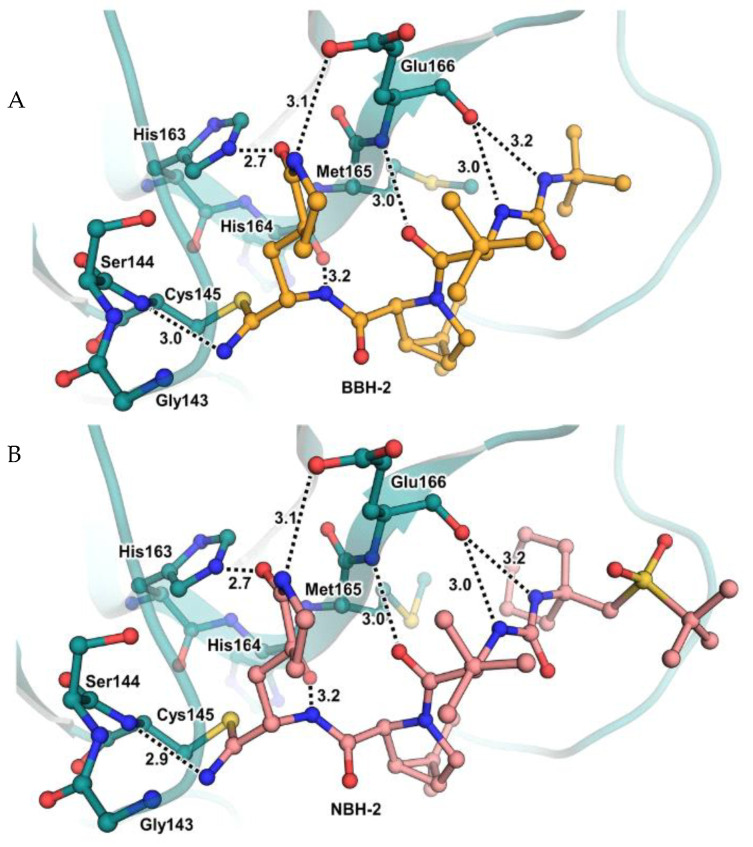
(**A**) X-ray crystal structures of SARS-CoV-2 M^pro^ in complex with **12** (PDB: 7TEH); (**B**) X-ray crystal structures of SARS-CoV-2 M^pro^ in complex with **13** (PDB:7TFR).

**Figure 11 biomolecules-13-01339-f011:**
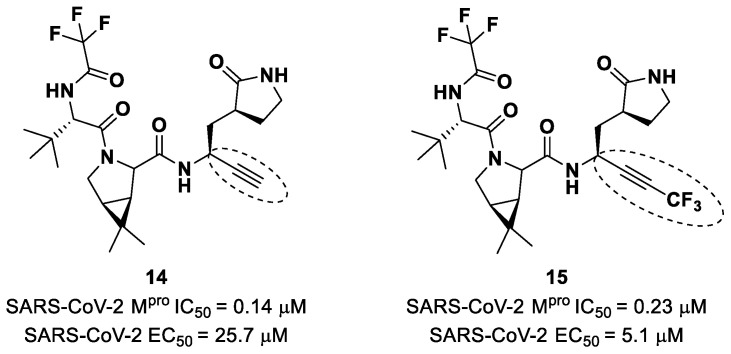
Chemical structure and biological activity of compounds **14** and **15**. The part marked with dotted circle highlights the isoelectronic replacement of the nitrile group.

**Figure 12 biomolecules-13-01339-f012:**
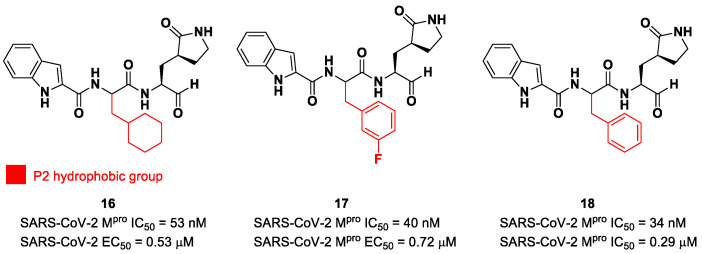
Chemical structure and biological activity of the first aldehyde derivatives (**16**–**18**) as inhibitors of SARS-CoV-2 M^pro^.

**Figure 13 biomolecules-13-01339-f013:**
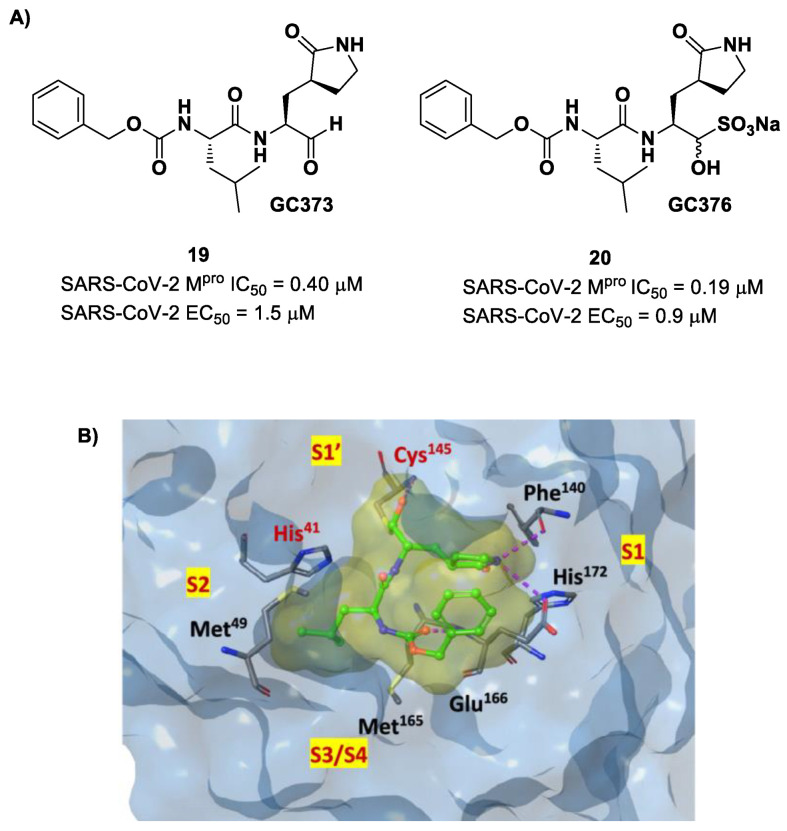
(**A**) Chemical structure and biological activity of compounds **19** and **20**; (**B**) X-ray structure of SARS-CoV-2 M^pro^ in complex with compound **19** (PDB: 6WTJ).

**Figure 14 biomolecules-13-01339-f014:**
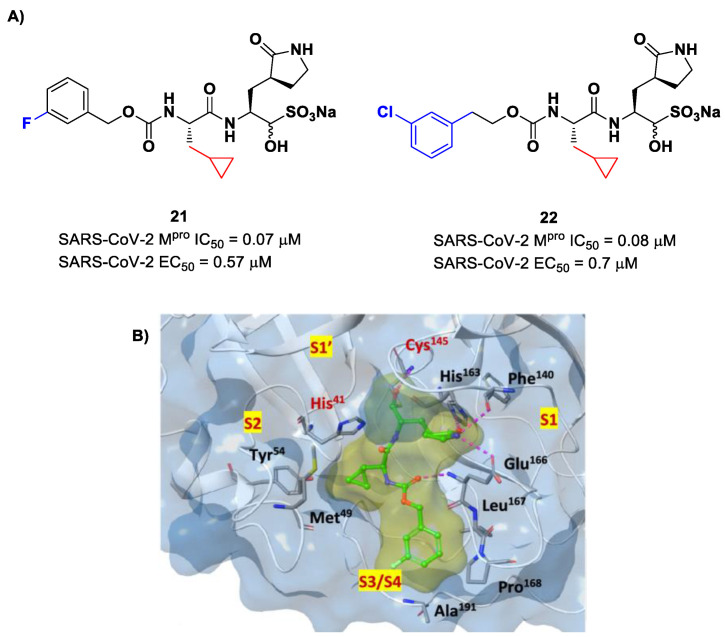
(**A**) Chemical structure and biological activity of compounds **21** and **22**. Those parts marked with colors indicate modifications with respect to the lead structure **20**; (**B**) X-ray structure of SARS-CoV-2 M^pro^ in complex with compound **21** (PDB: 7LCO).

**Figure 15 biomolecules-13-01339-f015:**
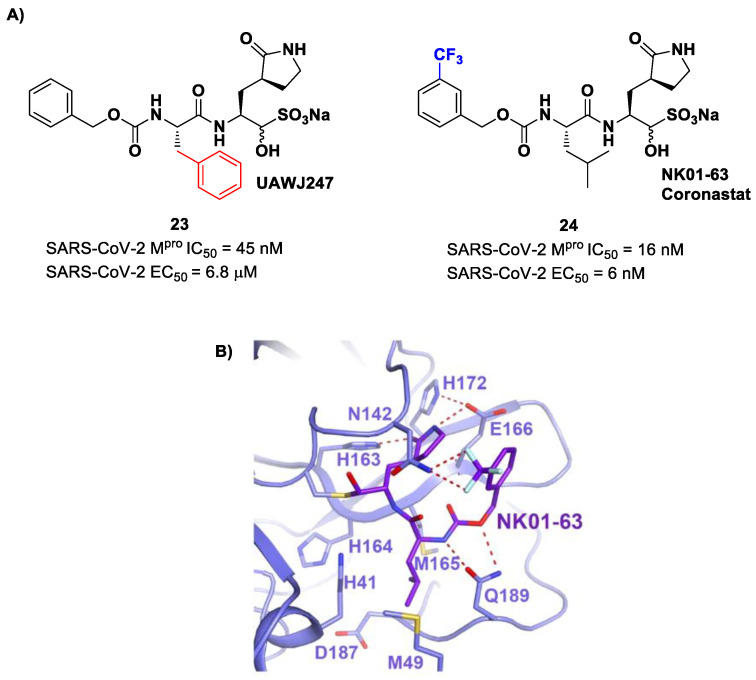
(**A**) Chemical structure and biological activity of compounds **23** and **24**. Those parts marked with colors indicate modifications with respect to the lead structure **20**; (**B**) X-ray structure of SARS-CoV-2 M^pro^ in complex with compound **24** (PDB: 7TIZ).

**Figure 16 biomolecules-13-01339-f016:**
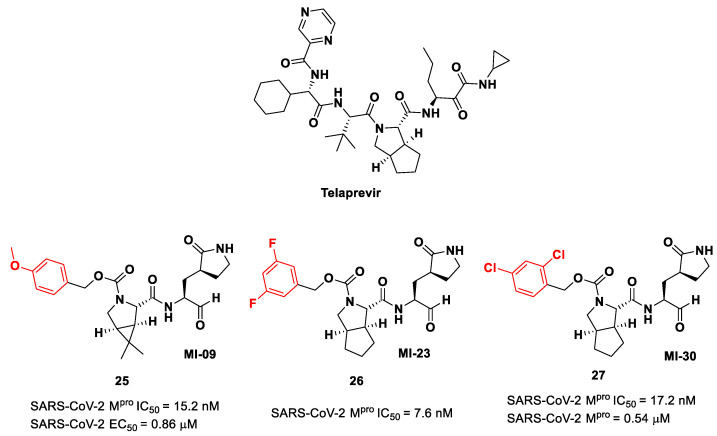
Chemical structure of Telaprevir and chemical structure and biological activity of compounds **25**–**27**.

**Figure 17 biomolecules-13-01339-f017:**
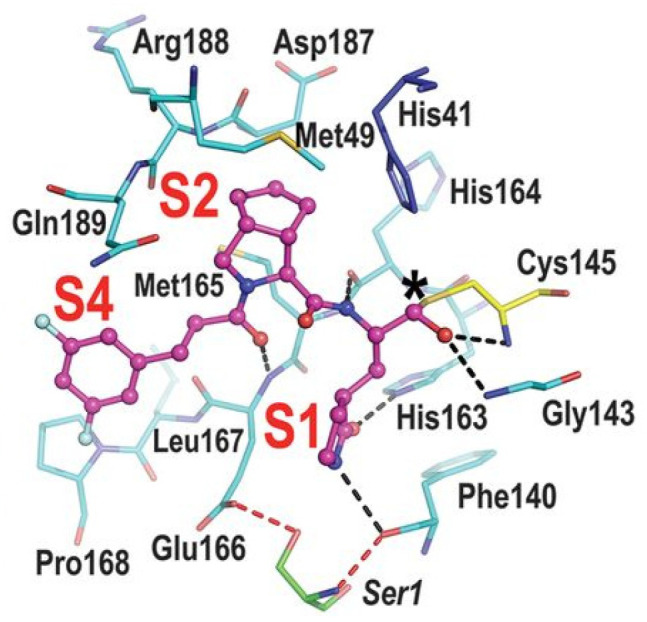
X-ray structure of SARS-CoV-2 M^pro^ in complex with compound **26** (PDB: 7D3I).

**Figure 18 biomolecules-13-01339-f018:**
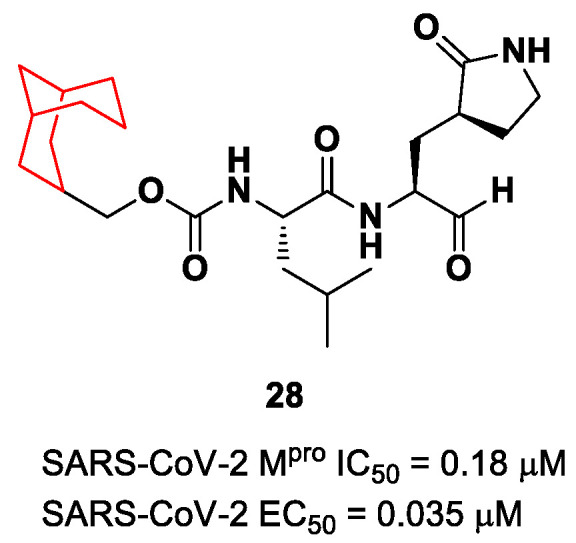
Chemical structure and biological activity of compound **28**.

**Figure 19 biomolecules-13-01339-f019:**
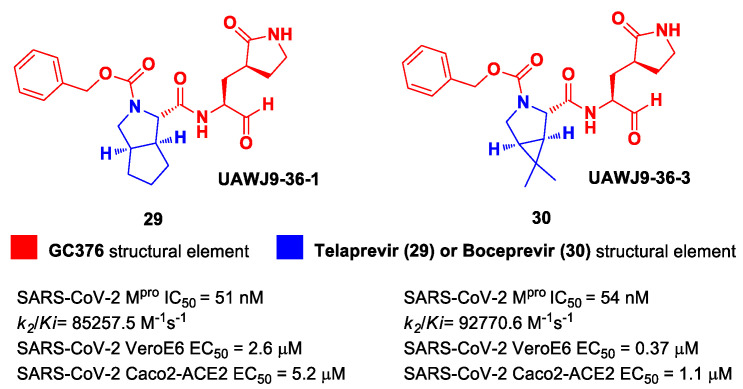
Chemical structure and biological activity of compounds **29** and **30**.

**Figure 20 biomolecules-13-01339-f020:**
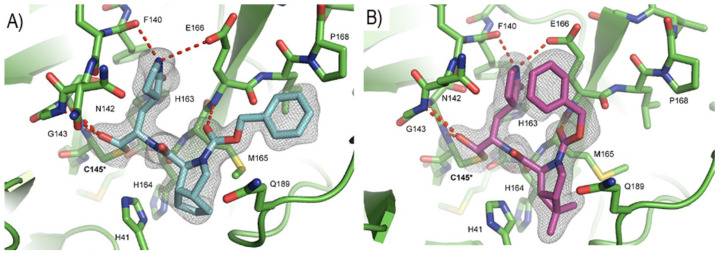
(**A**) X-ray structure of SARS-CoV-2M^pro^ in complex with compound **29** (PDB: 7LIH); (**B**) X-ray structure of SARS-CoV-2M^pro^ in complex with compound **30** (PDB: 7LIY).

**Figure 21 biomolecules-13-01339-f021:**
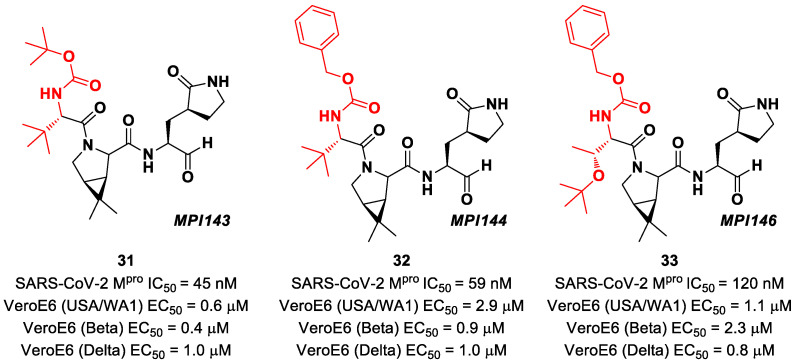
Chemical structure and biological activity of compounds **31**–**33**.

**Figure 22 biomolecules-13-01339-f022:**
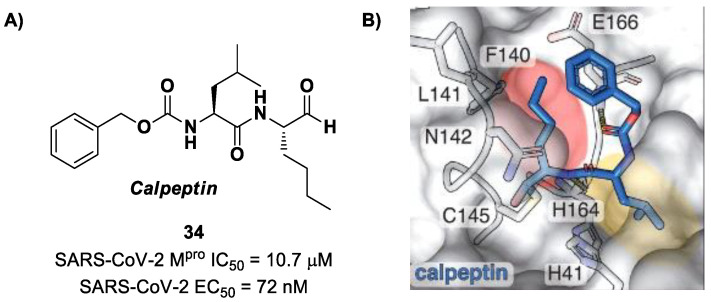
(**A**) Chemical structure and biological activity of compound **34**; (**B**) X-ray structure of SARS-CoV-2 M^pro^ in complex with **34** (PDB: 7AKU).

**Figure 23 biomolecules-13-01339-f023:**
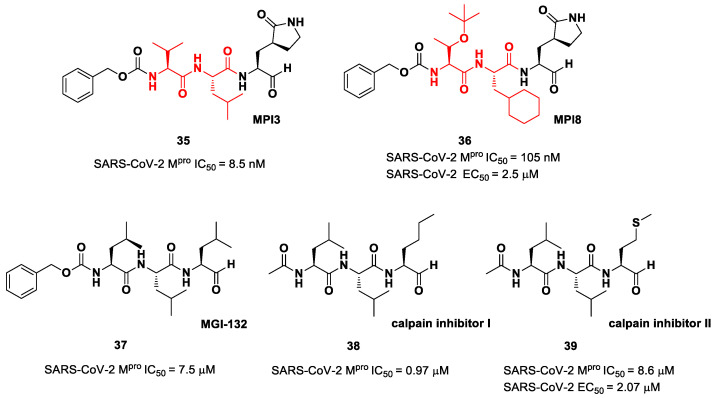
Chemical structure and biological activity of compounds **35**–**39**.

**Figure 24 biomolecules-13-01339-f024:**
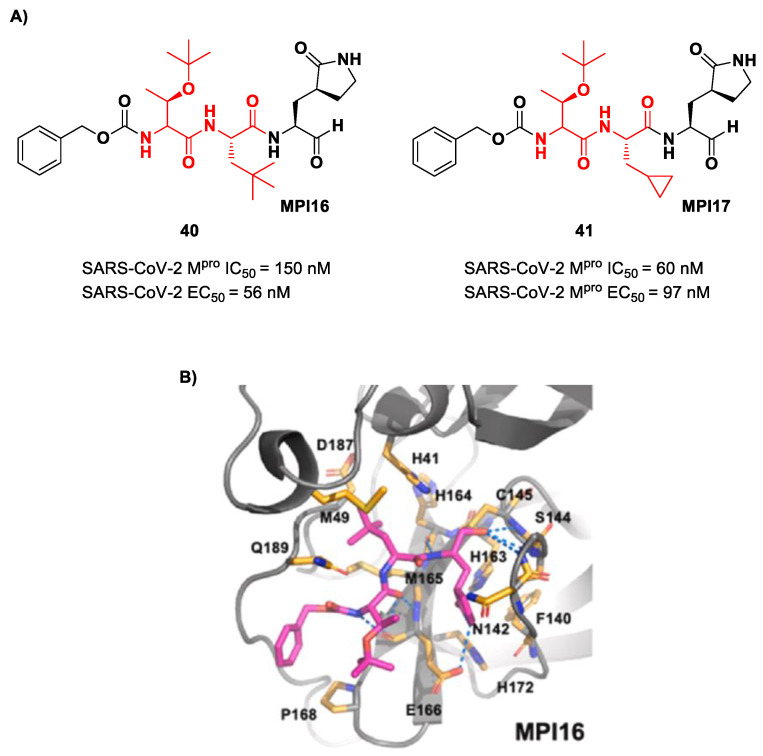
(**A**) Chemical structure and biological activity of compounds **40** and **41**; (**B**) X-ray structures of SARS-CoV M^pro^ in complex with compound **40** (PDB: 7RVQ).

**Figure 25 biomolecules-13-01339-f025:**
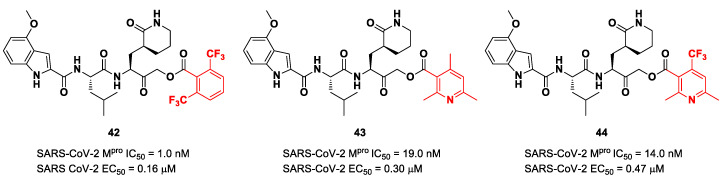
Chemical structure and biological activity of compounds **42**, **43** and **44**.

**Figure 26 biomolecules-13-01339-f026:**
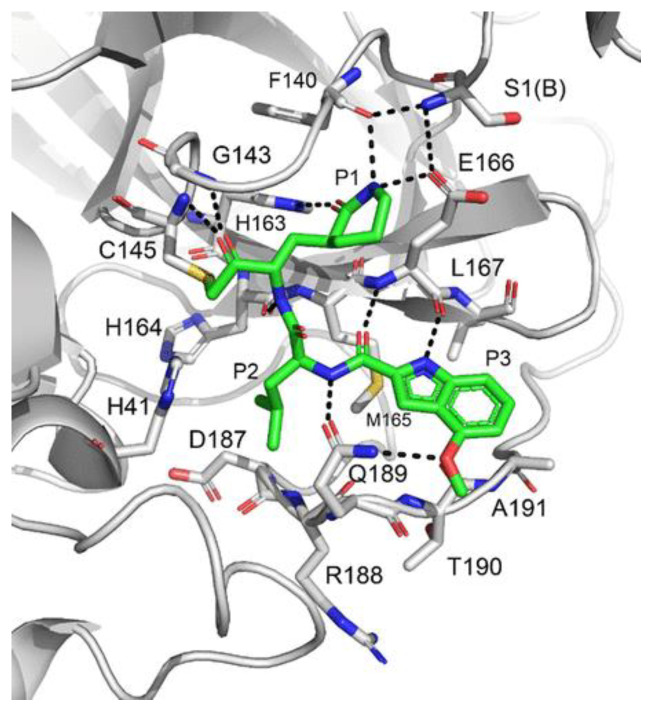
X-ray structure of SARS-CoV-2 M^pro^ in complex with compound **43** (PDB: 7MBI).

**Figure 27 biomolecules-13-01339-f027:**
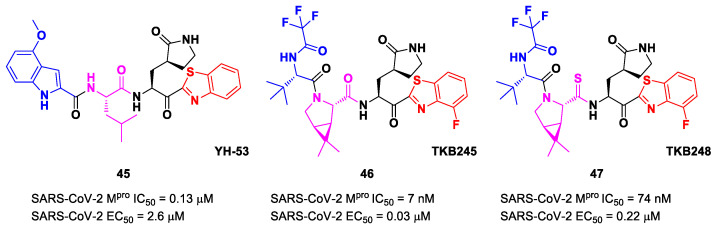
Chemical structure and biological activity of compounds **45**, **46** and **47**.

**Figure 28 biomolecules-13-01339-f028:**
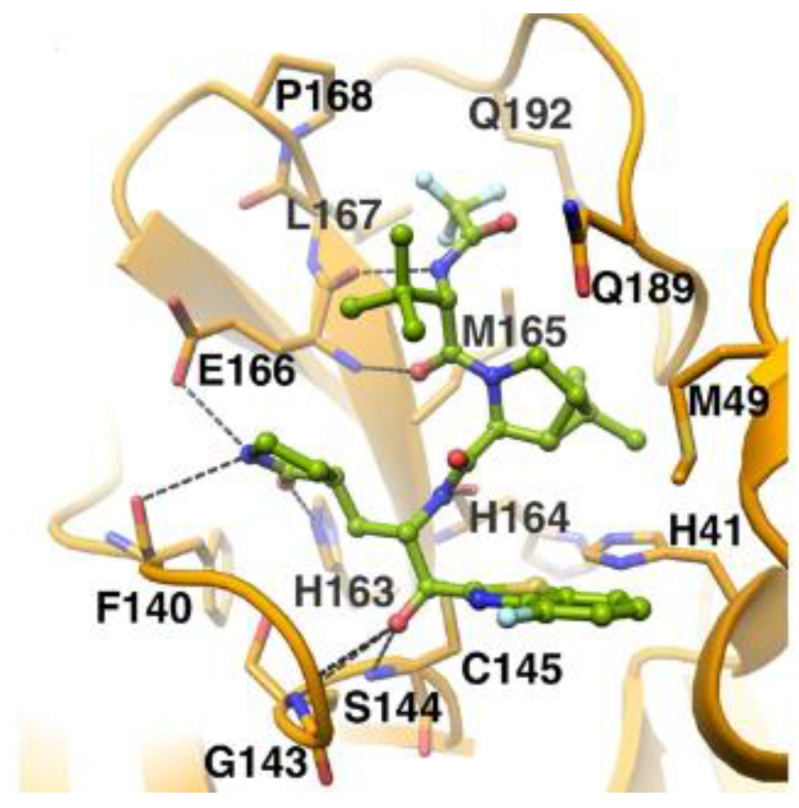
X-ray structure of SARS-CoV-2 M^pro^ in complex with **46** (PBD: 8DOX).

**Figure 29 biomolecules-13-01339-f029:**
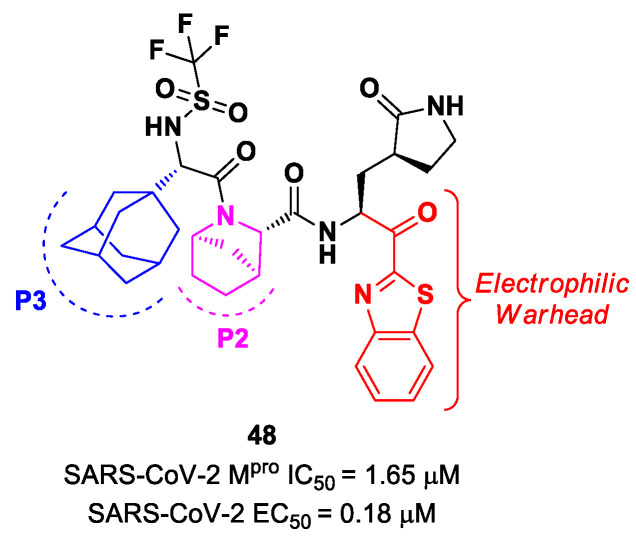
Chemical structure and biological activity of compound **48**.

**Figure 30 biomolecules-13-01339-f030:**
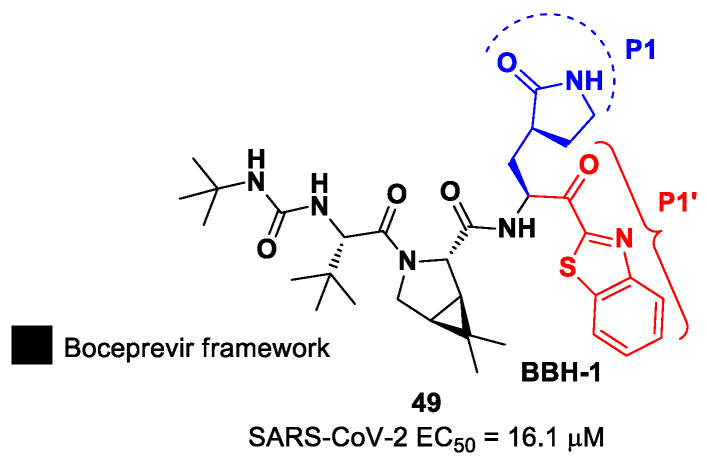
Chemical structure and biological activity of compound **49**.

**Figure 31 biomolecules-13-01339-f031:**
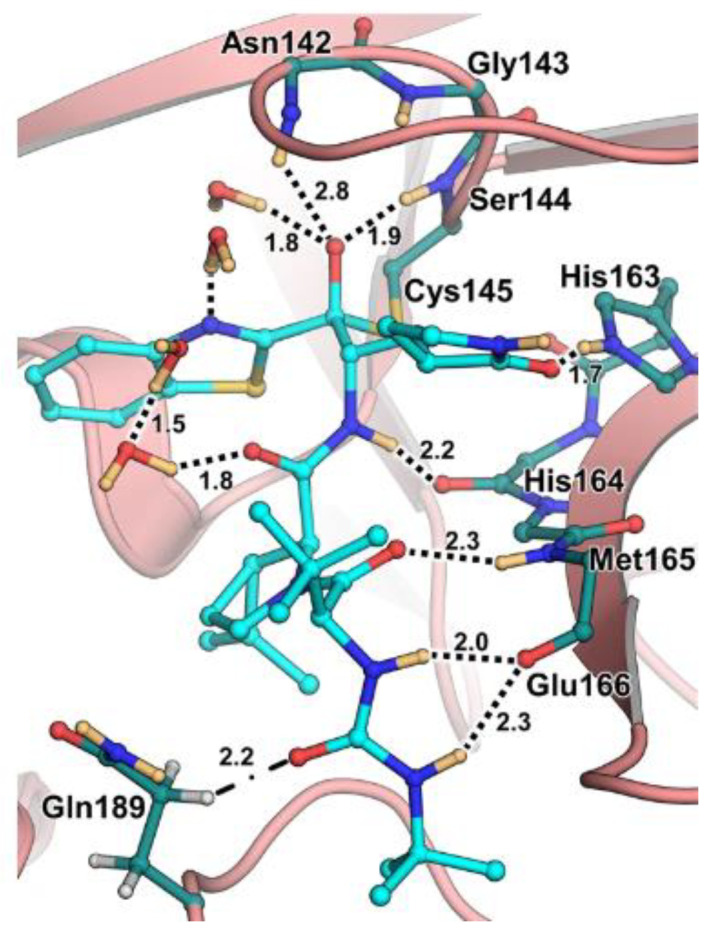
X-ray/neutron (XN) crystal structure of SARS-CoV-2 M^pro^ in complex with **49** (PDB: 7TD1).

**Figure 32 biomolecules-13-01339-f032:**
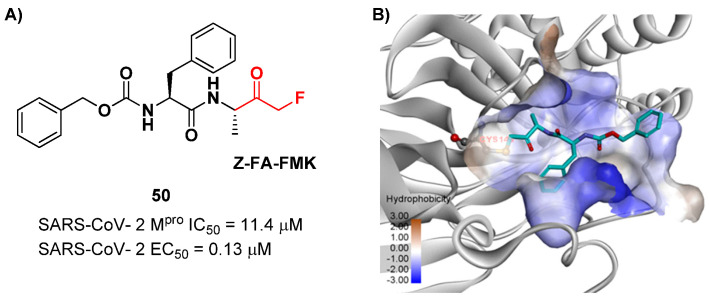
(**A**) Chemical structure and biological activity of compound **50**; (**B**) predicted binding mode of **50** with SARS-CoV-2M^pro^.

**Figure 33 biomolecules-13-01339-f033:**
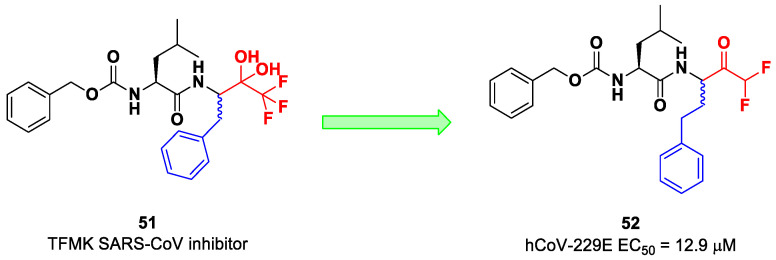
Chemical structure and biological activity of compounds **51** and **52**.

**Figure 34 biomolecules-13-01339-f034:**
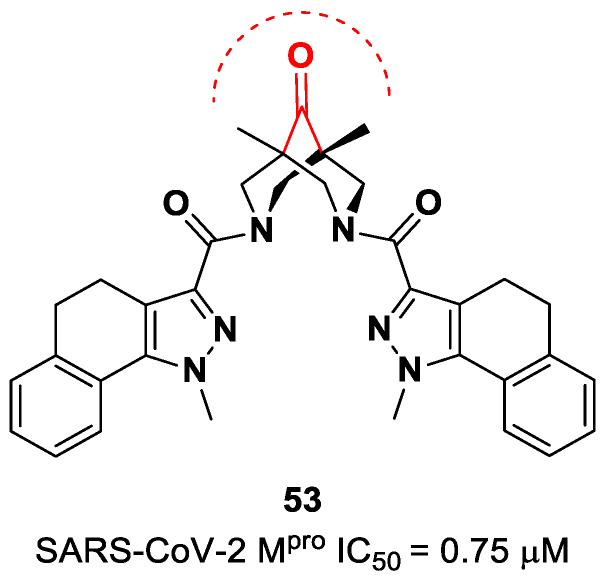
Chemical structure and biological activity of compound **53**.

**Figure 35 biomolecules-13-01339-f035:**
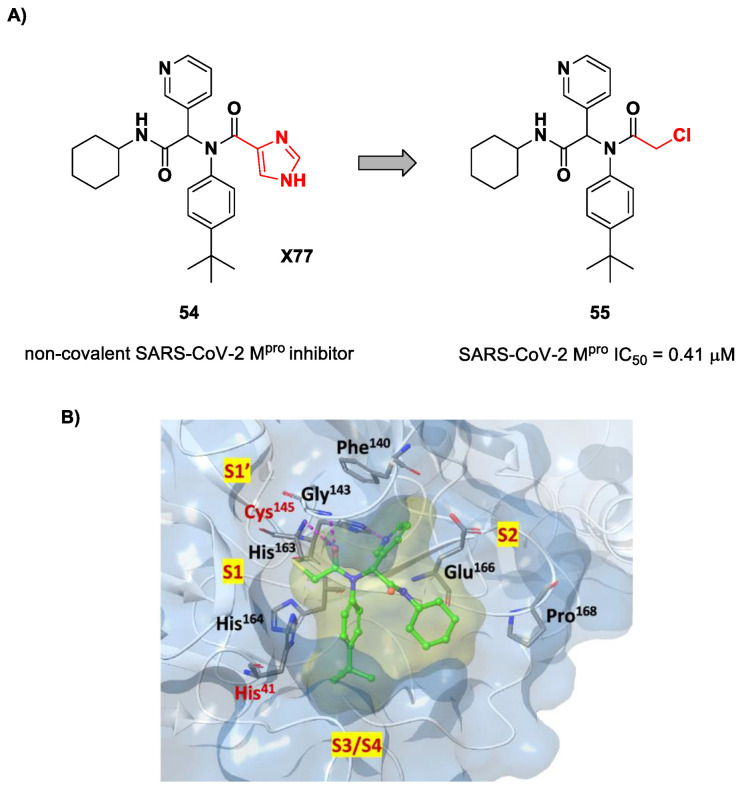
(**A**) Chemical structure and biological activity of compounds **54** and **55**; (**B**) X-ray structure of SARS-CoV-2 M^pro^ in complex with compound **55** (PDB: 7MLF).

**Figure 36 biomolecules-13-01339-f036:**
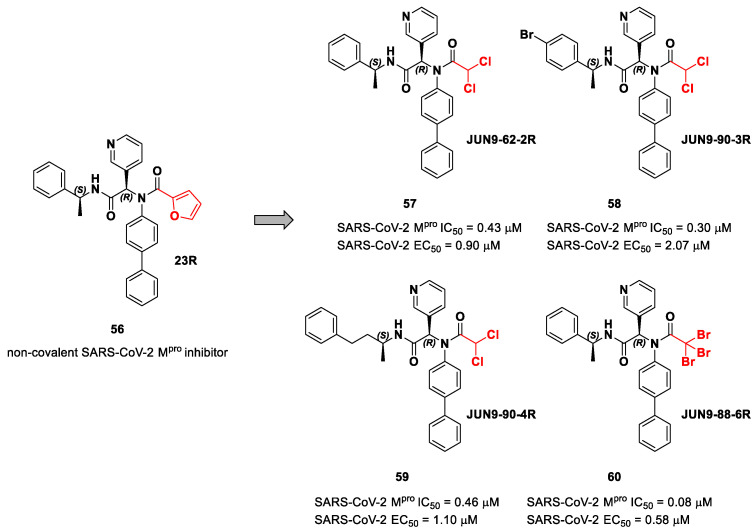
Chemical structure and biological activity of compounds **56**–**60**.

**Figure 37 biomolecules-13-01339-f037:**
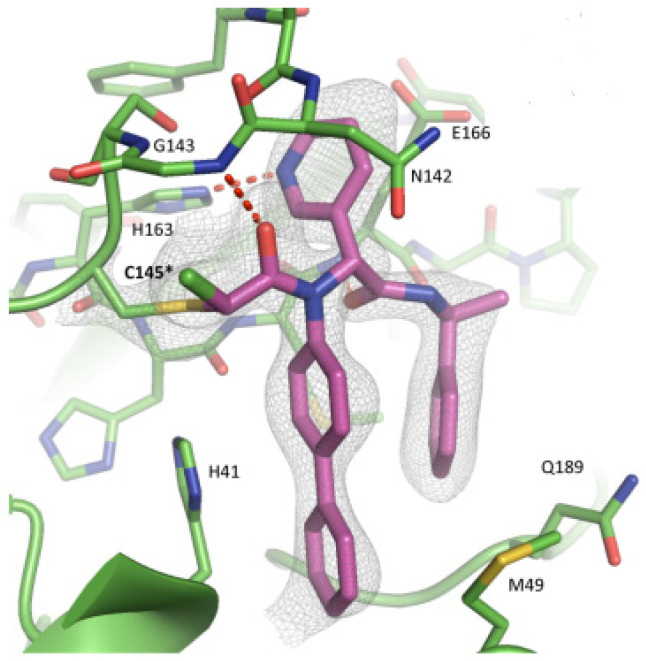
X-ray structure of SARS-CoV-2 M^pro^ in complex with compound **57** (PDB: 7RN1).

**Figure 38 biomolecules-13-01339-f038:**
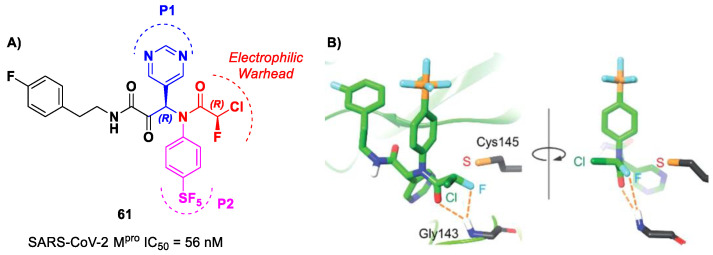
(**A**) Chemical structure and biological activity of compound **61**; (**B**) docking of SARS-CoV-2 M^pro^ in complex with compound **61**.

**Figure 39 biomolecules-13-01339-f039:**
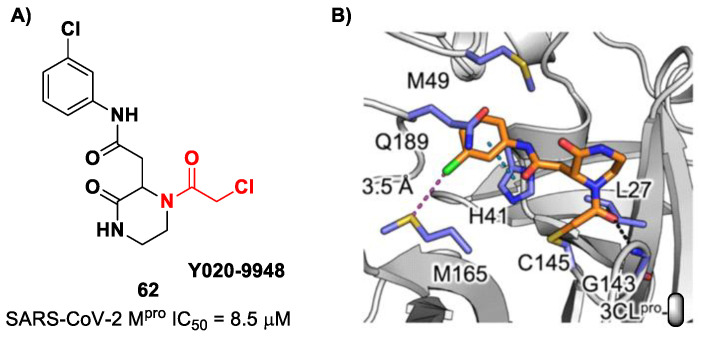
(**A**) Chemical structure and biological activity of compound **62**; (**B**) X-ray structure of SARS-CoV-2 M^pro^ in complex with compound **62** (PBD: 7VVT).

**Figure 40 biomolecules-13-01339-f040:**
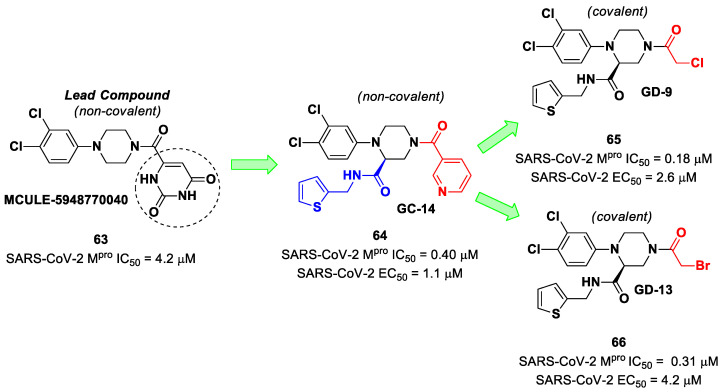
Chemical structure and biological activity of compounds **63**–**66**.

**Figure 41 biomolecules-13-01339-f041:**
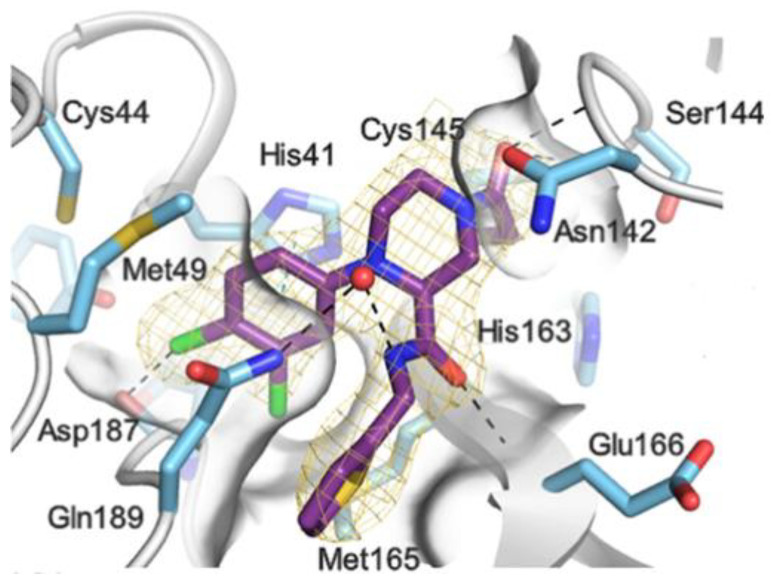
X-ray structure of SARS-CoV-2 M^pro^ in complex with compound **66** (PDB: 8B56).

**Figure 42 biomolecules-13-01339-f042:**
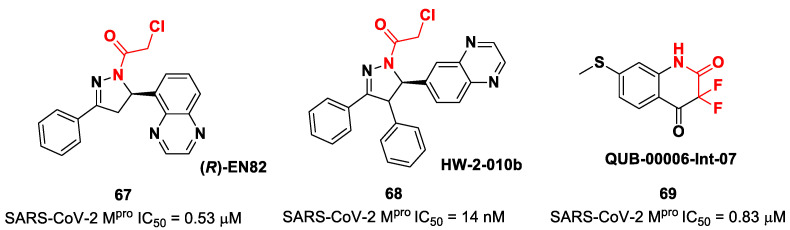
Chemical structure and biological activity of compounds **67**–**69**.

**Figure 43 biomolecules-13-01339-f043:**
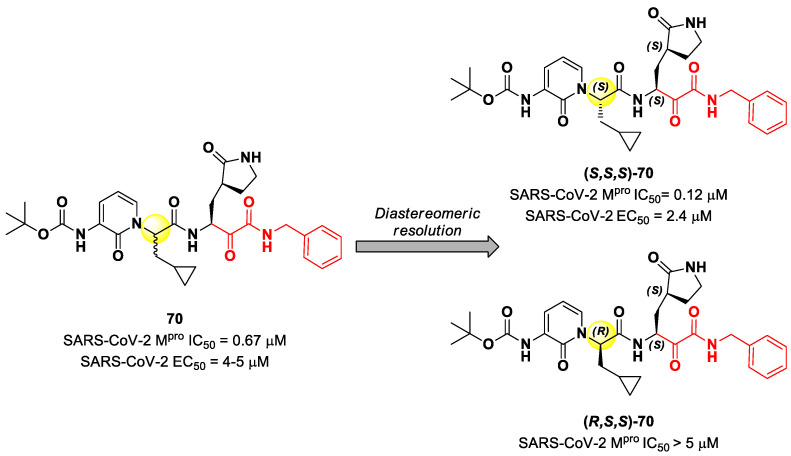
Structure and biological activity of **70** and resolution of the two diastereomers.

**Figure 44 biomolecules-13-01339-f044:**
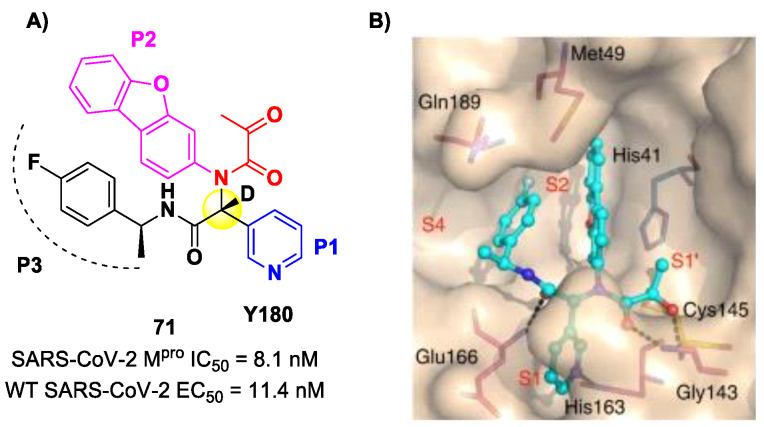
(**A**) Chemical structure and biological activity of compound **71**; (**B**) X-ray structure of SARS-CoV-2 M^pro^ in complex with **71** (PDB: 7FAZ).

**Figure 45 biomolecules-13-01339-f045:**
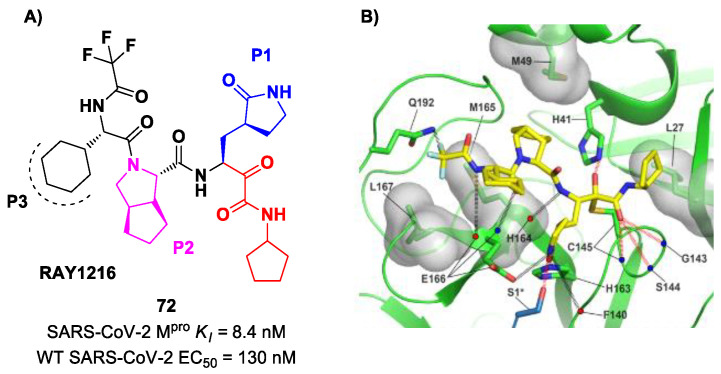
(**A**) Chemical structure and biological activity of compound **72**; (**B**) X-ray structure of SARS-CoV-2 M^pro^ with **72** (PDB: 8IGN).

**Figure 46 biomolecules-13-01339-f046:**
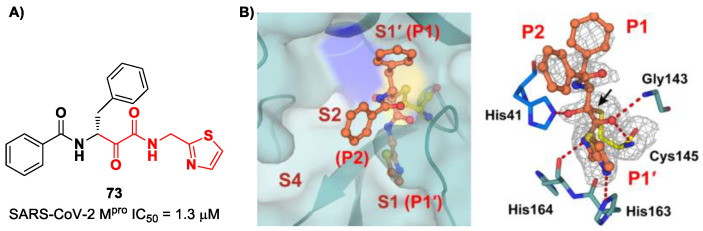
(**A**) Chemical structure and biological activity of compound **73**; (**B**) X-ray crystal structure of SARS-CoV-2 M^pro^ in complex with **73** (PDB: 8HHT).

**Figure 47 biomolecules-13-01339-f047:**
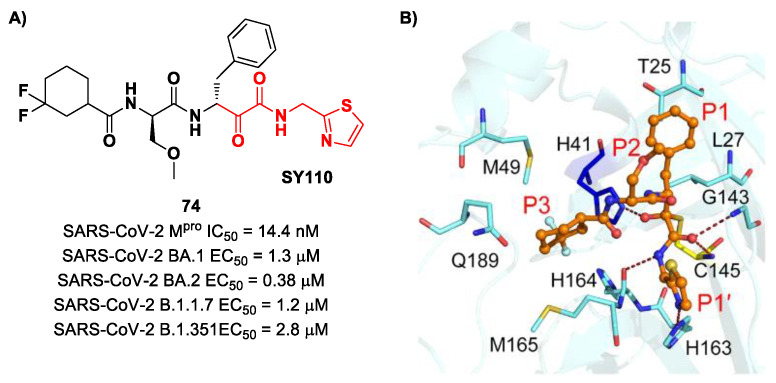
(**A**) Chemical structure and biological activity of compound **74**; (**B**) X-ray crystal structure of SARS-CoV-2 M^pro^ in complex with **74** (PDB: 8HHU).

**Figure 48 biomolecules-13-01339-f048:**
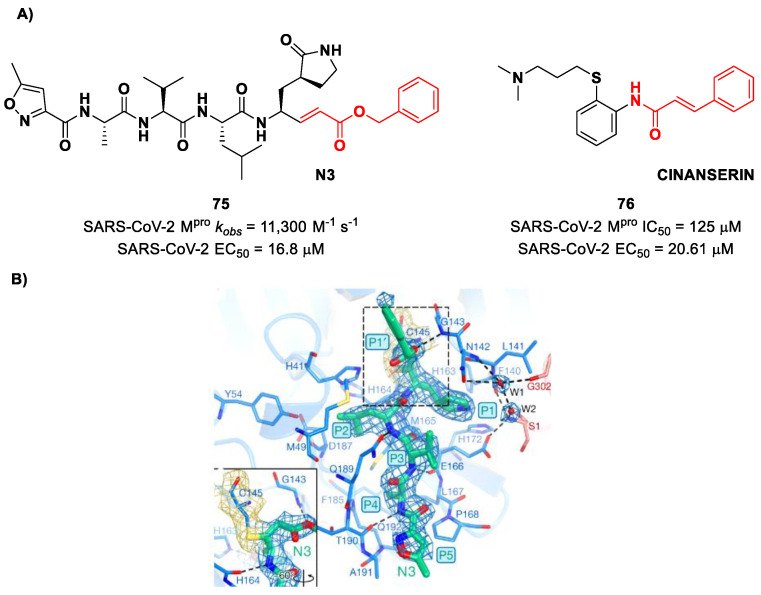
(**A**) Chemical structure and biological activity of compounds **75** and **76**; (**B**) X-ray structure of SARS-CoV-2 M^pro^ in complex with **75** (PDB: 6LU7).

**Figure 49 biomolecules-13-01339-f049:**
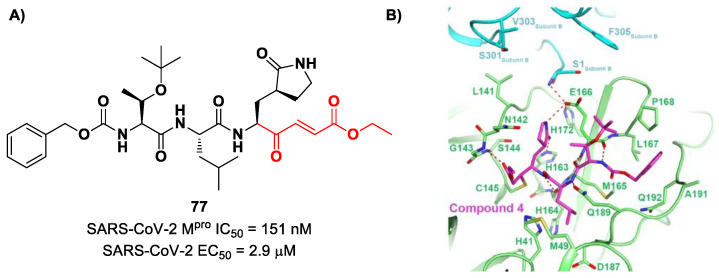
(**A**) Chemical structure and biological activity of compound **77**; (**B**) X-ray structure of SARS-CoV-2 M^pro^ in complex with **77** (PDB: 7JT7).

**Figure 50 biomolecules-13-01339-f050:**
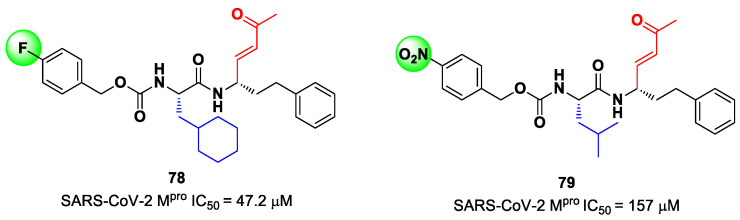
Chemical structure and biological activity of compounds **78** and **79**.

**Figure 51 biomolecules-13-01339-f051:**
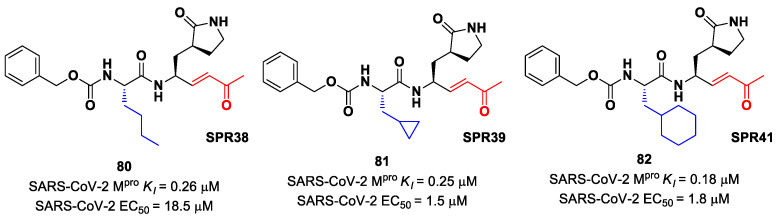
Chemical structure and biological activity of compounds **80**–**82**.

**Figure 52 biomolecules-13-01339-f052:**
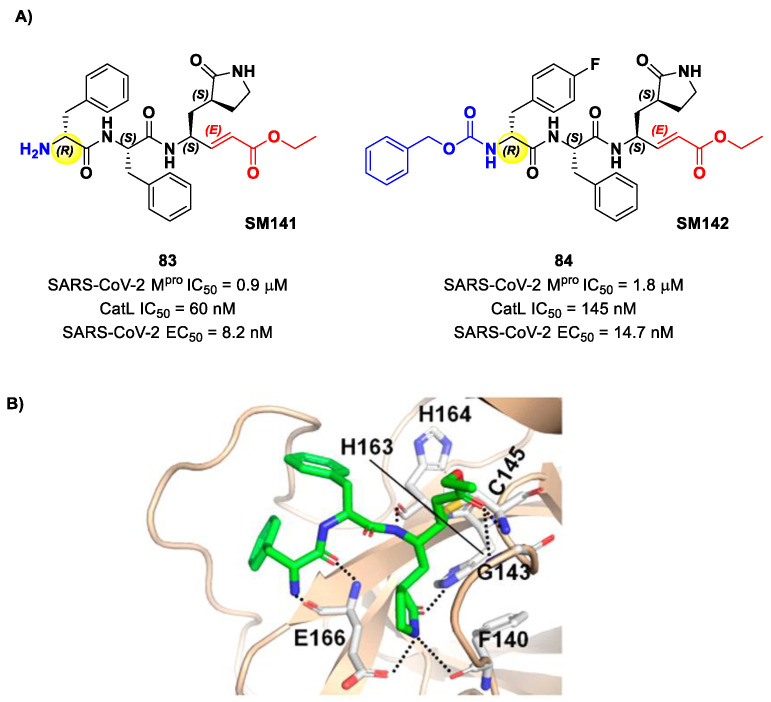
(**A**) Chemical structure and biological activity of compounds **83** and **84**; (**B**) X-ray structure of SARS-CoV-2M^pro^ in complex with **83** (PDB: 7MB0).

**Figure 53 biomolecules-13-01339-f053:**
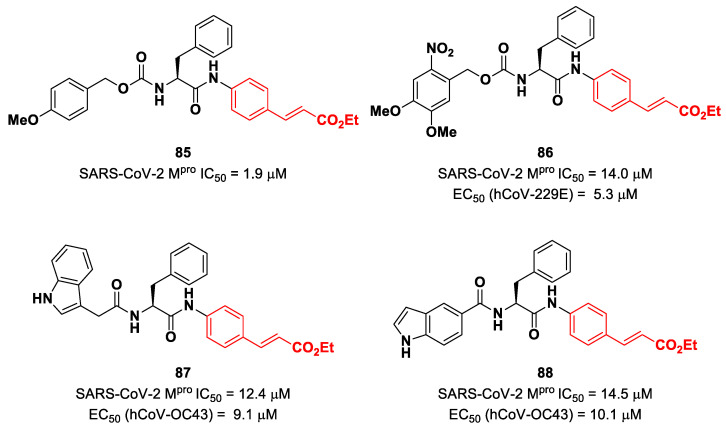
Chemical structure and biological activity of compounds **85**–**88**.

**Figure 54 biomolecules-13-01339-f054:**
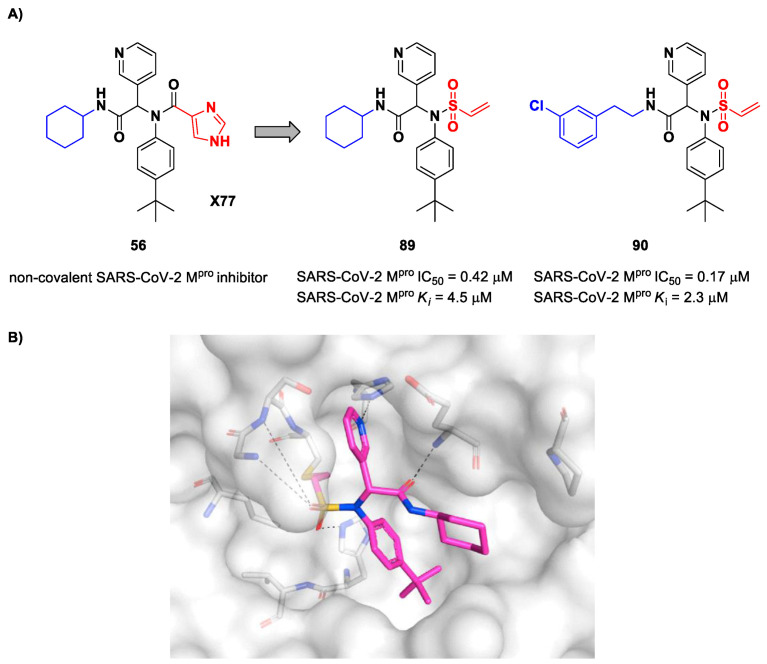
(**A**) Chemical structure and biological activity of compounds **89** and **90**; (**B**) X-ray structure of SARS-CoV-2 M^pro^ in complex with compound **89** (PDB: 7MLG).

**Figure 55 biomolecules-13-01339-f055:**
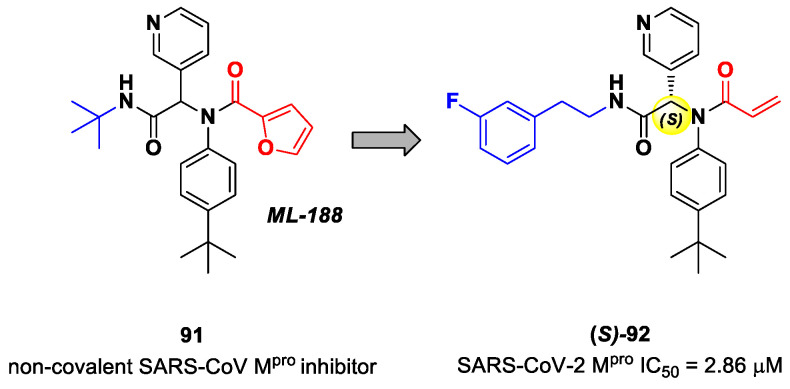
Chemical structure and biological activity of compounds **91** and **92**.

**Figure 56 biomolecules-13-01339-f056:**
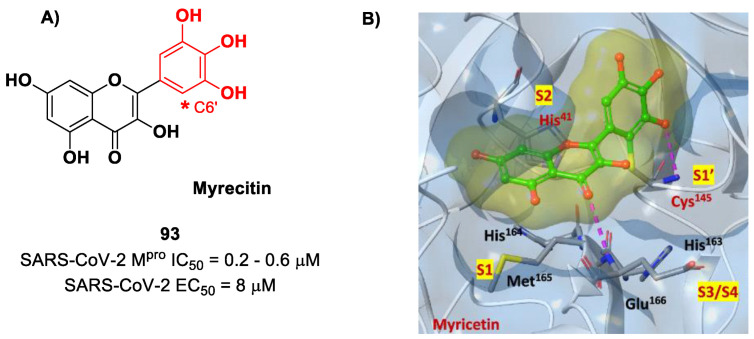
(**A**) Chemical structure and biological activity of compound **93**; (**B**) X-ray crystal structure of SARS-CoV-2 M^pro^ in complex with **93** (PDB: 7B3E).

**Figure 57 biomolecules-13-01339-f057:**
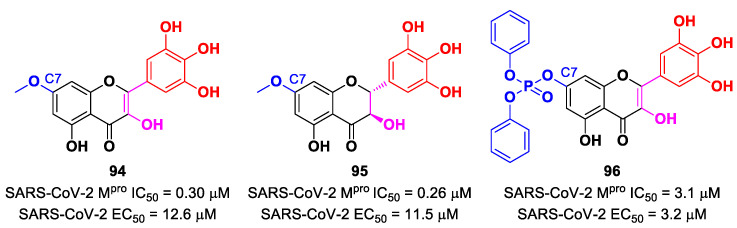
Chemical structure and biological activity of compounds **94**–**96**.

**Figure 58 biomolecules-13-01339-f058:**
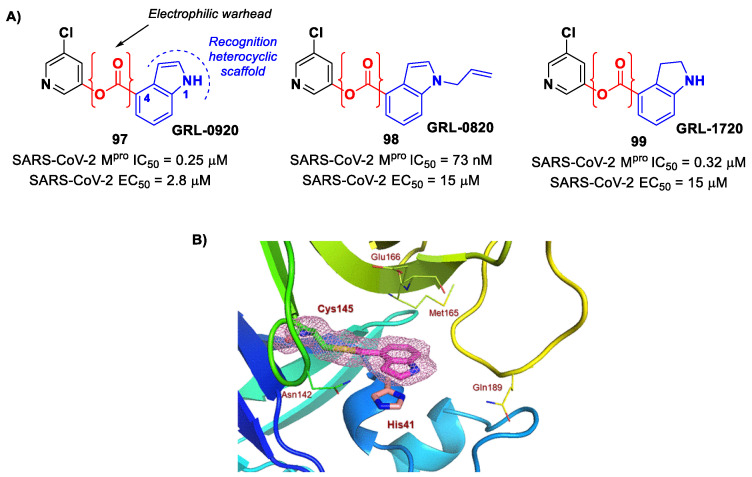
(**A**) Chemical structure and biological activity of compounds **97**–**99**; (**B**) X-ray crystal structure of SARS-CoV-2 M^pro^ in complex with **97** (PDB: 7RBZ).

**Figure 59 biomolecules-13-01339-f059:**
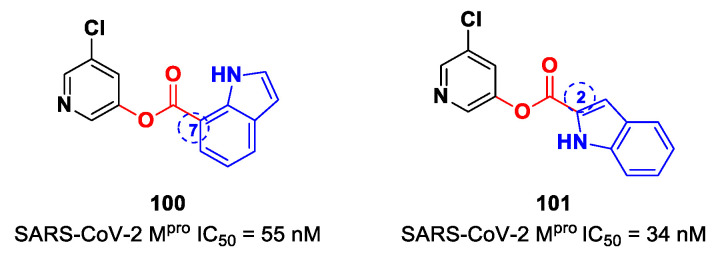
Chemical structure and biological activity of compounds **100** and **101**.

**Figure 60 biomolecules-13-01339-f060:**
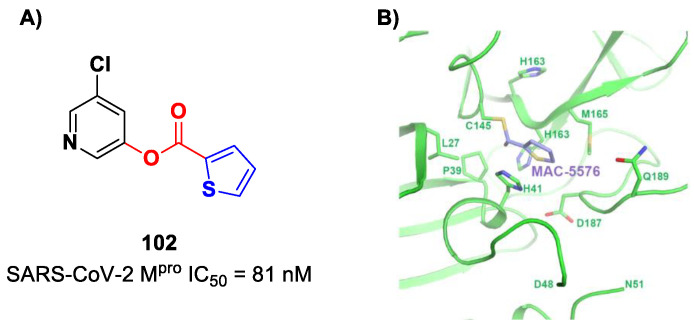
(**A**) Chemical structure and biological activity of compound **102**; (**B**) X-ray crystal structure of SARS-CoV-2 M^pro^ in complex with **102** (PDB: 7JT0).

**Figure 61 biomolecules-13-01339-f061:**
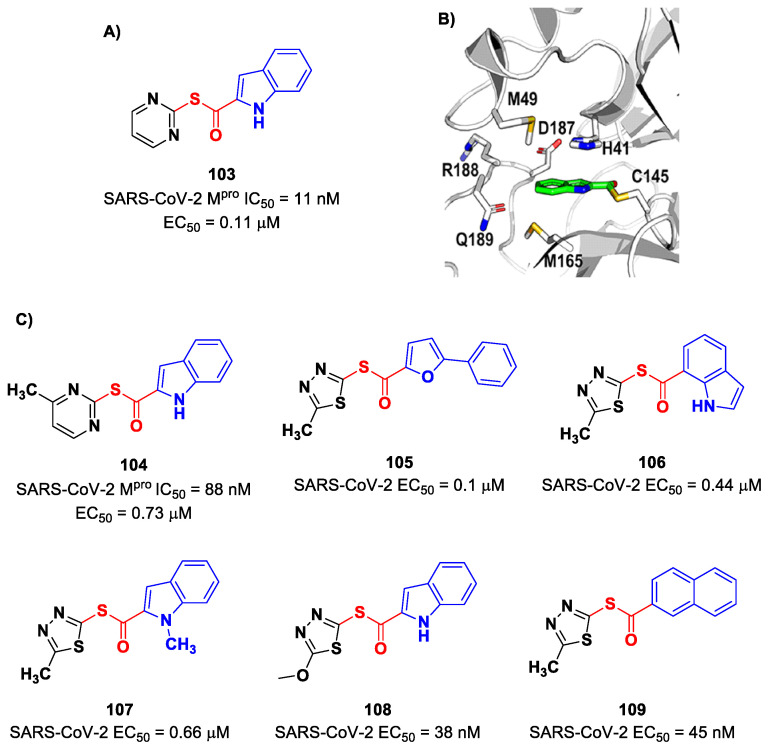
(**A**,**C**) Chemical structure and biological activity of compounds **103**–**109**; (**B**) X-ray crystal structure of SARS-CoV-2 M^pro^ in complex with **103** (PDB: 7X6K).

**Figure 62 biomolecules-13-01339-f062:**
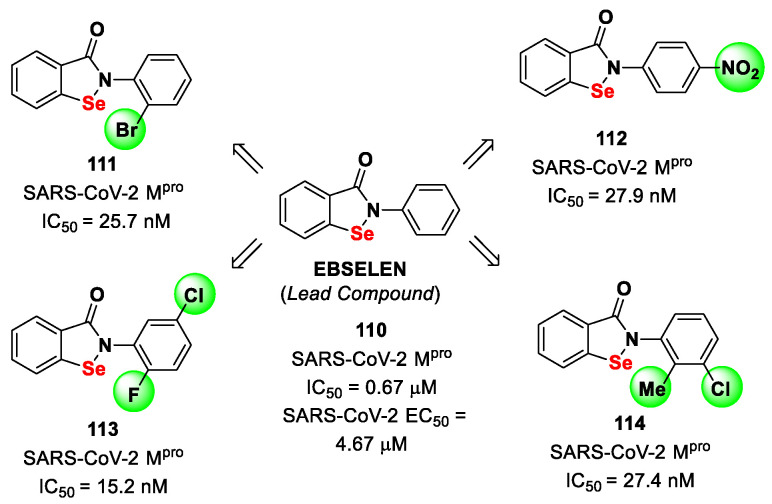
Chemical structure and biological activity of compounds **110**–**114**.

**Figure 63 biomolecules-13-01339-f063:**
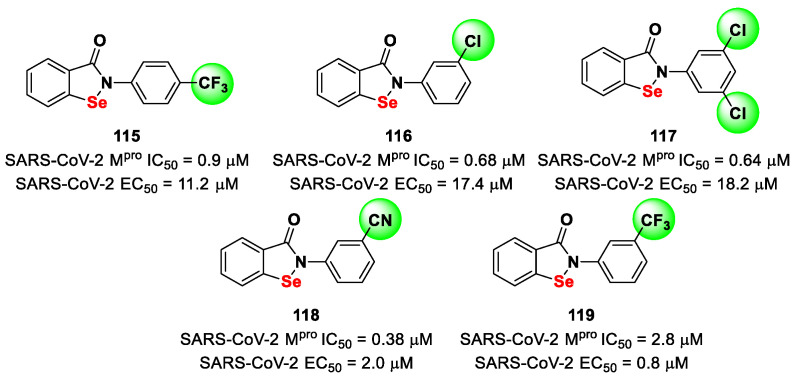
Chemical structure and biological activity of compounds **115**–**119**.

**Figure 64 biomolecules-13-01339-f064:**
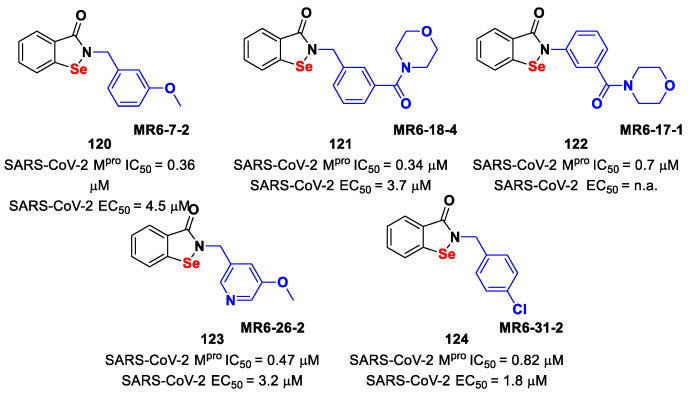
Chemical structure and biological activity of compounds **120**–**124**.

**Figure 65 biomolecules-13-01339-f065:**
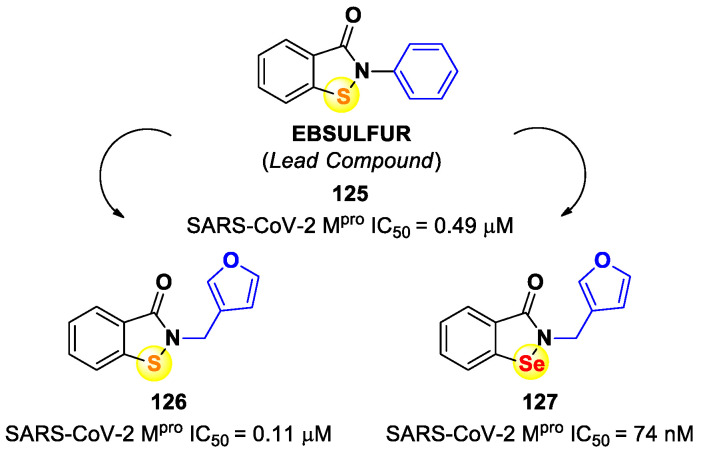
Chemical structure and biological activity of **Ebsulfur**, **126** and **127**.

**Figure 66 biomolecules-13-01339-f066:**
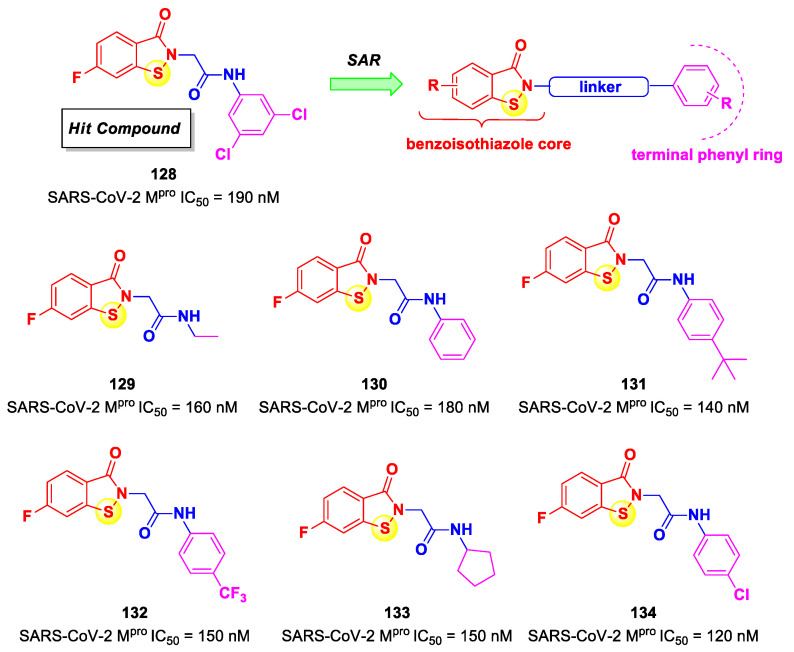
Chemical structure and biological activity of compounds **128**–**134**.

**Figure 67 biomolecules-13-01339-f067:**
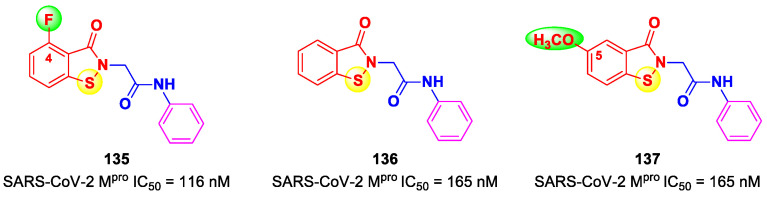
Chemical structure and biological activity of compounds **135**–**137**.

**Figure 68 biomolecules-13-01339-f068:**
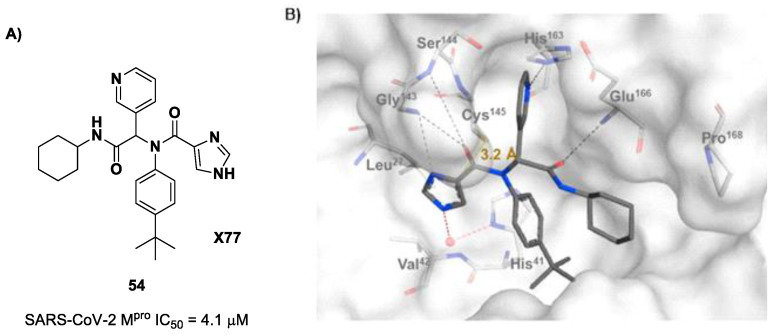
(**A**) Chemical structure and biological activity of the compound **54**; (**B**) X-ray crystal structure of SARS-CoV-2 M^pro^ in complex with **54** (PDB: 6W63), highlighting binding site interactions.

**Figure 69 biomolecules-13-01339-f069:**
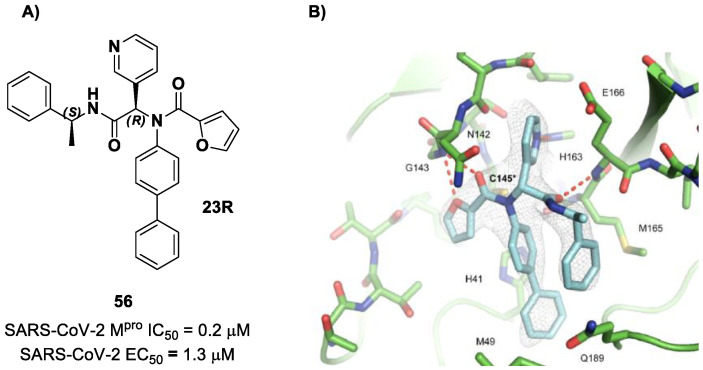
(**A**) Chemical structure and biological activity of the compound **56**; (**B**) X-ray crystal structure of SARS-CoV-2 M^pro^ in complex with **56** (PDB: 7KX5).

**Figure 70 biomolecules-13-01339-f070:**
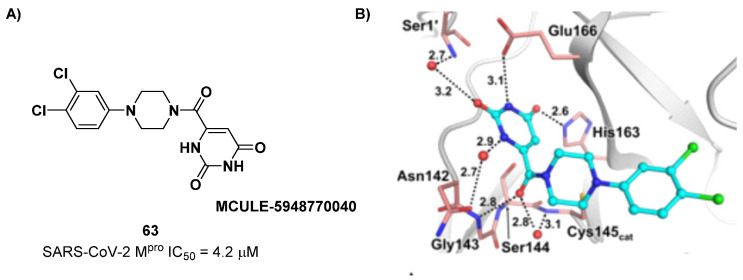
(**A**) Chemical structure and biological activity of compound **63**; (**B**) X-ray structure of SARS-CoV-2 M^pro^ in complex with **63** (PDB: 7LTJ).

**Figure 71 biomolecules-13-01339-f071:**
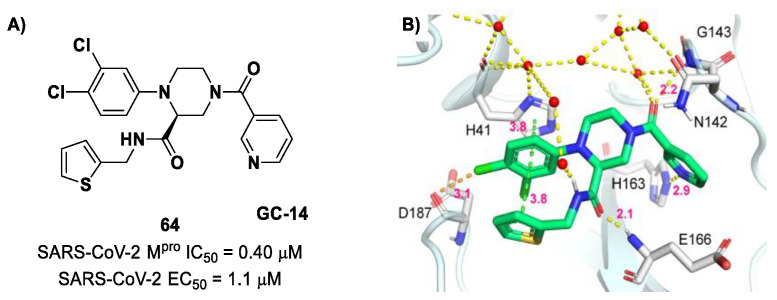
(**A**) Chemical structure and biological activity of compound **64**; (**B**) X-ray structure of SARS-CoV-2 M^pro^ in complex with **64** (PDB: 8ACL).

**Figure 72 biomolecules-13-01339-f072:**
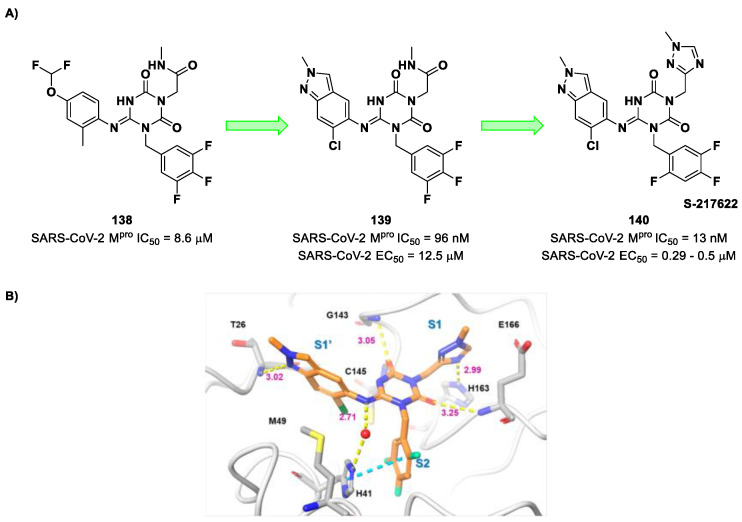
(**A**) Chemical structure and biological activity of compounds **138**–**140**; (**B**) X-ray crystal structure of SARS-CoV-2 M^pro^ in complex with **140** (PDB: 7VU6).

**Figure 73 biomolecules-13-01339-f073:**
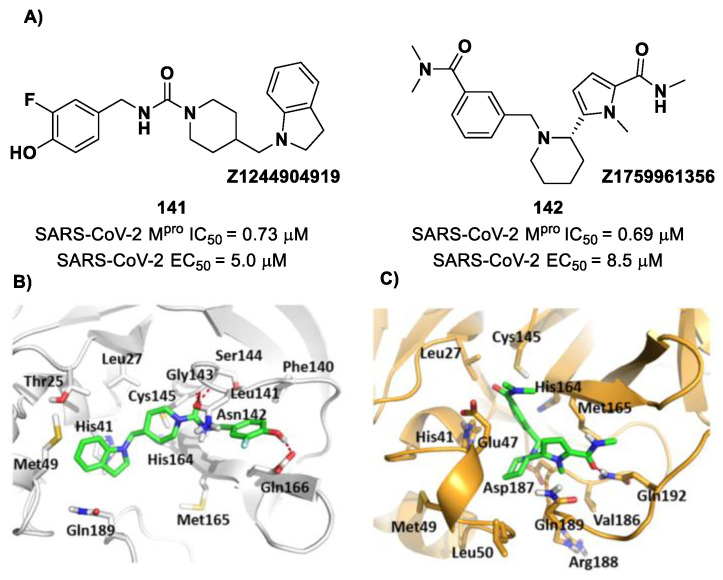
(**A**) Chemical structure and biological activity of compounds **141** and **142**; (**B**) docking of SARS-CoV-2 M^pro^ in complex with **141**; (**C**) docking of SARS-CoV-2 M^pro^ in complex with **142**.

**Figure 74 biomolecules-13-01339-f074:**
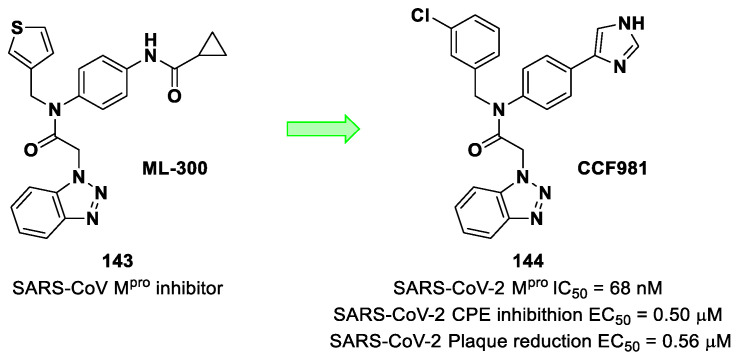
Chemical structure and biological activity of compounds **143** and **144**.

**Figure 75 biomolecules-13-01339-f075:**
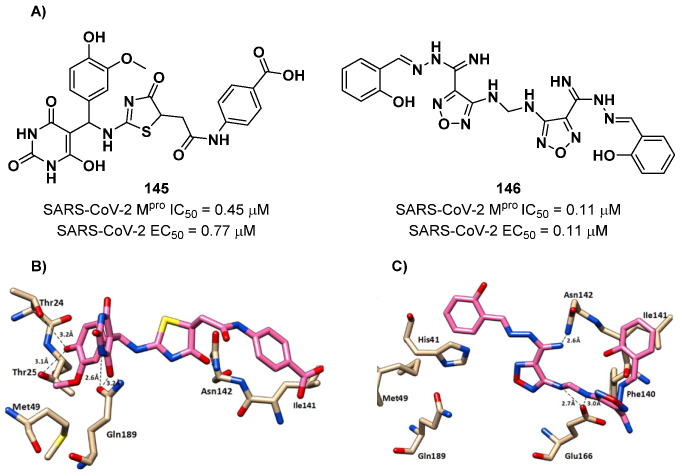
(**A**) Chemical structure and biological activity of compounds **145** and **146**. (**B**) Docking of SARS-CoV-2 M^pro^ in complex with **145**; (**C**) docking of SARS-CoV-2 M^pro^ in complex with **146**.

**Figure 76 biomolecules-13-01339-f076:**
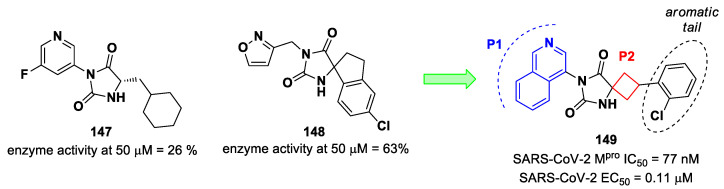
Chemical structure and biological activity of compounds **147**–**149**.

**Figure 77 biomolecules-13-01339-f077:**
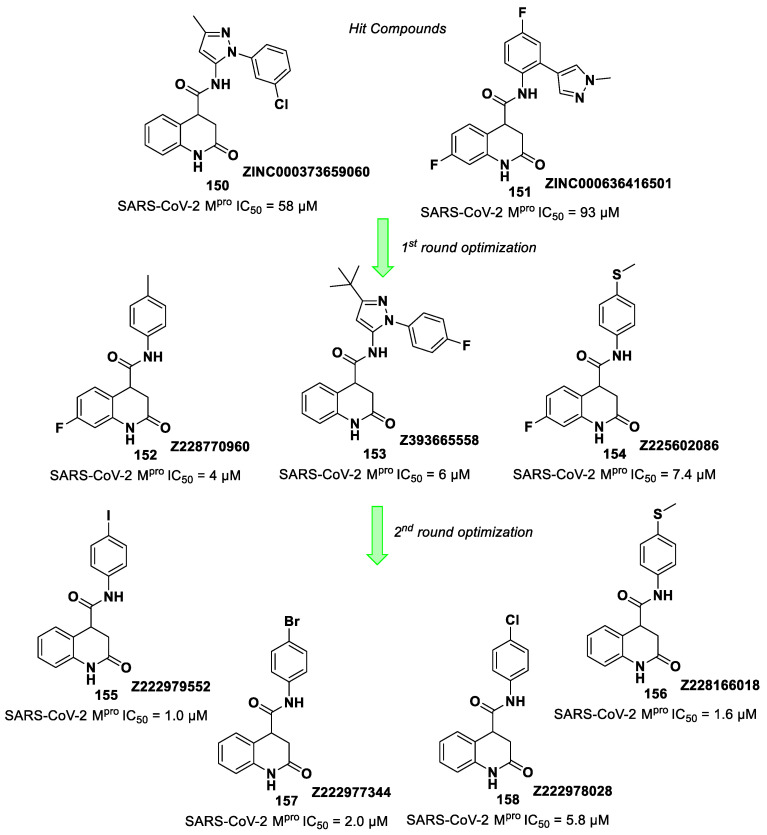
Chemical structure and biological activity of compounds **150**–**158**.

**Figure 78 biomolecules-13-01339-f078:**
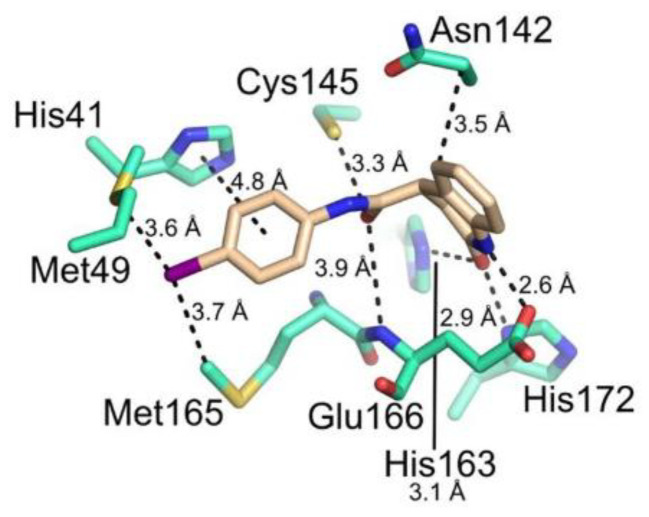
X-ray crystal structure of SARS-CoV-2 M^pro^ in complex with **155** (PDB:7P2G).

**Figure 79 biomolecules-13-01339-f079:**
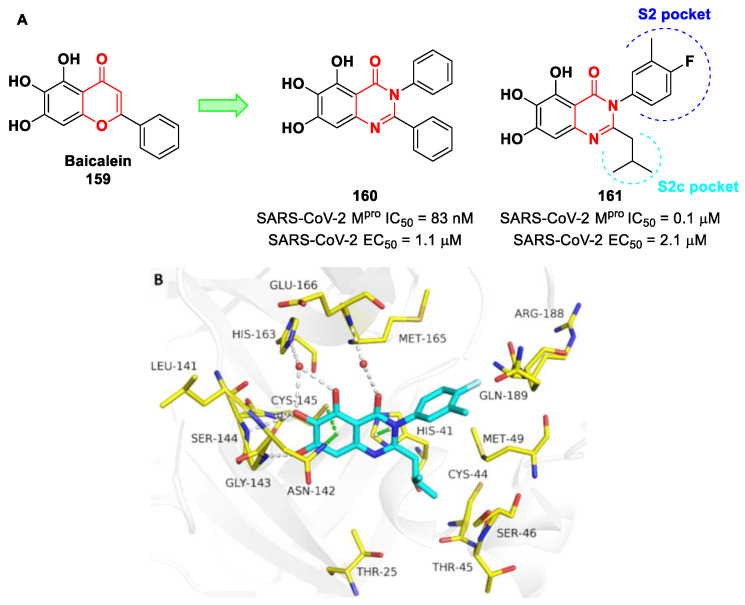
(**A**) Chemical structure and biological activity of compounds **159**–**161.** (**B**) X-ray crystal structure of **161** in complex with SARS-CoV-2 M^pro^ (PDB: 8I4S).

**Figure 80 biomolecules-13-01339-f080:**
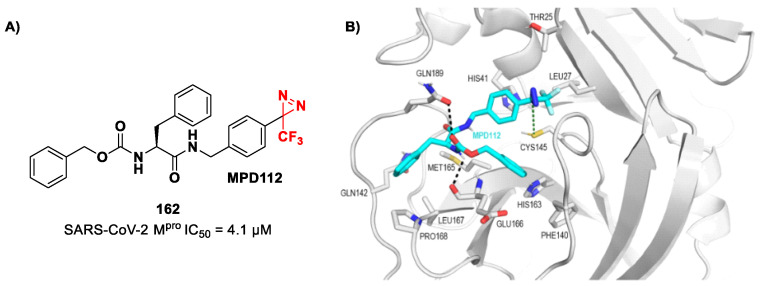
(**A**) Chemical structure and biological activity of compound **159**; (**B**) SARS-CoV-2 M^pro^/**162** complex suggested by molecular docking.

**Figure 81 biomolecules-13-01339-f081:**
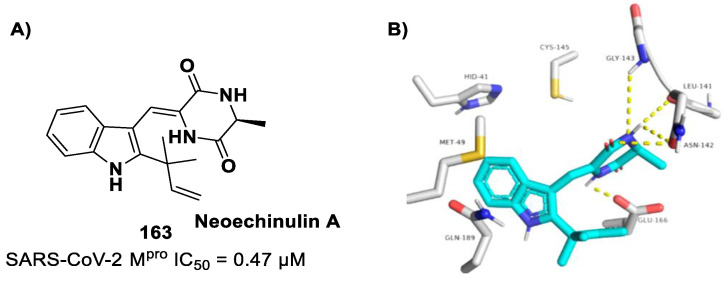
(**A**) Chemical structure and biological activity of compound **163**. (**B**) Docking of SARS-CoV-2 M^pro^ in complex with **163**.

**Figure 82 biomolecules-13-01339-f082:**
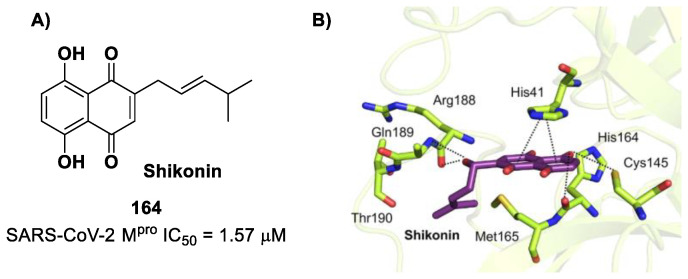
(**A**) Chemical structure and biological activity of compounds **164**; (**B**) X-ray crystal structure of SARS-CoV-2 M^pro^ in complex with **164** (PDB: 7CA8).

**Figure 83 biomolecules-13-01339-f083:**
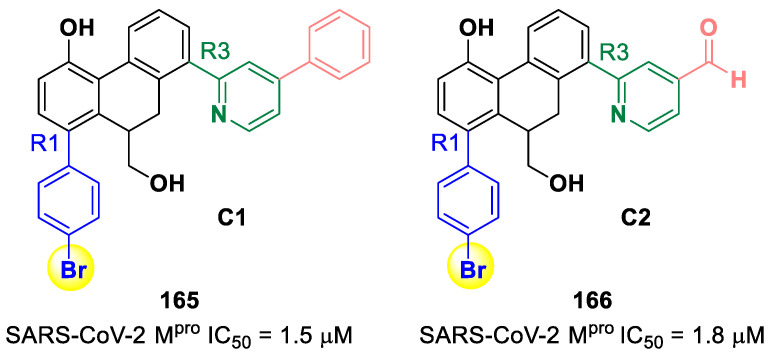
Chemical structure and biological activity of compound **165** and **166**.

**Figure 84 biomolecules-13-01339-f084:**
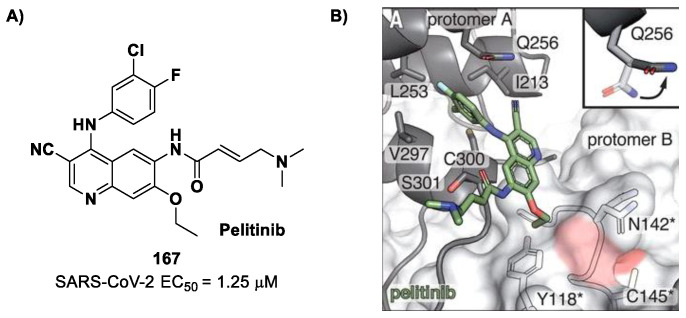
(**A**) Chemical structure and biological activity of compound **167**; (**B**) X-ray crystal structure of SARS-CoV-2 M^pro^ in complex with **167** (PDB: 7AXM).

**Figure 85 biomolecules-13-01339-f085:**
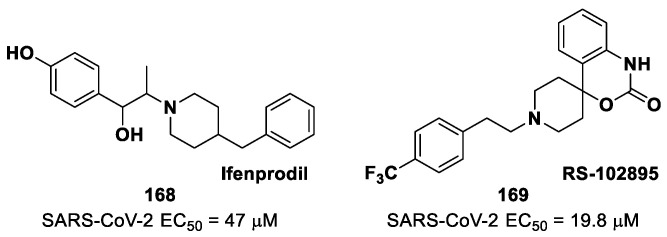
Chemical structure and biological activity of compounds **168** and **169**.

**Figure 86 biomolecules-13-01339-f086:**
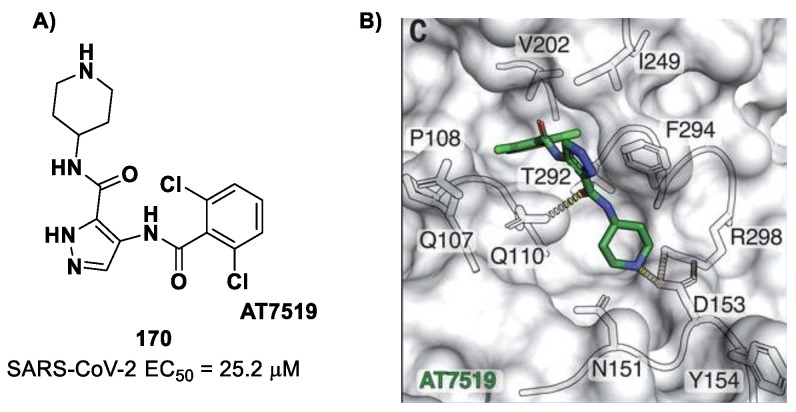
(**A**) Chemical structure and biological activity of compound **170**; (**B**) X-ray crystal structure of SARS-CoV-2 M^pro^ in complex with **170** (PDB: 7AGA).

**Figure 87 biomolecules-13-01339-f087:**
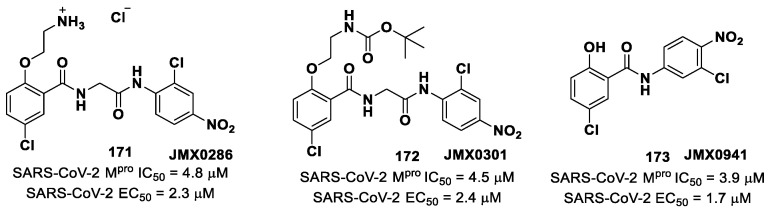
Chemical structure and biological activity of compounds **171**–**173**.

**Figure 88 biomolecules-13-01339-f088:**
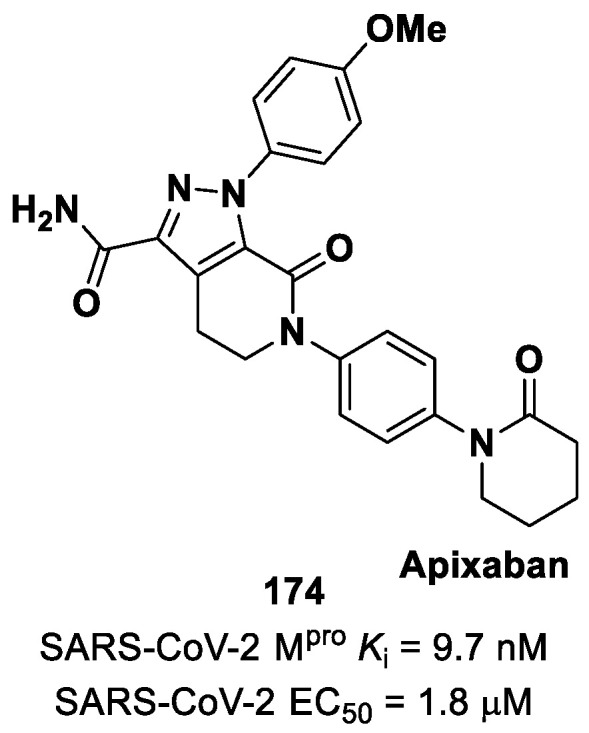
Chemical structure and biological activity of compound **174**.

## Data Availability

No new data are furnished with the present Review.

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
