# Peer review of "Recent Advances in SARS-CoV-2 Main Protease Inhibitors: From Nirmatrelvir to Future Perspectives"

_biomolecules, 2023, doi:10.3390/biom13091339_

Round 1
Reviewer 1 Report
In the review ‘Recent advances on SARS-CoV-2 Mpro inhibitors: from Nirmatrelvir to future perspectives’ Andrea Citarella and coauthors provides an overview of recent advances in the development of SARS-CoV-2 Mpro inhibitors for the treatment of COVID-19. They discuss the role of Mpro in the replication of the virus and the potential of disrupting its catalytic activity. They also highlight some of the most promising inhibitors currently being studied, as well as future perspectives for the development of more effective treatments. The review article is structured, and easy to follow.
However, in my opinion the review can be further enhanced if text information can be tabulated to be more attractive to the readers.
It might be useful to add one or more tables summarizing findings of major published studies, for example-
· Table for In vitro/In Vivo Inhibitors: Inhibitor Name in study, Chemical structure, Potency in biochemical assays (IC50), Potency in cellular assays (EC50), SI, Discovery method, Targeted subsites, References
· Table for in silico inhibitors: Name, PDB code, PDB entry, Chemical formula, Molecule name, Chemical structure, etc.
Furthermore, a brief paragraph describing cell-based, biochemical, in-silico and in-vivo protease inhibitors screening models might also be useful.
Lot of missing current citations were found.
Author Response
Response to Reviewers
Reviewer n°1:
Comments and Suggestions for Authors
In the review ‘Recent advances on SARS-CoV-2 Mpro inhibitors: from Nirmatrelvir to future perspectives’ Andrea Citarella and coauthors provides an overview of recent advances in the development of SARS-CoV-2 Mpro inhibitors for the treatment of COVID-19. They discuss the role of Mpro in the replication of the virus and the potential of disrupting its catalytic activity. They also highlight some of the most promising inhibitors currently being studied, as well as future perspectives for the development of more effective treatments. The review article is structured, and easy to follow.
- We sincerely thank the reviewer for its comments.
However, in my opinion the review can be further enhanced if text information can be tabulated to be more attractive to the readers.
It might be useful to add one or more tables summarizing findings of major published studies, for example-
- Table for In vitro/In Vivo Inhibitors: Inhibitor Name in study, Chemical structure, Potency in biochemical assays (IC50), Potency in cellular assays (EC50), SI, Discovery method, Targeted subsites, References
- Table for in silico inhibitors: Name, PDB code, PDB entry, Chemical formula, Molecule name, Chemical structure, etc.
- We thank the reviewer for its suggestions. A comprehensive table for In vitro/In vivo inhibitors is now included in the Supplementary material with the specifics suggested by the Reviewer. All the information requested by the Reviewer has been included except for SI. This is because the cytotoxicity data are not always present in the cited papers for the related inhibitor and the healthy cell lines employed for this selectivity determination are often different. Besides, quite often the authors calculated Ki instead of IC50 for Mpro So, we determined that this information could me somehow misleading. No in silico inhibitors are discussed in the review as this topic is out of the scope of this Special Issue (State of the art in antiviral therapy and viral diseases diagnosis: an update to 2022-2023). Therefore, we decided to focus only on literature compounds possessing biological activity.
Furthermore, a brief paragraph describing cell-based, biochemical, in-silico and in-vivo protease inhibitors screening models might also be useful.
- This review is a follow-up of our previous review article (Biomolecules 2021, 11, 607) in which we already reported detailed sections for both enzymatic assays (section #4) and cell-based assays (section). So, it would be repetitive to write down again the same sections. In-silico screening models are out of the scope of this special issue as explained above. We also think that reporting the in-vivo model protocols is not in line with the focus of this review article. This is a specific pharmacology matter (which is not in our field of expertise), and the animal protocols are quite different (mouse, rat, monkey models). Anyway, we properly cited the research articles pertaining these models. We hope this is sufficient.
Lot of missing current citations were found.
- We included new citations coming from the latest research works published during this summer as suggested also by another Reviewer.

Reviewer 2 Report
The manuscript entitled « Recent advances on SARS-CoV-2 Mpro inhibitors: from Nirma-2 trelvir to future perspectives” by Citarella et al. is a review providing the last updates of the development of SARS-CoV-2 Main Protease (Mpro) inhibitors. After a clear description of the Mpro characteristics that make it suitable for anti-viral drug screening, the authors describe the different class of Mpro inhibitors with great details of the chemical and structural point of view. The authors provide a comprehensive catalogue of the most promising Mpro inhibitors and describe their structure and biological properties. The review gives an idea of the scope of the work accomplished to develop Mpro inhibitors as therapeutic targets. Starting from Nirmatrelvir, the first Mpro inhibitor administrated in the Paxlovid®, the first anti-SARS-CoV-2 oral drug, the review provides an updated state-of-the-art of COVID-19 chemical treatments. Such treatments are also developed in view of other coronaviruses infection and future pandemics. In all, it seems to me as being a solid and comprehensive review which provides detailed information insightful regarding SARS CoV-2 anti-viral therapeutics.
Minor errors in the text have been identified, the list is provided below :
- lane 213 : “led” and not “lead”
- lane 375 : “pockect” misspelled
- Figure 10 : labels A and B are misplaced
- Figure 13B: the Gly143, Ser144 residues described in the text are not seen.
- lane 353: cite Figure 14B
- lane 401 “compounds”
- lane 451 remove “in”
- lane 582 remove Mpro from the designation of SARS-CoV-2, apply to all other occurences
- lane 583 : explain what is the role a P-glycoprotein inhibitor here
- lane 925 : “moderately” misspelled
- lane 940: “ranged” misspelled
- lane 946: “replaced” misspelled
- lane 997: it lacks a word between “most” and “derivatives”
- lane 1049 : it lacks “inhibitors” after Mpro
- lane 1090: change reference citation format
- lane 1124: “series” misspelled
- lane 1143:”inhibitors” misspelled
- lane 1181: “the” misspelled
- lane 1184: remove “in”
- lane 1284: remove “driven by” or “ guided by”
- lane 1286: “emerged” misspelled
- lane 1400: change reference citation format
- reference 107 : change DOI
Author Response
Response to Reviewers
Reviewer n°2:
Comments and Suggestions for Authors
The manuscript entitled « Recent advances on SARS-CoV-2 Mpro inhibitors: from Nirma-2 trelvir to future perspectives” by Citarella et al. is a review providing the last updates of the development of SARS-CoV-2 Main Protease (Mpro) inhibitors. After a clear description of the Mpro characteristics that make it suitable for anti-viral drug screening, the authors describe the different class of Mpro inhibitors with great details of the chemical and structural point of view. The authors provide a comprehensive catalogue of the most promising Mpro inhibitors and describe their structure and biological properties. The review gives an idea of the scope of the work accomplished to develop Mpro inhibitors as therapeutic targets. Starting from Nirmatrelvir, the first Mpro inhibitor administrated in the Paxlovid®, the first anti-SARS-CoV-2 oral drug, the review provides an updated state-of-the-art of COVID-19 chemical treatments. Such treatments are also developed in view of other coronaviruses infection and future pandemics. In all, it seems to me as being a solid and comprehensive review which provides detailed information insightful regarding SARS CoV-2 anti-viral therapeutics.
- We sincerely thank the reviewer for its very positive feedback.
Minor errors in the text have been identified, the list is provided below :
- lane 213 : “led” and not “lead”
- lane 375 : “pockect” misspelled
- Typo corrected.
- Figure 10 : labels A and B are misplaced
- Revised accordingly.
- Figure 13B: the Gly143, Ser144 residues described in the text are not seen.
- Corrected in the main text.
- lane 353: cite Figure 14B
- lane 401 “compounds”
- lane 451 remove “in”
- lane 582 remove Mpro from the designation of SARS-CoV-2, apply to all other occurrences
- Corrected accordingly.
- lane 583 : explain what is the role a P-glycoprotein inhibitor here
- Done as suggested.
- lane 925 : “moderately” misspelled
- Typo corrected.
- lane 940: “ranged” misspelled
- Typo corrected.
- lane 946: “replaced” misspelled
- Typo corrected.
- lane 997: it lacks a word between “most” and “derivatives”
- Revised accordingly.
- lane 1049 : it lacks “inhibitors” after Mpro
- Revised accordingly.
- lane 1090: change reference citation format
- lane 1124: “series” misspelled
- Typo corrected.
- lane 1143:”inhibitors” misspelled
- Typo corrected.
- lane 1181: “the” misspelled
- Typo corrected.
- lane 1184: remove “in”
- lane 1284: remove “driven by” or “ guided by”
- Revised accordingly.
- lane 1286: “emerged” misspelled
- Typo corrected.
- lane 1400: change reference citation format
- reference 107 : change DOI
- Done accordingly.
Reviewer 3 Report
The main protease (Mpro) of SARS-CoV-2 is an attractive target in antiCOVID-19 therapy for its high conservation and major role in the virus life cycle. The covalent Mpro inhibitor nirmatrelvir (in combination with ritonavir, a pharmacokinetic enhancer) and the non-covalent inhibitor ensitrelvir have shown efficacy in clinical trials and have been approved for therapeutic use. Effective antiviral drugs are needed to fight the pandemic, while Mpro inhibitors could be promising alternatives due to their high selectivity and favorable druggability. Numerous Mpro inhibitors with desirable properties have been developed based on available crystal structures of Mpro. In this article, the authors describe medicinal chemistry strategies applied for the discovery and optimization of Mpro inhibitors, followed by a general overview and critical analysis of the available information. Prospective viewpoints and insights into current strategies for the development of Mpro inhibitors are also discussed.
This is a superb review article, and I believe its publication will have a beneficial impact on the field.
I have two suggestions:
1. I suggest that the author add some of the latest literature in this field.
[1] https://pubs.rsc.org/en/content/articlelanding/2023/md/d3md00306j
[2] Ngo C, Fried W, Aliyari S, Feng J, Qin C, Zhang S, Yang H, Shanaa J, Feng P, Cheng G, Chen XS, Zhang C. Alkyne as a Latent Warhead to Covalently Target SARS-CoV-2 Main Protease. J Med Chem. 2023 Aug 18. doi: 10.1021/acs.jmedchem.3c00810. Epub ahead of print. PMID: 37595260.
[3] Ren P, Li H, Nie T, Jian X, Yu C, Li J, Su H, Zhang X, Li S, Yang X, Peng C, Yin Y, Zhang L, Xu Y, Liu H, Bai F. Discovery and Mechanism Study of SARS-CoV-2 3C-like Protease Inhibitors with a New Reactive Group. J Med Chem. 2023 Aug 18. doi: 10.1021/acs.jmedchem.3c00818. Epub ahead of print. PMID: 37594952.
[4] Geng ZZ, Atla S, Shaabani N, Vulupala V, Yang KS, Alugubelli YR, Khatua K, Chen PH, Xiao J, Blankenship LR, Ma XR, Vatansever EC, Cho CD, Ma Y, Allen R, Ji H, Xu S, Liu WR. A Systematic Survey of Reversibly Covalent Dipeptidyl Inhibitors of the SARS-CoV-2 Main Protease. J Med Chem. 2023 Aug 10. doi: 10.1021/acs.jmedchem.3c00221. Epub ahead of print. PMID: 37561993.
[5] https://www.sciencedirect.com/science/article/pii/S2211383523002939
2. I suggest that the author search the relevant patents, provide them in the form of a list or attachment, and make a brief comment on the patents. In my opinion, if the inhibitor research of a certain target ignores the patent, it will be incomplete.
Author Response
Response to Reviewers
Reviewer n°3:
Comments and Suggestions for Authors
The main protease (Mpro) of SARS-CoV-2 is an attractive target in antiCOVID-19 therapy for its high conservation and major role in the virus life cycle. The covalent Mpro inhibitor nirmatrelvir (in combination with ritonavir, a pharmacokinetic enhancer) and the non-covalent inhibitor ensitrelvir have shown efficacy in clinical trials and have been approved for therapeutic use. Effective antiviral drugs are needed to fight the pandemic, while Mpro inhibitors could be promising alternatives due to their high selectivity and favorable druggability. Numerous Mpro inhibitors with desirable properties have been developed based on available crystal structures of Mpro. In this article, the authors describe medicinal chemistry strategies applied for the discovery and optimization of Mpro inhibitors, followed by a general overview and critical analysis of the available information. Prospective viewpoints and insights into current strategies for the development of Mpro inhibitors are also discussed.
This is a superb review article, and I believe its publication will have a beneficial impact on the field.
- We sincerely thank the reviewer for its comments.
I have two suggestions:
- I suggest that the author add some of the latest literature in this field.
[1] https://pubs.rsc.org/en/content/articlelanding/2023/md/d3md00306j
[2] Ngo C, Fried W, Aliyari S, Feng J, Qin C, Zhang S, Yang H, Shanaa J, Feng P, Cheng G, Chen XS, Zhang C. Alkyne as a Latent Warhead to Covalently Target SARS-CoV-2 Main Protease. J Med Chem. 2023 Aug 18. doi: 10.1021/acs.jmedchem.3c00810. Epub ahead of print. PMID: 37595260.
[3] Ren P, Li H, Nie T, Jian X, Yu C, Li J, Su H, Zhang X, Li S, Yang X, Peng C, Yin Y, Zhang L, Xu Y, Liu H, Bai F. Discovery and Mechanism Study of SARS-CoV-2 3C-like Protease Inhibitors with a New Reactive Group. J Med Chem. 2023 Aug 18. doi: 10.1021/acs.jmedchem.3c00818. Epub ahead of print. PMID: 37594952.
[4] Geng ZZ, Atla S, Shaabani N, Vulupala V, Yang KS, Alugubelli YR, Khatua K, Chen PH, Xiao J, Blankenship LR, Ma XR, Vatansever EC, Cho CD, Ma Y, Allen R, Ji H, Xu S, Liu WR. A Systematic Survey of Reversibly Covalent Dipeptidyl Inhibitors of the SARS-CoV-2 Main Protease. J Med Chem. 2023 Aug 10. doi: 10.1021/acs.jmedchem.3c00221. Epub ahead of print. PMID: 37561993.
[5] https://www.sciencedirect.com/science/article/pii/S2211383523002939
- We thank the Reviewer for the suggestions. We previously did not include these papers because they came out during last month, after the submission of the review. We have now included all these interesting findings in the new version of the review article.
- I suggest that the author search the relevant patents, provide them in the form of a list or attachment, and make a brief comment on the patents. In my opinion, if the inhibitor research of a certain target ignores the patent, it will be incomplete.
- We thank the reviewer for its suggestions. We are aware that the progress of the research published as a patent has represented an important contribution in the development of Mpro However, in our review we have decided to focus on the inhibitors in the literature for reasons of length and completeness. To pay homage to patent research, we have included a few sentences about it and cited a very complete and detailed patent review on the recent developments of Mpro inhibitors.
Round 2
Reviewer 3 Report
Accept in present form